# New Formulas of Feedback Capacity for AGN Channels with Memory: A Time-Domain Sufficient Statistic Approach [note 1]

**DOI:** 10.3390/e27020207

**Published:** 2025-02-15

**Authors:** Charalambos D. Charalambous, Christos Kourtellaris, Stelios Louka

**Affiliations:** Department of Electrical and Computer Engineering, University of Cyprus, P.O. Box 20537, Nicosia CY-1678, Cyprus; kourtellaris.christos@ucy.ac.cy (C.K.);

**Keywords:** feedback capacity, non-stationary, unstable, Gaussian channels, memory, sufficient statistic

## Abstract

Recently, several papers identified technical issues related to equivalent time-domain and frequency-domain “characterization of the *n*–block or transmission” feedback capacity formula and its asymptotic limit, the feedback capacity, of additive Gaussian noise (AGN) channels, first introduce by Cover and Pombra in 1989 (IEEE Transactions on Information Theory). The main objective of this paper is to derive new results on the Cover and Pombra characterization of the *n*–block feedback capacity formula, and to clarify the main points of confusion regarding the time-domain results that appeared in the literature. The first part of this paper derives new equivalent time-domain sequential characterizations of feedback capacity of AGN channels driven by non-stationary and non-ergodic Gaussian noise. It is shown that the optimal channel input processes of the new equivalent sequential characterizations are expressed as functionals of a *sufficient statistic and a Gaussian orthogonal innovations process.* Further, the Cover and Pombra *n*–block capacity formula is expressed as a functional of two generalized matrix difference Riccati equations (DREs) of the filtering theory of Gaussian systems, contrary to results that appeared in the literature and involve only one DRE. It is clarified that prior literature deals with a simpler problem that presupposes the state of the noise is known to the encoder and the decoder. In the second part of this paper, the existence of the asymptotic limit of the *n*–block feedback capacity formula is shown to be equivalent to the convergence properties of solutions of the two generalized DREs. Further, necessary and or sufficient conditions are identified for the existence of asymptotic limits, for stable and unstable Gaussian noise, when the optimal input distributions are asymptotically time-invariant but not necessarily stationary. This paper contains an in-depth analysis, with various examples, and identifies the technical conditions on the feedback code and state space noise realization, so that the time-domain capacity formulas that appeared in the literature, for AGN channels with stationary noises, are indeed correct.

## 1. Introduction, Motivation, Main Results, Current State of Knowledge

In the recent papers [1,2,3], concerns are raised whether the time-domain analysis in [4], deals with the Cover and Pombra [5] “characterization of the *n*–block or transmission” feedback capacity formula and its asymptotic limit, the feedback capacity, of additive Gaussian noise (AGN) channels. Furthermore, the recent comment paper [6] identified gaps in the proof of the simplified frequency-domain characterization of Theorem 4.1 in [4]. The main objective of this paper is to derive new results on the Cover and Pombra characterization of the *n*–block feedback capacity formula, and to clarify the main points of confusion regarding the time-domain results of [4] and related literature i.e., [7].

### 1.1. The Problem, Motivation, and Main Results

We consider the additive Gaussian noise (AGN) channel defined by [5](1)$$\begin{array}{c} Y_{t}=X_{t}+V_{t},\;t=1,\ldots ,n,\;\frac{1}{n}E\left\{\sum_{t=1}^{n}{\left(X_{t}\right)}^{2}\right\}\leq \kappa ,\;\kappa \in \left[0,\infty \right) \end{array}$$
where
$X^{n}\overset{▵}{=}\left\{X_{1},X_{2},\ldots ,X_{n}\right\}$ is the sequence of channel input random variables (RVs) $X_{t}:\Omega \to R$;$Y^{n}\overset{▵}{=}\left\{Y_{1},Y_{2},\ldots ,Y_{n}\right\}$ is the sequence of channel output RVs $Y_{t}:\Omega \to R$;$V^{n}\overset{▵}{=}\left\{V_{1},\ldots ,V_{n}\right\}$ is the sequence of jointly Gaussian distributed RVs $V_{t}:\Omega \to R$, with distribution $P_{V^{n}}\left(dv^{n}\right)$, which are not necessarily stationary or ergodic.We wish to examine the feedback capacity of the AGN channel (Equation 1) for two distinct formulations of code definition and a noise model, described below under Case (I) and Case (II) formulations. Special cases of these will be related to the existing literature.


**Case (I) Formulation.** This formulation respects the following two conditions.
(I.1). The feedback code does not assume knowledge of the initial state of the noise at the encoder and the decoder (see Definition 1), and(I.2) the noise sequence $V^{n}$ is represented by a partially observable (partially observable means that knowledge of $V^{t-1}$ and the initial state do not specify the state $S^{t},t=1,\ldots ,n$.) state space realization, with state sequence $S^{n}$ (see Definition 2).

For a formulation that respects (I.1) and (I.2), Cover and Pombra characterized the “*n*–finite transmission” feedback capacity [5] (Equations (10) and (11)), using the information measure ($C_{n}^{fb}\left(\kappa \right)$ is identified using the converse coding theorem [5]),(2)$$\begin{array}{c} C_{n}^{fb}\left(\kappa \right)\overset{▵}{=}\underset{P_{X_{t}|X^{t-1},Y^{t-1}},t=1,\ldots ,n:\;\frac{1}{n}E\left\{\sum_{t=1}^{n}{\left(X_{t}\right)}^{2}\right\}\leq \kappa }{\sup}\sum_{t=1}^{n}H\left(Y_{t}|Y^{t-1}\right)-H\left(V^{n}\right) \end{array}$$
provided the supremum exists, where H(·) denotes (differential) entropy.

**Case (I.a) Formulation.** Although not mentioned in [5], if the feedback code assumes knowledge of the initial state of the noise or the channel, $S_{1}=s$, at the encoder and the decoder (see Definition 3), it follows directly from [5] (Equations (10) and (11)), that (Equation 2) is replaced by the information measure(3)$$\begin{array}{c} C_{n}^{fb}\left(\kappa ,s\right)\overset{▵}{=}\underset{P_{X_{t}|X^{t-1},Y^{t-1},s},t=1,\ldots ,n:\;\frac{1}{n}E\left\{\sum_{t=1}^{n}{\left(X_{t}\right)}^{2}|S_{1}=s\right\}\leq \kappa }{\sup}\sum_{t=1}^{n}H\left(Y_{t}|Y^{t-1},s\right)-H\left(V^{n}|s\right). \end{array}$$

**Case (II) Formulation.** This formulation relaxes Conditions (I.1) and (I.2) to the following two conditions:
(II.1) The feedback code assumes knowledge of the initial state of the noise or the channel, $S_{1}=s$, at the encoder and the decoder (see Definition 3);(II.2) the noise sequence $V^{n}$ is represented by a fully observable state space realization (fully observable means knowledge of $V^{t-1}$ and initial state specify the state $S^{t},t=1,\ldots ,n$), with state sequence $S^{n}$ such that the noise $V^{t-1}$ and the initial state $S_{1}$ uniquely defines the noise state sequence $S^{t}$ and vice versa for t=1,…,n.Thus, Formulation (Equation 2), which respects Conditions (I.1) and (I.2), is the most general.

For a formulation that respects Conditions (II.1) and (II.2), Yang, Kavcic, and Tatikonda [8] characterized the *n*–finite transmission feedback capacity [8] (Section II (particularly Section II.C, I–III)), using the information measure,(4)$$\begin{array}{c} C_{n}^{fb,S}\left(\kappa ,s\right)\overset{▵}{=}\underset{P_{X_{t}|S^{t},Y^{t-1},s},t=1,\ldots ,n:\;\frac{1}{n}E\left\{\sum_{t=1}^{n}{\left(X_{t}\right)}^{2}|S_{1}=s\right\}\leq \kappa }{\sup}\sum_{t=1}^{n}H\left(Y_{t}|Y^{t-1},s\right)-H\left(V^{n}|s\right). \end{array}$$

Compared to $C_{n}^{fb}\left(\kappa \right)$ and $C_{n}^{fb}\left(\kappa ,s\right)$, the definition of $C_{n}^{fb,S}\left(\kappa ,s\right)$ imposes Condition (II.2) and is fundamentally different from the former, because the input distributions of $C_{n}^{fb,S}\left(\kappa ,s\right)$ are different from $C_{n}^{fb}\left(\kappa \right)$ and $C_{n}^{fb}\left(\kappa ,s\right)$. Hence, the information rates of the three formulas are generally different. However, for certain Gaussian noise models of $V^{n}$, it might be the case that, under Condition (II.1), the information measures $C_{n}^{fb}\left(\kappa ,s\right)$ and $C_{n}^{fb,S}\left(\kappa ,s\right)$ coincide. We provide several examples in the main body of this paper.


**Motivation and Fundamental Differences of Case (I) and Case (II) Formulations.**


At this point, we pause to discuss two technical issues of Case (I) and Case (II) formulations, which are not clarified in [4,7,9]. These technical issues are related to the time-domain characterization of feedback capacity, Theorem 6.1 in [4]; they are first discussed in [1,2,3,10]. A recent comment paper [6] also identified gaps in the proof of the frequency-domain characterization of capacity, Theorem 4.1 in [4], which affect the proofs of Theorems 4.1, 4.6, 5.3 and 6.1; Propositions 4.7 and 5.1; Remarks 4.5 and 5.2; and Lemma 6.1 in [4].

To illustrate the technical issues, we consider the autoregressive moving average stable noise denoted by ARMA (a,c),a∈[−1,1],c∈(−1,1),c≠a, studied by many authors [4,7,8,9,11,12], as a benchmark example.(5)$$\begin{array}{cc} \; & V_{t}=cV_{t-1}+W_{t}-aW_{t-1},\;\forall t\in Z_{+}\overset{▵}{=}\left\{1,2,\ldots \right\}, \end{array}$$(6)$$\begin{array}{cc} \; & V_{0}\in N\left(0,K_{V_{0}}\right),\;K_{V_{0}}\geq 0,\;W_{0}\in N\left(0,K_{W_{0}}\right),\;K_{W_{0}}\geq 0,W_{t}\in N\left(0,K_{W}\right),\;K_{W}>0,\;\forall t, \end{array}$$(7)$$\begin{array}{cc} \; & \left\{W_{0},W_{1},\ldots ,W_{n}\right\}\text{indep. seq. and indep. of}\;V_{0}. \end{array}$$
As in [4,7], we define the state variable of the noise by(8)$$\begin{array}{c} S_{t}\overset{▵}{=}\frac{cV_{t-1}-aW_{t-1}}{c-a},\;\forall t\in Z_{+} \end{array}$$
Then, the state space realization of $V^{n}$ is(9)$$\begin{array}{cc} \; & S_{t+1}=cS_{t}+W_{t},\;V_{t}=\left(c-a\right)S_{t}+W_{t},\;\forall t\in Z_{+}, \end{array}$$(10)$$\begin{array}{cc} \; & K_{S_{1}}=\frac{{\left(c\right)}^{2}K_{V_{0}}+{\left(a\right)}^{2}K_{W_{0}}}{{\left(c-a\right)}^{2}},\;K_{V_{0}}\geq 0,\;K_{W_{0}}\geq 0\;\text{both given}. \end{array}$$
Bounds on feedback capacity, when a=0, i.e., corresponding to an autoregressive noise are derived in [13,14,15] using linear coding schemes.

For the Case (I) formulation, the information measure $C_{n}^{fb}\left(\kappa \right)$, i.e., (Equation 2), corresponds to the supremum over channel input distributions $P_{X_{t}|X^{t-1},Y^{t-1}},t=1,\ldots ,n$.

For the Case (II) formulation, the information measure $C_{n}^{fb,S}\left(\kappa ,s\right)$, i.e., (Equation 4), corresponds to the supremum over channel input distributions $P_{X_{t}|S^{t},Y^{t-1},s},t=1,\ldots ,n$, and Conditions (II.1) and (II.2) are necessary. Alternatively, the necessary conditions for $C_{n}^{fb}\left(\kappa \right)$ to reduce to $C_{n}^{fb,S}\left(\kappa ,s\right)$ are the conditions stated in (Equation 12) and (Equation 13) (these also follow independently of the Case (I) formulation from the converse coding theorem).(11)$$\begin{array}{cc} \; & P_{X_{t}|X^{t-1},Y^{t-1}}=P_{X_{t}|V^{t-1},Y^{t-1}}\;\text{holds by channel def.}\;Y_{k}=X_{k}+V_{k},k=1,\ldots ,n; \end{array}$$(12)$$\begin{array}{cc} \; & =P_{X_{t}|V^{t-1},Y^{t-1},s}\;ifS_{1}=s\;\text{is known to the feedback code, i.e., Cond. (II.1)}; \end{array}$$(13)$$\begin{array}{cc} \; & =P_{X_{t}|S^{t},Y^{t-1},s}\;if\left(V^{t-1},S_{1}=s\right)\text{uniquely}\;\mathrm{def}.S^{t}\;\text{and vice-versa, i.e., Cond. (II.2)}. \end{array}$$
Thus, a necessary condition for (Equation 13) to hold is $S_{1}=\left(V_{0},W_{0}\right)=\left(v_{0},w_{0}\right)=s$, which is known to the encoder. It follows from [4] (Theorem 6.1; also see Lemma 6.1 and the comments above it, Equation (71)), that Conditions (II.1) and (II.2) are assumed; hence, these results are not developed for the Cover and Pombra [5] formulation. Additional information can be found in Remark 5.

Second, the analysis of the asymptotic per unit time limits of (Equation 2)–(Equation 4), i.e.,(14)$$\begin{array}{cc} C^{fb}\left(\kappa \right)\overset{▵}{=} & \lim_{n⟶\infty }\underset{{\left\{P_{X_{t}|X^{t-1},Y^{t-1}}\right\}}_{t=1}^{n},\;\frac{1}{n}E\left\{\sum_{t=1}^{n}{\left(X_{t}\right)}^{2}\right\}\leq \kappa }{\sup}\frac{1}{n}\left\{\sum_{t=1}^{n}H\left(Y_{t}|Y^{t-1}\right)-H\left(V^{n}\right)\right\}, \end{array}$$
is a non-trivial problem, and it requires certain technical necessary and/or sufficient conditions for the limits to exist, as well as for the rates to be independent of the initial distribution $P_{V_{1}}$ or the initial data, $S_{1}=s$ (see [3,10]), even if the noise process $V^{n}$ is stationary. It is easy to verify that the analysis in the past studies [4,7,9,11,12] considered the simpler problem $C_{n}^{fb,S}\left(\kappa ,s\right)$, and that the asymptotic limit $\lim_{n⟶\infty }\frac{1}{n}C_{n}^{fb,S}\left(\kappa ,s\right)$ does not correspond to the ergodic capacity. We clarify these points in our examples.

**Main Results.** The main results of this paper are briefly stated below.

(1) In the first part of this paper, we derive new equivalent sequential characterizations of the Cover and Pombra “*n*–block or transmission” feedback capacity formula [5] (Equation (11)), $C_{n}^{fb}\left(\kappa \right)$ (this first appeared in [10]). In particular, we derive equivalent realizations to the optimal channel input process $X^{n}$ [5] (Equation (11)), which are linear functionals of a *finite-dimensional sufficient statistic and an orthogonal innovations process*. From these new realizations follows the equivalent sequential characterizations of the “*n*–block or transmission” feedback capacity formula [5] (Equation (11)), which will henceforth be called the “*n*–finite transmission feedback information (*n*–FTFI) capacity”. The new *n*–FTFI capacity is expressed as a functional of *two generalized matrix difference Riccati equations (DREs) of the filtering theory of Gaussian systems*, instead of the one DRE given in [4,7,8]. In fact, we also show that the *n*–FTFI capacity of [4,7,8] corresponds to $C_{n}^{fb,S}\left(\kappa ,s\right)$.

(2) In the second part of this paper, we analyze the asymptotic per unit time limit of the sequential characterizations of the *n*–FTFI capacity, when the supremum and limit over n⟶∞ are interchanged in (Equation 14), denoted by $C^{fb,o}\left(\kappa \right)$. Then, we show $C^{fb,o}\left(\kappa \right)=C^{fb}\left(\kappa \right)$. We identify necessary and/or sufficient conditions for the asymptotic limit to exist, and for the optimal input process $X_{t},t=1,\ldots ,$ to be asymptotically stationary, in terms of the convergence properties of two generalized matrix difference Riccati equations (DREs) to their corresponding two generalized matrix algebraic Riccati equations (AREs). We make use of the so-called detectability, stabilizability, and unit circle controllablity conditions of generalized Kalman filters of Gaussian processes [16,17].

(3) From (1) and (2), we derive analogous results for $C_{n}^{fb}\left(\kappa ,s\right)$ and its per unit time asymptotic limit, denoted by $C^{fb,o}\left(\kappa ,s\right)$, as degenerate versions of $C_{n}^{fb}\left(\kappa \right)$ and $C^{fb,o}\left(\kappa \right)$. Further, we show that for certain noise models, and under certain conditions, it holds that $C^{fb,o}\left(\kappa ,s\right)=C^{fb}\left(\kappa \right)$, i.e., these values do not depend on the initial state or initial distributions.

(4) From (1) and (2), we derive analogous results for the Case (II) formulation, i.e., $C_{n}^{fb,S}\left(\kappa ,s\right)$ and its per unit time asymptotic limit denoted by $C^{fb,S}\left(\kappa ,s\right)$, and we show these are fundamentally different from the Case (I) formulation, $C_{n}^{fb}\left(\kappa \right)$ and $C^{fb}\left(\kappa \right)$, as well as $C_{n}^{fb}\left(\kappa ,s\right)$ and $C^{fb}\left(\kappa ,s\right)$. In particular, we show that the characterizations of *n*–FTFI capacity, $C_{n}^{fb,S}\left(\kappa ,s\right)$, for the Case (II) formulation follow directly from the Case (I) formulation, as a special case (an independent derivation is also presented). Moreover, $C_{n}^{fb,S}\left(\kappa ,s\right)$ is a functional of one generalized DRE, while $C_{n}^{fb}\left(\kappa \right),C_{n}^{fb}\left(\kappa ,s\right)$ are functionals of two generalized DREs.

### 1.2. The Code Definitions and Noise Models

**Case (I) Feedback Code and Noise Definitions.** For the Case (I) formulation, we consider the code of Definition 1 (due to [5]).

**Definition 1.** 
*Time-varying feedback code [5]*

*A noiseless time-varying feedback code for the AGN Channel (Equation 1) is denoted by $\left(2^{nR},n\right)$, n=1,2,… and consists of the following elements and assumptions:*

*(i) The uniformly distributed messages $W:\Omega \to M_{n}\overset{▵}{=}\left\{1,2,\ldots ,2^{nR}\right\}$.*

*(ii) The time-varying encoder strategies, often called codewords of block length n, defined by (the superscript e(·) on $E^{e}$ indicates that the distribution depends on the strategy $e\left(\cdot \right)\in E_{\left[1,n\right]}\left(\kappa \right)$).*

(15)
$$\begin{array}{cc} E_{\left[1,n\right]}\left(\kappa \right)≜ & \{X_{1}=e_{1}\left(W\right),X_{2}=e_{2}\left(W,X_{1},Y_{1}\right)\ldots ,X_{n}=e_{n}\left(W,X^{n-1},Y^{n-1}\right):\\ \; & \;\frac{1}{n}E^{e}\left(\sum_{t=1}^{n}{\left(X_{t}\right)}^{2}\right)\leq \kappa \}. \end{array}$$

*(iii) The average error probability of the decoder functions $y^{n}⟼d_{n}\left(y^{n}\right)\in M_{n}$, defined by*

(16)
$$\begin{array}{c} P_{error}^{\left(n\right)}=P\left\{d_{n}\left(Y^{n}\right)\neq W\right\}=\frac{1}{2^{nR}}\sum_{w=1}^{2^{nR}}P\left\{d_{n}\left(Y^{n}\right)\neq w|W=w\right\}. \end{array}$$

*(iv) The channel input sequence “$X^{n}\overset{▵}{=}\left\{X_{1},\ldots ,X_{n}\right\}$ is causally related (a notion found in [5], page 39, above Lemma 5) to $V^{n}$”, which is equivalent to the following decomposition of the joint probability distribution of $\left(X^{n},V^{n}\right)$:*

(17)
$$\begin{array}{cc} P_{X^{n},V^{n}}= & P_{V_{n}|V^{n-1},X^{n}}\;P_{X_{n}|X^{n-1},V^{n-1}}\;\ldots \;P_{V_{2}|V_{1},X^{2}}P_{X_{2}|X_{1},V_{1}}P_{V_{1}|X_{1}}P_{X_{1}} \end{array}$$


(18)
$$\begin{array}{cc} = & P_{V^{n}}\prod_{t=1}^{n}P_{X_{t}|X^{t-1},V^{t-1}},\;that\;is,\;P_{V_{t}|V^{t-1},X^{t}}=P_{V_{t}|V^{t-1}}. \end{array}$$

*That is, $X^{t}↔V^{t-1}↔V_{t}$ is a Markov chain, for t=1,…,n. As usual, the messages W are independent of the channel noise $V^{n}$.*

*A rate R is called an achievable rate with feedback coding, if there exists a sequence of codes $(2^{nR},n),n=1,2,\ldots$, such that $P_{error}^{\left(n\right)}⟶0$ as n⟶∞. The feedback capacity $C^{fb}\left(\kappa \right)$ is defined as the supremum of all achievable rates R.*


*We note that, in general, $C^{fb}\left(\kappa \right)$ depends on the initial distribution $P_{V_{1}}$; the ergodic capacity requires that $C^{fb}\left(\kappa \right)$ is independent of $P_{V_{1}}$ (see [18]).*


We consider a noise model which is consistent with [5], i.e., $V^{n}$ is jointly Gaussian distributed, $P_{V^{n}}=\times_{t=1}^{n}P_{V_{t}|V^{t-1}}$, and induced by the partially observable state space (PO-SS) realization of Definition 2.

**Definition 2.** 
*A time-varying PO-SS realization of Gaussian noise $V^{n}\in N\left(0,K_{V^{n}}\right)$ is defined by*



(19)
$$\begin{array}{cc} \; & S_{t+1}=A_{t}S_{t}+B_{t}W_{t},\;t=1,\ldots ,n-1 \end{array}$$


(20)
$$\begin{array}{cc} \; & V_{t}=C_{t}S_{t}+N_{t}W_{t},\;t=1,\ldots ,n, \end{array}$$


(21)
$$\begin{array}{cc} \; & S_{1}\in N\left(\mu_{S_{1}},K_{S_{1}}\right),\;K_{S_{1}}⪰0, \end{array}$$


(22)
$$\begin{array}{cc} \; & W_{t}\in N\left(0,K_{W_{t}}\right),\;K_{W_{t}}⪰0,\;t=1\ldots ,n\;indep.Gaussian\;process,\;W^{t}\;indep.of\;S_{1}, \end{array}$$


(23)
$$\begin{array}{cc} \; & S_{t}:\Omega \to R^{n_{s}},\;W_{t}:\Omega \to R^{n_{w}},\;V_{t}:\Omega \to R^{n_{v}},\;R_{t}\overset{▵}{=}N_{t}K_{W_{t}}N_{t}^{T}≻0,\;t=1,\ldots ,n \end{array}$$

*where $n_{v}=1$, $n_{s},n_{w}$ are arbitrary positive integers, $\left(A_{t},B_{t},C_{t},N_{t},\mu_{S_{1}},K_{S_{1}},K_{W_{t}}\right)$ are nonrandom, measurable in t, with bounded entries, and $n_{s},n_{w}$ are finite positive integers.*

*The above components $B_{t}W_{t}$ and $N_{t}W_{t}$ can be taken as*

(24)
$$\begin{array}{cc} \; & B_{t}W_{t}=B_{t}^{1}W_{t}^{1}+B_{t}^{2}W_{t}^{2},\;N_{t}W_{t}=N_{t}^{1}W_{t}^{1}+N_{t}^{2}W_{t}^{2},\;t=1,\ldots ,n, \end{array}$$


(25)
$$\begin{array}{cc} \; & W^{1,n}\;and\;W^{1,n}\;areindependent, \end{array}$$

*where $B_{t}^{1}$ or $B_{t}^{2}$ can be zero and $N_{t}^{1}$ or $N_{t}^{2}$ can be zero ∀t.*

*A time-invariant PO-SS realization of the Gaussian noise $V^{n}\in N\left(0,K_{V^{n}}\right)$ is defined by (Equation 19)–(Equation 23), with $\left(A_{t},B_{t},C_{t},N_{t},K_{W_{t}}\right)=\left(A,B,C,N,K_{W}\right),\forall t$.*



For the Case (I) formulation, we use the terminology “partially observable”, which is standard in filtering theory [16], because the noise $V^{n}$ induces a distribution $P_{V^{n}}=\times_{t=1}^{n}P_{V_{t}|V^{t-1}}$, and $P_{V_{t}|V^{t-1}}$ cannot be expressed as a function of the state of the noise, i.e., $V^{t-1}$ does not uniquely define $S^{t}$. However, if $S_{1}=s$ is known to the encoder, then it can be as easily verified from the ARMA (a,c), that $V^{t-1}$ uniquely defines $S^{t}$, recursively. However, for the PO-SS realization, with $B_{t}^{2}=0,N_{t}^{1}=0,\forall t$, even if the initial state $S_{1}=s$ is known to the encoder, $V^{t-1}$ does not uniquely define $S^{t}$ because there are two independent noises $W^{1,n-1}$ and $W^{2,n}$ that enter the equations of $S^{n}$ and $V^{n}$. The PO-SS realization is often adopted in many practical problems of engineering and science to realize jointly Gaussian processes $V^{n}$.

We should emphasize that for the Case (I) formulation to be consistent with the Cover and Pombra [5] formulation, both the code of Definition 1 and the PO-SS realization of Definition 2 must respect the following two conditions:

**(A1)** 
*The initial state $S_{1}$ of the noise is not known at the encoder nor the decoder;*

**(A2)** 
*At each t, the representation of the noise $V^{t-1}$ by the PO-SS realization of Definition 2 does not uniquely determine the state of the noise $S^{t}$ and vice-versa, i.e., it is a partially observable realization.*

**Case (II) Formulation of Feedback Code and Noise Definitions.** For the Case (II) formulation, we presuppose the following:

**Condition 1.** 
*The initial state of the noise or channel $S_{1}=s$ is known to the encoder and decoder;*

**Condition 2.** 
*Given a fixed initial state $S_{1}=s$, known to the encoder and the decoder, at each t, the channel noise $V^{t-1}$ uniquely defines the state of the noise $S^{t}$ and vice-versa.*

Thus, for the Case (II) formulation, the code is that of Definition 3, below (hence different from Definition 1).

**Definition 3.** 
*A code with initial state known at the encoder and the decoder*

*A variant of the code of Definition 1 is feedback code with the initial state of the noise or channel $S_{1}=s$, known to the encoder and decoder strategies, denoted by $\left(s,2^{nR},n\right)$, n=1,2,….*

*The code $\left(s,2^{nR},n\right)$, n=1,2,… is defined as in Definition 2, with (ii), (iii), and (iv) being replaced by*

$$\begin{array}{cc} E_{\left[1,n\right]}^{s}\left(\kappa \right)≜ & \{X_{1}=e_{1}\left(W,S_{1}\right),X_{2}=e_{2}\left(W,S_{1},X_{1},Y_{1}\right)\ldots ,X_{n}=e_{n}\left(W,S_{1},X^{n-1},Y^{n-1}\right): \end{array}$$


(26)
$$\begin{array}{cc} \; & \frac{1}{n}E^{e}\left\{\sum_{i=1}^{n}{\left(X_{t}\right)}^{2}|S_{1}=s\right\}\leq \kappa \},\;\left(y^{n},s\right)⟼d_{n}^{s}\left(y^{n},s\right)\in M_{n}, \end{array}$$


(27)
$$\begin{array}{cc} P_{X^{n},V^{n}|S_{1}}= & P_{V^{n}|S_{1}}\prod_{t=1}^{n}P_{X_{t}|X^{t-1},V^{t-1},S_{1}},\;that\;is,\;P_{V_{t}|V^{t-1},X^{t},S_{1}}=P_{V_{t}|V^{t-1},S_{1}}. \end{array}$$

*The feedback capacity is denoted by $C^{fb}\left(\kappa ,s\right)$ and should be distinguished from $C^{fb}\left(\kappa \right)$.*

*The feedback capacity is denoted by and $C^{fb,S}\left(\kappa ,s\right)$ if, in addition, Condition 2 holds.*

*The initial state may include $S_{1}\overset{▵}{=}\left(V_{-\infty }^{0},Y_{-\infty }^{0}\right)$, etc.*



It will become apparent that past studies [4,7,8] considered feedback capacity, $C^{fb,S}\left(\kappa ,s\right)$, and not $C^{fb}\left(\kappa ,s\right)$ or $C^{fb}\left(\kappa \right)$.

For the Case (II) formulation, it is obvious (from the converse to the coding theorem) that the optimal channel input conditional distribution is expressed as a function of the state of the noise, $S^{n}$, due to (Equation 12), (Equation 13).

### 1.3. Approach of This Paper

Our approach and analysis of information measures (Equation 2)–(Equation 4), as well sa their per unit time limits, is based on the following two step procedure:

**Step # 1.** We apply a linear transformation to the Cover and Pombra optimal channel input process [5] (Equation (11)), (see (Equation 33)–(Equation 39) to equivalently represent it by a linear functional of the past channel noise sequence, the past channel output sequence, and an orthogonal Gaussian process, i.e., an innovations process. That is, $X^{n}$ is uniquely represented, since it is expressed in terms of the orthogonal process.

**Step # 2.** We express the optimal input process by a functional of a *sufficient statistic*, which satisfies a Markov recursion, and an *orthogonal innovations process*. It then follows that the Cover and Pombra characterization of the “*n*–block” formula [5] (Equation (10)) (see (Equation 33) and (Equation 34)) is equivalently represented by a sequential characterization. The problem of feedback capacity is then expressed as the maximization over two sequences of time-varying strategies of the channel input process of the difference of (differential) entropies of the innovations processes of $Y^{n}$ and $V^{n}$ (analog of entropies in the right-hand side of (Equation 2)).(28)$$\begin{array}{cc} H\left(Y^{n}\right)-H\left(V^{n}\right)= & \sum_{t=1}^{n}\left\{H\left(Y_{t}|Y^{t-1}\right)-H\left(V_{t}|V^{t-1}\right)\right\} \end{array}$$(29)$$\begin{array}{cc} = & \sum_{t=1}^{n}\left\{H\left(Y_{t}-E\left\{Y_{t}|Y^{t-1}\right\}|Y^{t-1}\right)-H\left(V_{t}-E\left\{V_{t}|V^{t-1}\right\}|V^{t-1}\right)\right\} \end{array}$$(30)$$\begin{array}{cc} \overset{\left(a\right)}{=} & \sum_{t=1}^{n}\left\{H\left(Y_{t}-E\left\{Y_{t}|Y^{t-1}\right\}\right)-H\left(V_{t}-E\left\{V_{t}|V^{t-1}\right\}\right)\right\} \end{array}$$(31)$$\begin{array}{cc} = & \sum_{t=1}^{n}\left\{H\left(I_{t}\right)-H\left({\hat{I}}_{t}\right)\right\},\;I_{t}\overset{▵}{=}Y_{t}-E\left\{Y_{t}|Y^{t-1}\right\},\;{\hat{I}}_{t}\overset{▵}{=}V_{t}-E\left\{V_{t}|V^{t-1}\right\} \end{array}$$
where (a) is due to the Gaussianity of $Y^{n}$ and $V^{n}$, as well as due the independence of the innovations processes $I_{t}$ and $Y^{t-1}$ and of innovations processes ${\hat{I}}_{t}$ and $V^{t-1}$.

The asymptotic analysis of $\lim_{n⟶\infty }\frac{1}{n}C_{n}^{fb}\left(\kappa \right)$ (or with limit and supremum interchanged) is then addressed from the asymptotic properties of entropy rates and the average power,(32)$$\begin{array}{c} \lim_{n⟶\infty }\frac{1}{n}\sum_{t=1}^{n}\left\{H\left(I_{t}\right)-H\left({\hat{I}}_{t}\right)\right\},\;\lim_{n⟶\infty }\frac{1}{n}E\left\{\sum_{t=1}^{n}{\left(X_{t}\right)}^{2}\right\} \end{array}$$
over the channel input distributions, and where the covariance of the innovations process of $Y^{n}$ is a functional of the solutions of two generalized matrix DREs. We identify necessary and/or sufficient conditions for the existence of limits, irrespective of whether the noise $V^{n}$ is non-stationary, unstable, or stationary. Further, we show that, in general, the characterizations of feedback capacity for the Case (I) and Case (II) formulations are fundamentally different, and we identify conditions based on the feedback codes and noise to coincide.

### 1.4. Review of Related Literature

Asymptotic feedback capacity formulas and bounds for AGN channels, driven by stationary and asymptotically stationary (often limited memory) noise, are derived since the early 1970s in an anthology of papers based on information theoretic formulas, under various assumptions [7,8,9,11,12,13,14,19,20,21,22,23,24,25,26,27,28,29,30,31,32,33], in two directions.

**(D1)** Characterizations and explicit formulas of asymptotic feedback capacity that correspond to feedback codes, when the initial state of the noise (the initial state $S_{1}$ is any a priori information) $S_{1}=s_{1}$ is known or not known to the encoder and the decoder;**(D2)** Bounds on asymptotic feedback capacity that correspond to linear feedback coding schemes of communicating Gaussian random variables (RVs), Θ:Ω→R, and coding schemes of communicating digital messages $W:\Omega \to \{1,2,\ldots ,2^{nR}\}$, when the initial state the $S_{1}=s_{1}$ is known or not known to the encoder and the decoder.

#### 1.4.1. The Cover and Pombra Characterizations of Capacity Pombra [5]

Cover and Pombra characterized the *n*–FTFI capacity for non-stationary and non-ergodic noise $V^{n}$, [5] (Equation (10)), by (we use H(X) to denote differential entropy of a continuous-valued RV *X*; hence, we indirectly assume the probability density functions exist)(33)$$\begin{array}{cc} C_{n}^{fb}\left(\kappa \right)\overset{▵}{=} & \max_{\left(B^{n},K_{{\bar{Z}}^{n}}\right):\;\frac{1}{n}tr\left\{E\left(X^{n}{\left(X^{n}\right)}^{T}\right)\right\}\leq \kappa }H\left(Y^{n}\right)-H\left(V^{n}\right) \end{array}$$(34)$$\begin{array}{cc} = & \max_{\left(B^{n},K_{{\bar{Z}}^{n}}\right):\;\frac{1}{n}tr\left\{B^{n}\;K_{V^{n}}\;{\left(B^{n}\right)}^{T}+K_{{\bar{Z}}^{n}}\right\}\leq \kappa }\frac{1}{2}\log\frac{|\left(B^{n}+I_{n\times n}\right)K_{V^{n}}{\left(B^{n}+I_{n\times n}\right)}^{T}+K_{{\bar{Z}}^{n}}|}{|K_{V^{n}}|} \end{array}$$
where the distribution $P_{Y^{n}}$ is induced by a jointly Gaussian channel input process $X^{n}$ [5] (Equation (11)): (35)$$\begin{array}{cc} \; & X_{t}=\sum_{j=1}^{t-1}B_{t,j}V_{j}+{\bar{Z}}_{t},\;t=1,\ldots ,n, \end{array}$$(36)$$\begin{array}{cc} \; & X^{n}=B^{n}V^{n}+{\bar{Z}}^{n},\;Y^{n}=\left(B^{n}+I_{n\times n}\right)V^{n}+{\bar{Z}}^{n}, \end{array}$$(37)$$\begin{array}{cc} \; & {\bar{Z}}^{n}\;\text{is jointly Gaussian},\;N(0,K_{{\bar{Z}}^{n}}),\;{\bar{Z}}^{n}\;\text{is independent of}\;V^{n}, \end{array}$$(38)$$\begin{array}{cc} \; & X^{n}\overset{▵}{=}{\left[\begin{array}{cccc} X_{1} & X_{2} & \ldots & X_{n} \end{array}\right]}^{T}\;\text{similarly for the rest},\;B^{n}\;\text{lower diagonal matrix}, \end{array}$$(39)$$\begin{array}{cc} \; & \frac{1}{n}E\left\{\sum_{t=1}^{n}{\left(X_{t}\right)}^{2}\right\}=\frac{1}{n}tr\;\left\{E\left(X^{n}{\left(X^{n}\right)}^{T}\right)\right\}\leq \kappa . \end{array}$$
The notation $N(0,K_{{\bar{Z}}^{n}})$ means that the random variable ${\bar{Z}}^{n}$ is jointly Gaussian with mean $E\{{\bar{Z}}^{n}\}=0$ and covariance matrix $K_{{\bar{Z}}^{n}}=E\left\{{\bar{Z}}^{n}{\left({\bar{Z}}^{n}\right)}^{T}\right\}$, and $I_{n\times n}$ denotes an *n* by the *n* identity matrix. Feedback capacity, $C^{fb}\left(\kappa \right)$, is characterized by the per unit time limit of the *n*–FTFI capacity [5] (Theorem 1).(40)$$\begin{array}{c} C^{fb}\left(\kappa \right)\overset{▵}{=}\lim_{n⟶\infty }\frac{1}{n}C_{n}^{fb}\left(\kappa \right). \end{array}$$

Over the years, considerable efforts have been devoted to compute $C_{n}^{fb}\left(\kappa \right)$ and $C^{fb}\left(\kappa \right)$ [4,7,8,9,11,33], often under simplified assumptions on the channel noise. In addition, bounds are described in [27,28,29], while numerical methods are developed in [31] for time-invariant AGN channels, driven by stationary noise. In [4,7,11,33,34], the authors considered a variant of (Equation 40) by interchanging the per unit time limit and maximization operations under the following assumption: the joint process $(X^{n},Y^{n}),n=1,2,\ldots$ is either *jointly stationary or asymptotically stationary*, and *the joint distribution of the joint process $(X^{n},Y^{n}),n=1,2,\ldots$ is time-invariant*. We describe [4,7,8] below.

#### 1.4.2. The Yang, Kavcic and Tatikonda [8] Characterization of Maximal Information Rate

In [8], the authors analyzed the feedback capacity of the AGN channel (Equation 1), driven by a stationary noise, described the power spectral density (PSD) functions $S_{V}\left(e^{j\theta }\right),\theta \in \left[-\pi ,\pi \right]$:(41)$$\begin{array}{c} S_{V}\left(e^{j\theta }\right)\overset{▵}{=}K_{W}\frac{\left(1-\sum_{k=1}^{L}a\left(k\right)e^{jk\theta }\right)\left(1-\sum_{k=1}^{L}a\left(k\right)e^{-jk\theta }\right)}{\left(1-\sum_{k=1}^{L}c\left(k\right)e^{jk\theta }\right)\left(1-\sum_{k=1}^{L}c\left(k\right)e^{-jk\theta }\right)},\;|c\left(k\right)|<1,\;|a\left(k\right)|<1,\;c\left(k\right)\neq a\left(k\right). \end{array}$$
The analysis in [8] is based on time-domain methods and corresponds to the Case (II) formulation (see [8] (Section II; in particular, Section II.C, I–III, Theorem 1, Section III) by considering a specific time-invariant, stable, state space realization of the noise PSD (Equation 41), such that Conditions (II.1), and (II.2) hold, i.e.,


*The initial state of the noise, $S_{1}=s$, is known to the encoder and the decoder, and the initial state and noise $\left(s,V^{t-1}\right)$ uniquely define the noise state $S^{t}$, and vice-versa, for all t.*


The time-domain characterization of feedback capacity, called the maximal information rate [8] (Theorem 6), corresponds to the supremum and limit being interchanged, and involves only one matrix Riccati equation of the linear filtering theory. However, ref. [8] (Theorem 6) does not state the conditions based on which the maximal information rate is valid (i.e., existence of asymptotic limit).

#### 1.4.3. The Kim [4] Characterization of Feedback Capacity

The author in [4] also analyzed the feedback capacity of the AGN channel (Equation 1), driven by stationary noise described by the PSD (Equation 41) and by a time-invariant, stable, state space realization of the noise $V^{n}$ (see [4] (Section VI)). A major point of confusion is that the characterization of feedback capacity in the time domain [4] (Theorem 6.1) does not state whether this corresponds to Case (I) or Case (II) formulations. The reader, however, can verify from [4] (Lemma 6.1 and comments above it) that the time-domain characterization of feedback capacity [4] (Theorem 6.1) corresponds to the Case (II) formulation, as stated in the study by Yang, Kavcic and Tatikonda [8]. In fact, the characterization of feedback capacity [4] (Theorem 6.1) involves only one Riccati equation of the linear filtering theory, as in [8]. We reconfirm this point at various parts of this paper (see, for example, Section 2.6).

#### 1.4.4. The Gattami [7] Characterization of Feedback Capacity and Semi-Definite Progamming Formulation

The authors in [7] re-visited the feedback capacity of the AGN channel (Equation 1) driven by a stationary noise described by the PSD (Equation 41) and with a time-invariant, stable, state space realization of the noise $V^{n}$. One of the main results of [7] is the feedback capacity characterization for the Case (II) formulation, i.e., that involves only one matrix Riccati equation of the filtering theory, precisely as in [4,8]. Another main result of [7] is the re-formulation of the optimization problem using semi-definite programming.

In the following remark, we will provide additional insights into the results of [4,7,8].

**Remark 1.** 
*On the formulas in [4,7,8].*

*Refs. [4,7] considered the stable, time-invariant PO-SS realization of Definition 2, with $W_{t}:\Omega \to R,N_{t}=1,\forall t$, i.e., $n_{w}=1$.*


*In [4], (Theorem 6.1) (see [4] (above and below Equation (70))), the asymptotic characterization of feedback capacity involves one filtering matrix ARE and is achieved by a channel input, which is different at times $n=1,2,\ldots ,n_{s}$ from subsequent time $n=n_{s}+1,\ldots$, given by*

(42)
$$\begin{array}{cc} \; & X_{n}=Z_{n}\in N\left(0,K_{Z}\right),K_{Z}>0,\;n=1,\ldots ,n_{s}, \end{array}$$


(43)
$$\begin{array}{cc} \; & X_{n}=\chi \left(S_{n}-E\left\{S_{n}|Y^{n-1}\right\}\right),\;n=n_{s}+1,n_{s}+2,\ldots . \end{array}$$

*where χ is a constant vector, and $Z_{n},n=1,2,\ldots ,n_{s}$ are IID Gaussian RVs.*

*In [7] (Theorem 3), the asymptotic characterization of feedback capacity involves one filtering matrix ARE and is incurred by the time-invariant channel input*

(44)
$$\begin{array}{c} X_{n}=\chi \left(S_{n}-E\left\{S_{n}|Y^{n-1}\right\}\right)+Z_{n},\;Z_{n}\in N\left(0,K_{Z}\right),\;K_{Z}\geq 0,\;n=1,2,\ldots \end{array}$$

*$Z_{n},n=1,2,\ldots$, are IID Gaussian RVs.*

*In [8], a state space realization of the PSD is considered, and the maximal information rate is presented, (Equation (125)) and [8] (Theorem 6, Equation (138), $I_{max}$), which involves one filtering matrix ARE. It is achieved by*

(45)
$$\begin{array}{cc} \; & X_{n}=\chi \left(S_{n-1}-E\left\{S_{n-1}|Y^{n-1},S_{0}=s_{0}\right\}\right)+Z_{n},\;X_{1}=Z_{1}, \end{array}$$


(46)
$$\begin{array}{cc} \; & Z_{n}\in N\left(0,K_{Z}\right),\;K_{Z}\geq 0,\;n=1,2,\ldots , \end{array}$$

*where $Z_{n},n=1,2,\ldots$ are IID Gaussian RVs.*

*The above references computed the feedback capacity and maximal information rate for the ARMA (a,c),a∈[−1,1],c∈(−1,1),c≠a and arrived at the conclusion that it is precisely Butman’s [13] and Wolfowitz’s [14] lower bound.*

*It will become apparent that [4,7,8] arrived at the above expressions by considering problem $C^{fb,S}\left(\kappa ,s\right)$, and not $C^{fb}\left(\kappa \right)$ or $C^{fb}\left(\kappa ,s\right)$.*


Sequential equivalent characterizations of the Cover and Pombra [5] *n*–FTFI capacity, $C_{n}^{fb}\left(\kappa \right)$ and $C^{fb}\left(\kappa \right)$ capacity of the AGN channel (Equation 1) driven by a non-stationary and non-ergodic Gaussian noise $V^{n}$, as well as by time-varying, unstable (and stable) state space realizations of Definition 2, are derived in [1,2,3,10], including relations between Case (I) and Case (II) formulations. In particular, ref. [10] proved that $C_{n}^{fb}\left(\kappa \right)$ of the AGN channel with state space noise of Definition 2 involves two matrix Riccati equations of the linear filtering theory and that, under certain conditions, $C_{n}^{fb}\left(\kappa \right)$ reduces to $C_{n}^{fb,S}\left(\kappa ,s\right)$, which involves only one matrix Riccati equation. Corresponding expressions are obtained for their per unit time asymptotic limits. The methods and results of [10] are generalized to multiple-input multiple-output (MIMO) Gaussian channels in [3]. Further, ref. [2] derived closed-form expressions of $C^{fb}\left(\kappa \right)$ for AGN channels driven by the stable and unstable ARMA (a,c),a∈(−∞,∞),c∈(−∞,∞),c≠a noise, showing the connection between Case (I) and Case (II) formulations. Ref. [3] generalized the earlier investigation in [1], which considered the autoregressive unit memory stable and unstable noise. An investigation of nonfeedback capacity of stable and unstable noise is outlined in [35]. The connection of ergodic theory and feedback capacity of unstable channels is discussed in [36,37].

MIMO Gaussian channels are also investigated in [38]. However, many of the expressions in [38] are previously obtained in [3,10]. The analysis in [38] does not include a derivation of the optimal channel input that achieves $C_{n}^{fb}\left(\kappa \right)$ and $C^{fb}\left(\kappa \right)$, and does not discuss the connection between Case (I) and Case (II) formulations. The closed-form expressions of capacity of the examples in [38], are special cases of expressions in [1] and in [2], which treated the noise ARMA (a,c),∀a∈(−∞,∞),∀c∈(−∞,∞),c≠a.

We structure this paper as follows.

In Section 2, we derive the new sequential characterizations of the *n*–FTFI capacity for the Cover and Pombra formulation of feedback capacity of the AGN channel (Equation 1), i.e., for the Case (I) formulation, $C_{n}^{fb}\left(\kappa \right)$. We also derive analogous sequential characterizations for $C_{n}^{fb}\left(\kappa ,s\right)$ and for the Case (II) formulation, $C_{n}^{fb,S}\left(\kappa ,s\right)$, i.e., when Conditions 1 and 2 hold, to illustrate their fundamental differences.

In Section 3, we present the asymptotic analysis of feedback capacity for the Case (I) formulation. In Section 4, we treat the Case (II) formulation.

This paper contains several examples and makes comparisons to the existing literature.

## 2. Sequential Characterizations of n–FTFI Capacity for Case (I) Formulation

In this section, we derive equivalent sequential characterizations for the following:

(i) $C_{n}^{fb}\left(\kappa \right)$ defined by (Equation 2) of the Case (I) formulation, i.e., for the Cover and Pombra *n*–FTFI capacity characterization (Equation 34);

(ii) $C_{n}^{fb}\left(\kappa ,s\right)$ defined by (Equation 3), as a degenerated case of $C_{n}^{fb}\left(\kappa \right)$;

(iii) $C_{n}^{fb,S}\left(\kappa ,s\right)$ defined by (Equation 4) of the Case (II) formulation, as a degenerated case of $C_{n}^{fb}\left(\kappa \right)$.

We organize the presentation of the material as follows.

(1) Section 2.1. Here, we introduce our notation.

(2) Section 2.2. The main result is Theorem 1, which provides an equivalent sequential characterization of the Cover and Pombra characterization of the *n*–FTFI capacity, $C_{n}^{fb}\left(\kappa \right)$, i.e., of (Equation 33) and (Equation 34). Our derivation proceeds as follows. We apply a linear transformation to the Cover and Pombra Gaussian optimal channel input $X^{n}$ (Equation 35), to represent $X_{t}$, with a linear function of $\left(V^{t-1},Y^{t-1}\right)$ or equivalently $\left(X^{t-1},Y^{t-1}\right)$ and an orthogonal Gaussian innovations process $Z_{t}$, which is independent of $\left(Z^{t-1},X^{t-1},V^{t-1},Y^{t-1}\right)$ for t=1,…,n.

Subsequently, we apply Theorem 1 to the time-varying PO-SS$\left(a_{t},c_{t},b_{t}^{1},b_{t}^{2},d_{t}^{1},d_{t}^{2}\right)$ noise (see Example 1) to the non-stationary autoregressive moving average, ARMA (a,c),a∈(−∞,∞),c∈(−∞,∞) noise, and to the stationary ARMA (a,c),a∈(−1,1),c∈(−1,1) noise (see Example 2), which is found in many references, such as [4]. It will become apparent that our characterizations of *n*–FTFI capacity are fundamentally different from past studies.

(3) Section 2.3. The main result is Theorem 3, which provides a simplified characterization of the sequential characterization of the *n*–FTFI capacity, $C_{n}^{fb}\left(\kappa \right)$, given in Theorem 1 (i.e., the equivalent of (Equation 34)), for the time-varying AGN channel (Equation 1) driven by the PO-SS realization of Definition 2, for the code of Definition 1. The *n*–FTFI capacity of Theorem 3 is expressed in terms of solutions to two DREs. Our derivation is based on identifying a *finite-dimensional sufficient statistic* to express $X_{t}$ as a functional of the sufficient statistic, instead of $\left(V^{t-1},Y^{t-1}\right)$ or $\left(X^{t-1},Y^{t-1}\right)$, and an orthogonal Gaussian innovations process.

(4) Section 2.4. The main results is Corollary 6, which is an application of Theorem 3 (i.e., the sufficient statistic representation), to the ARMA (a,c),a∈(−∞,∞),c∈(−∞,∞) noise of Example 2. This example shows that the *n*–FTFI capacity is expressed in terms of solutions to two DREs.

From Corollary 6, the following will become apparent:

(i) Neither the time-domain characterization [4] (Theorem 6.1) (see [4]) (Theorem 5.3) nor the frequency domain characterization [4] (Theorem 4.1) correspond to the Cover and Pombra characterization of feedback capacity.

(5) Section 2.5. The main results is Corollary 7, which gives the *n*–FTFI capacity for the Case (II) formulation, as a degenerate case of the Case (I) formulation, i.e., of Theorem 3.

(6) Section 2.6. The main result is Proposition 2, which further clarifies the following.

(i) The formulation of [8] and the formulation that led to [4] (Theorem 6.1) are based on the Case (II) formulation, as well as (ii) some of the oversights in [4,7,9,11,12].

### 2.1. Notation

Throughout this paper, we use the following notation:
$Z\overset{▵}{=}\left\{\ldots ,-1,0,1,\ldots \right\},Z_{+}\overset{▵}{=}\left\{1,\ldots \right\},Z_{+}^{n}\overset{▵}{=}\left\{1,2,\ldots ,n\right\}$, where *n* is a finite positive integer.$R\overset{▵}{=}\left(-\infty ,\infty \right)$, and $R^{m}$ is the vector space of tuples of the real numbers for an integer $n\in Z_{+}$.$C\overset{▵}{=}\left\{a+jb:\left(a,b\right)\in R\times R\right\}$ is the space of complex numbers.$R^{n\times m}$ is the set of *n* by *m* matrices with entries from the set of real numbers for integers $\left(n,m\right)\in Z_{+}\times Z_{+}$.$D_{o}\overset{▵}{=}\left\{c\in C:|c|<1\right\}$ is the open unit disc of the space of complex number C.$S_{+}^{n\times n},n\in Z_{+}$ (resp. $S_{++}^{n\times n}$) denotes the set of positive semidefinite (resp. positive definite) symmetric matrices with elements in the real numbers and of size n×n. Thus, $A\in S_{+}^{n\times n}$ if for all $w\in R^{n}$, $w^{T}Aw\geq 0$. Positive semidefiniteness is denoted by A⪰0 and (strict) positive definiteness by A≻0. $I_{n\times n}\in S_{++}^{n\times n},n\in Z_{+}$ denotes the identity matrix, $tr\left(A\right)$ denotes the trace of any matrix $A\in R^{n\times n},n\in Z_{+}$.spec(A)⊂C is the Spectrum of a matrix $A\in R^{q\times q},q\in Z_{+}$ (the set of all its eigenvalues). A matrix $A\in R^{q\times q}$ is called exponentially stable if all its eigenvalues are within the open unit disc, i.e., $spec\left(A\right)\subset D_{o}$.$\left(\Omega ,F,P\right)$ denotes a probability space. Given a random variable $X:\Omega \to R^{n_{x}},n_{x}\in Z_{+}^{n}$, its induced distribution on $R^{n_{x}}$ is denoted by $P_{X}$.$P_{X}\in N\left(\mu_{X},K_{X}\right),K_{X}⪰0$ denotes a Gaussian distributed RV *X*, with mean value $\mu_{X}$ and covariance matrix $K_{X}=cov\left(X,X\right)⪰0$, defined by(47)$$\begin{array}{c} \mu_{X}\overset{▵}{=}E\left\{X\right\},\;K_{X}=cov\left(X,X\right)\overset{▵}{=}E\left\{\left(X-E\left\{X\right\}\right){\left(X-E\left\{X\right\}\right)}^{T}\right\}. \end{array}$$Given another Gaussian random variable $Y:\Omega \to R^{n_{y}},n_{y}\in Z_{+}^{n}$, which is jointly Gaussian distributed with *X*, i.e., the joint distribution is $P_{X,Y}$, the conditional covariance of *X* given *Y* is defined by(48)$$\begin{array}{cc} K_{X|Y}=cov\left(X,X|Y\right)\overset{▵}{=} & E\left\{\left(X-E\left\{X|Y\right\}\right){\left(X-E\left\{X|Y\right\}\right)}^{T}|Y\right\} \end{array}$$(49)$$\begin{array}{cc} = & E\left\{\left(X-E\left\{X|Y\right\}\right){\left(X-E\left\{X|Y\right\}\right)}^{T}\right\} \end{array}$$
where the last equality is due to a property of jointly Gaussian distributed RVs.Given three arbitrary RVs (X,Y,Z) with induced distribution $P_{X,Y,Z}$, the RVs (X,Z) are called conditionally independent given the RV *Y* if $P_{Z|X,Y}=P_{Z|Y}$. This conditional independence is often denoted by X↔Y↔Z, which is a Markov chain.

### 2.2. Preliminary Characterizations of n–FTFI Capacity of AGN Channels Driven by Correlated Noise

We start with preliminary calculations for the feedback code of Definition 1, which we use to prove Theorem 1. These calculations are introduced for the sake of clarity and to establish our notation.
For the feedback code of Definition 1, by the channel definition (Equation 1), i.e., (Equation 18), the conditional distribution of $Y_{t}$, given $Y^{t-1}=y^{t-1},X^{t}=x^{t}$, is$$\begin{array}{cc} P\left\{Y_{t}\in dy|Y^{t-1}=y^{t-1},X^{t}=x^{t}\right\}\overset{\left(a\right)}{=} & P\left\{Y_{t}\in dy|Y^{t-1}=y^{t-1},X^{t}=x^{t},V^{t-1}=v^{t-1}\right\} \end{array}$$(50)$$\begin{array}{cc} \overset{\left(b\right)}{=} & P_{V_{t}|V^{t-1}}\left(v_{t}:x_{t}+v_{t}\in dy\right),\;\;t=2,\ldots ,n \end{array}$$(51)$$\begin{array}{cc} = & P_{Y_{t}|X_{t},V^{t-1}} \end{array}$$(52)$$\begin{array}{cc} \equiv & P_{t}\left(dy|x_{t},v^{t-1}\right), \end{array}$$(53)$$\begin{array}{cc} P\left\{Y_{1}\in dy|Y^{0}=y^{0},X^{1}=x^{1}\right\}= & P_{Y_{1}|X_{1}}\equiv P_{1}\left(dy|x_{1}\right). \end{array}$$
where (a) is due to (Equation 1) and (b) is due to (Equation 18). We introduce the set of channel input distributions with feedback, which are consistent with the code of Definition 1, not necessarily generated by the messages *W*, as follows:(54)$$\begin{array}{c} P_{\left[1,n\right]}\left(\kappa \right)\overset{▵}{=}\left\{P_{t}\left(dx_{t}|x^{t-1},y^{t-1}\right)\overset{▵}{=}P_{X_{t}|X^{t-1},Y^{t-1}},t=1,\ldots ,n:\frac{1}{n}E^{P}\left(\sum_{t=1}^{n}{\left(X_{t}\right)}^{2}\right)\leq \kappa \right\}. \end{array}$$
By Definition 1, we have $E_{\left[1,n\right]}\left(\kappa \right)\subseteq P_{\left[1,n\right]}\left(\kappa \right)$. Moreover, by the channel definition, any pair of the sequence triple $\left(V^{t},X^{t},Y^{t}\right)$ uniquely defines the remaining sequence. Thus, the identity holds:(55)$$\begin{array}{cc} \; & {\bar{P}}_{\left[1,n\right]}\left(\kappa \right)\overset{▵}{=}\left\{{\bar{P}}_{t}\left(dx_{t}|v^{t-1},y^{t-1}\right),t=1,\ldots ,n:\frac{1}{n}E^{\bar{P}}\left(\sum_{t=1}^{n}{\left(X_{t}\right)}^{2}\right)\leq \kappa \right\}=P_{\left[1,n\right]}\left(\kappa \right). \end{array}$$
We also emphasize that, by Definition 1, for a given feedback encoder strategy $e\left(\cdot \right)\in E_{\left[1,n\right]}\left(\kappa \right)$, i.e., $x_{1}=e_{1}\left(w\right),x_{2}=e_{2}\left(w,x_{1},y_{1}\right),\ldots ,x_{n}=e_{n}\left(w,x^{n-1},y^{n-1}\right)$ the conditional distributions of $Y_{t}$ given $\left(Y^{t-1},W\right)=\left(y^{t-1},w\right)$ is obtained as follows:(56)$$\begin{array}{cc} P_{Y_{t}|W,Y^{t-1}}^{e}\left(dy_{t}|,y^{t-1},w\right)\overset{\left(a\right)}{=} & P_{t}\left(dy_{t}|\left\{e_{j}\left(w,x^{j-1},y^{j-1}\right):j=1,\ldots ,t\right\},y^{t-1},w\right) \end{array}$$(57)$$\begin{array}{cc} \overset{\left(b\right)}{=} & P_{t}\left(dy_{t}|\left\{e_{j}\left(w,x^{j-1},y^{j-1}\right):j=1,\ldots ,t\right\},y^{t-1},v^{t-1},w\right) \end{array}$$(58)$$\begin{array}{cc} \overset{\left(c\right)}{=} & P_{t}\left(dy_{t}|\left\{e_{j}\left(w,x^{j-1},y^{j-1}\right):j=1,\ldots ,t\right\},v^{t-1},w\right) \end{array}$$(59)$$\begin{array}{cc} \overset{\left(d\right)}{=} & P_{t}\left(dy_{t}|\left\{e_{j}\left(w,x^{j-1},y^{j-1}\right):j=1,\ldots ,t\right\},v^{t-1}\right) \end{array}$$(60)$$\begin{array}{cc} \overset{\left(e\right)}{=} & P_{t}\left(dy_{t}|e_{t}\left(w,x^{t-1},y^{t-1}\right),v^{t-1}\right) \end{array}$$(a) is due to knowledge of the distribution of the strategies $e_{j}\left(\cdot \right),j=1,\ldots ,t$, the code definition, and the recursive substitution, $x_{1}=e_{1}\left(w\right),x_{2}=e_{2}\left(w,x_{1},y_{1}\right),\ldots ,e_{t}\left(w,x^{t-1},y^{t-1}\right)$, where $x^{t-1}$ is specified by the knowledge of the strategies, $e_{j}\left(\cdot \right),j=1,\ldots ,t-1$ and the knowledge of $\left(y^{t-2},w\right)$,(b) is due to knowing $x_{j}=e_{j}\left(w,x^{j-1},y^{j-1}\right),y_{j},j=1,\ldots ,t-1$ specifies $v_{j}=y_{j}-x_{j},j=1,\ldots ,t-1$,(c) is due to the fact that, any pair of the triple $\left(x^{t},y^{t},v^{t}\right)$ specifies the remaining sequence, i.e., knowing $\left(x^{t-1},v^{t-1}\right)$ specifies $y^{t-1}$, and $y^{t-1}$ is thus redundant,(d) is due to the conditional independence $P_{V_{t}|V^{t-1},X^{t},W}=P_{V_{t}|V^{t-1},X^{t}}$,(e) is due to (Equation 18), i.e., $P_{V_{t}|V^{t-1},X^{t}}=P_{V_{t}|V^{t-1}}$, and the channel definition.

By the channel definition $Y_{t}=X_{t}+V_{t},t=1,\ldots ,n$, each $e\left(\cdot \right)\in E_{\left[0,n\right]}\left(\kappa \right)$ is also expressed as(61)$$\begin{array}{cc} \; & x_{1}=e_{1}\left(w\right)={\bar{e}}_{1}\left(w\right),\;x_{2}=e_{2}\left(w,x_{1},y_{1}\right)={\tilde{e}}_{2}\left(w,x_{1},v_{1},y_{1}\right)\overset{\left(a\right)}{=}{\bar{e}}_{2}\left(w,v_{1},y_{1}\right),\;\ldots ,\\ \; & x_{n}=e_{n}\left(w,x^{n-1},y^{n-1}\right)={\tilde{e}}_{n}\left(w,x^{n-1},v^{n-1},y^{n-1}\right)\overset{\left(a\right)}{=}{\bar{e}}_{n}\left(w,v^{n-1},y^{n-1}\right),\;w\in M^{\left(n\right)}. \end{array}$$
where (a) is due to the channel definition—i.e., the presence of $x^{t-1}$ in ${\tilde{e}}_{t}\left(\cdot ,v^{t-1},\cdot \right)$ can be removed, since it is redundant—and specified by $\left(v^{t-1},y^{t-1}\right)$. Consequently, we have the identity(62)$$\begin{array}{cc} {\bar{E}}_{\left[1,n\right]}\left(\kappa \right)≜ & \{x_{1}={\bar{e}}_{1}\left(w\right),x_{2}={\bar{e}}_{2}\left(w,v_{1},y_{1}\right)\ldots ,x_{n}={\bar{e}}_{n}\left(w,v^{n-1},y^{n-1}\right):\\ \; & \frac{1}{n}E^{\bar{e}}\left(\sum_{i=1}^{n}{\left(X_{t}\right)}^{2}\right)\leq \kappa \}=E_{\left[1,n\right]}\left(\kappa \right). \end{array}$$

**Notation 1.** 
*For the feedback code of Definition 3, with initial state $S_{1}=s$, known to the encoder and the decoder, the above sets $P_{\left[1,n\right]}\left(\kappa \right),{\bar{P}}_{\left[1,n\right]}\left(\kappa \right),E_{\left[1,n\right]},{\bar{E}}_{\left[1,n\right]}$ are replaced by $P_{\left[0,n\right]}^{s}\left(\kappa \right),{\bar{P}}_{\left[1,n\right]}^{s}\left(\kappa \right)$, $E_{\left[1,n\right]}^{s},{\bar{E}}_{\left[1,n\right]}^{s}$ to indicate that the distributions and codes are ${\bar{P}}_{t}\left(dx_{t}|v^{t-1},y^{t-1},s\right),t=1,\ldots ,x_{1}={\bar{e}}_{1}\left(w,s\right),x_{2}={\bar{e}}_{2}\left(w,v_{1},y_{1},s\right)\ldots ,x_{n}={\bar{e}}_{n}\left(w,v^{n-1},y^{n-1},s\right)$, etc., and these depend on s.*


In the next theorem, we present our preliminary equivalent sequential characterization of the Cover and Pombra characterization $C_{n}^{fb}\left(\kappa \right)$, i.e., of (Equation 33), under encoder strategies $E_{\left[0,n\right]}\left(\kappa \right)={\bar{E}}_{\left[0,n\right]}\left(\kappa \right)$ and channel input distributions $P_{\left[0,n\right]}\left(\kappa \right)={\bar{P}}_{\left[0,n\right]}\left(\kappa \right)$. Unlike the Cover and Pombra [5] realization of $X^{n}$, given by (Equation 35), at each time *t*, $X_{t}$ is driven by an orthogonal Gaussian process $Z_{t}$.

**Theorem 1.** 
*Information structures of maximizing distributions for AGN Channels*

*Consider the AGN channel (Equation 1), i.e., with noise distribution $P_{V^{n}}$, and the code of Definition 1. Then, the following hold:*

*(a) The following inequality holds:*

(63)
$$\begin{array}{c} \underset{{\bar{E}}_{\left[1,n\right]}\left(\kappa \right)}{\sup}\sum_{t=1}^{n}H^{\bar{e}}\left(Y_{t}|Y^{t-1}\right)\leq \underset{{\bar{P}}_{\left[1,n\right]}\left(\kappa \right)}{\sup}\sum_{t=1}^{n}H^{\bar{P}}\left(Y_{t}|Y^{t-1}\right) \end{array}$$

*where the conditional (differential) entropy $H^{\bar{e}}\left(Y_{t}|Y^{t-1}\right)$ is evaluated with respect to the probability distribution $P_{t}^{\bar{e}}\left(dy_{t}|y^{t-1}\right)$, defined by*

(64)
$$\begin{array}{c} P_{t}^{\bar{e}}\left(dy_{t}|y^{t-1}\right)=\int P_{t}\left(dy_{t}|{\bar{e}}_{t}\left(w,v^{t-1},y^{t-1}\right),v^{t-1}\right)\;P_{t}^{\bar{e}}\left(dw|y^{t-1}\right),\;t=0,\ldots ,n. \end{array}$$

*and $H^{\bar{P}}\left(Y_{t}|Y^{t-1}\right)$ is evaluated with respect to the probability distribution $P_{t}^{\bar{P}}\left(dy_{t}|y^{t-1}\right)$, defined by*

(65)
$$\begin{array}{c} P_{t}^{\bar{P}}\left(dy_{t}|y^{t-1}\right)=\int P_{t}\left(dy_{t}|x_{t},v^{t-1}\right)\;P_{t}^{\bar{P}}\left(dx_{t}|v^{t-1},y^{t-1}\right)\;P_{t}^{\bar{P}}\left(dv^{t-1}|y^{t-1}\right),\;t=0,\ldots ,n. \end{array}$$


*(b) The optimal channel input distribution $\left\{\bar{P}\left(dx_{t}|v^{t-1},y^{t-1}\right),t=1,\ldots ,n\right\}\in {\bar{P}}_{\left[1,n\right]}\left(\kappa \right)$, which maximizes $\sum_{t=1}^{n}H^{\bar{P}}\left(Y_{t}|Y^{t-1}\right)$ of part (a), i.e., the right-hand side of (Equation 63), is induced by an input process $X^{n}$, which is conditionally Gaussian, with linear conditional mean, nonrandom conditional covariance and given by*

(66)
$$\begin{array}{cc} \; & E^{\bar{P}}\left\{X_{t}|V^{t-1},Y^{t-1}\right\}=\left\{\begin{array}{ccc} \Gamma_{t}^{1}V^{t-1}+\Gamma_{t}^{2}Y^{t-1}, & for & t=2,\ldots ,n\\ 0, & for & t=1, \end{array}\right. \end{array}$$


(67)
$$\begin{array}{cc} \; & K_{X_{t}|V^{t-1},Y^{t-1}}\overset{▵}{=}cov\left(X_{t},X_{t}|V^{t-1},Y^{t-1}\right)=K_{Z_{t}}⪰0,\;t=1,\ldots ,n \end{array}$$

*and such that the average constraint holds and (Equation 18) is respected.*

*(c) The optimal channel input distribution $\left\{\bar{P}\left(dx_{t}|v^{t-1},y^{t-1}\right),t=1,\ldots ,n\right\}\in {\bar{P}}_{\left[1,n\right]}\left(\kappa \right)$ of part (b) is induced by a jointly Gaussian process $X^{n}$, with a realization given by*

(68)
$$\begin{array}{cc} \; & X_{t}=\sum_{j=1}^{t-1}\Gamma_{t,j}^{1}V_{j}+\sum_{j=1}^{t-1}\Gamma_{t,j}^{2}Y_{j}+Z_{t},\;X_{1}=Z_{1},\;t=2,\ldots ,n, \end{array}$$


(69)
$$\begin{array}{cc} \; & \;=\Gamma_{t}^{1}V^{t-1}+\Gamma_{t}^{2}Y^{t-1}+Z_{t}, \end{array}$$


(70)
$$\begin{array}{cc} \; & Z_{t}\in N\left(0,K_{Z_{t}}\right),\;t=1,\ldots ,n\;a\;Gaussian\;sequence, \end{array}$$


(71)
$$\begin{array}{cc} \; & Z_{t}\;independent\;of\;\left(V^{t-1},X^{t-1},Y^{t-1}\right),\;t=1,\ldots ,n, \end{array}$$


(72)
$$\begin{array}{cc} \; & Z^{n}\;independent\;of\;V^{n}, \end{array}$$


(73)
$$\begin{array}{cc} \; & \frac{1}{n}E^{\bar{P}}\left\{\sum_{t=1}^{n}{\left(X_{t}\right)}^{2}\right\}\leq \kappa , \end{array}$$


(74)
$$\begin{array}{cc} \; & \left(\Gamma_{t}^{1},\Gamma_{t}^{2},K_{Z_{t}}\right)\in \left(-\infty ,\infty \right)\times \left(-\infty ,\infty \right)\times \left[0,\infty \right)\;nonrandom. \end{array}$$


*(d) An equivalent characterization of the n–FTFI capacity $C_{n}^{fb}\left(\kappa \right)$, defined by (Equation 33) and (Equation 34), is given by*

(75)
$$\begin{array}{cc} C_{n}^{fb}\left(\kappa \right)= & \underset{\frac{1}{n}E^{\bar{P}}\left\{\sum_{t=1}^{n}{\left(X_{t}\right)}^{2}\right\}\leq \kappa }{\sup}\sum_{t=1}^{n}H^{\bar{P}}\left(Y_{t}|Y^{t-1}\right)-H\left(V^{n}\right) \end{array}$$


(76)
$$\begin{array}{cc} = & \underset{\frac{1}{n}E^{\bar{P}}\left\{\sum_{t=1}^{n}{\left(X_{t}\right)}^{2}\right\}\leq \kappa }{\sup}\sum_{t=1}^{n}\left\{H^{\bar{P}}\left(I_{t}\right)-H\left({\hat{I}}_{t}\right)\right\} \end{array}$$

*where $I_{t}\;and\;{\hat{I}}_{t}$ are innovations processes defined by*

(77)
$$\begin{array}{c} I_{t}\overset{▵}{=}Y_{t}-E^{\bar{P}}\left\{Y_{t}|Y^{t-1}\right\},\;{\hat{I}}_{t}\overset{▵}{=}V_{t}-E\left\{V_{t}|V^{t-1}\right\} \end{array}$$

*where parts (b) and (c) hold, and where the supremum is over all $(\Gamma_{t}^{1},\Gamma_{t}^{2},K_{Z_{t}}),t=1,\ldots ,n$ of the realization of part (c), which induces the distribution ${\bar{P}}_{t}\left(dx_{t}|v^{t-1},y^{t-1}\right),t=1,\ldots ,n$.*



**Proof.** See Section A.1.  □

**Remark 2.** 
*For the code of Definition 3 that assumes knowledge of the initial state $S_{1}=s$, it is easy to verify that $C_{n}^{fb}\left(\kappa ,s\right)$ is directly obtained from Theorem 1, as a degenerate case (an independent derivation is easily produced following the derivation of Corollary 11, with slight variations).*


By utilizing Theorem 1, we can derive the converse coding theorems stated below for the feedback codes of Definitions 1 and 3.

**Theorem 2.** 
*Converse coding theorems for codes of Definitions 1 and 3*

*Consider the AGN channel (Equation 1).*

*(a) Any achievable rate R for the code of Definition 1 satisfies*

(78)
$$\begin{array}{cc} R\leq & C^{fb}\left(\kappa \right)\overset{▵}{=}\lim_{n⟶\infty }\frac{1}{n}C_{n}^{fb}\left(\kappa \right), \end{array}$$


(79)
$$\begin{array}{cc} C_{n}^{fb}\left(\kappa \right)= & \underset{{\bar{P}}_{t}\left(dx_{t}|v^{t-1},y^{t-1}\right),t=1,\ldots ,n:\;\frac{1}{n}E^{\bar{P}}\left\{\sum_{t=1}^{n}{\left(X_{t}\right)}^{2}\right\}\leq \kappa }{\sup}\sum_{t=1}^{n}H^{\bar{P}}\left(Y_{t}|Y^{t-1}\right)-H\left(V^{n}\right) \end{array}$$

*provided the supremum exists and the limit exists, where the right-hand side of (Equation 79) is given in Theorem 1(d).*

*(b) Any achievable rate R for the code of Definition 3 (with initial state $S_{1}=s$) satisfies*

(80)
$$\begin{array}{cc} R\leq & C^{fb}\left(\kappa ,s\right)\overset{▵}{=}\lim_{n⟶\infty }\frac{1}{n}C_{n}^{fb}\left(\kappa ,s\right), \end{array}$$


(81)
$$\begin{array}{cc} C_{n}^{fb}\left(\kappa ,s\right)= & \underset{{\bar{P}}_{t}\left(dx_{t}|v^{t-1},y^{t-1},s\right),t=1,\ldots ,n:\;\frac{1}{n}E_{s}^{\bar{P}}\left\{\sum_{t=1}^{n}{\left(X_{t}\right)}^{2}|S_{1}\right\}\leq \kappa }{\sup}\sum_{t=1}^{n}H^{\bar{P}}\left(Y_{t}|Y^{t-1},s\right)-H\left(V^{n}|s\right). \end{array}$$

*where $E_{s}^{\bar{P}}\left\{\cdot \right\}$ means the expectation is for a fixed $S_{1}=s$, provided the supremum exists and the limit exists, and where the right-hand side of (Equation 81) is obtained from Theorem 1, part (d), by replacing all conditional distributions, entropies, etc., for fixed initial state $S_{1}=s$ (see Notation 1).*



**Proof.** Following on from standard arguments, we use Fano’s inequality (see also [5]) and Theorem 1.  □

In the next remark, we clarify the equivalence of Theorem 1(d) to Cover and Pombra [5].

**Remark 3.** 
*Relation of Theorem 1 and Cover and Pombra [5]*

*(a) From the realization of $X^{n}$ given by (Equation 68), we can recover the Cover and Pombra [5] realization (Equation 35) by recursive substitution of $Y^{t-1}$ into the right-hand side of (Equation 68), as follows:*

(82)
$$\begin{array}{cc} X_{t}= & \sum_{j=1}^{t-1}\Gamma_{t,j}^{1}V_{j}+\sum_{j=1}^{t-1}\Gamma_{t,j}^{2}Y_{j}+Z_{t} \end{array}$$


(83)
$$\begin{array}{cc} = & \sum_{j=1}^{t-1}\Gamma_{t,j}^{1}V_{j}+\sum_{j=1}^{t-2}\Gamma_{t,j}^{2}Y_{j}+\Gamma_{t,t-1}^{2}\left(X_{t-1}+Z_{t-1}\right)+Z_{t} \end{array}$$


(84)
$$\begin{array}{cc} = & \sum_{j=1}^{t-1}B_{t,j}V_{j}+{\bar{Z}}_{t},\;by\;recursive\;substitution\;of\;X_{1},\ldots ,X_{t-1},Y_{1},\ldots ,Y_{t-2} \end{array}$$

*for some ${\bar{Z}}_{t}\in \left(0,K_{{\bar{Z}}_{t}}\right)$ which is jointly correlated, and some nonrandom $B_{t,j}$, as given by (Equation 35) and (Equation 36).*

*(b) Unlike the Cover and Pombra [5] realization of $X^{n}$, i.e., (Equation 35), the realization of $X^{n}$ given by (Equation 68) or in vector form by (Equation 69) is such that, at each time t, $X_{t}$ depends on $\left(V^{t-1},Y^{t-1},Z_{t}\right)$ or in vector form on $\left(V^{t-1},Y^{t-1},Z_{t}\right)$, where $Z^{t}$ is an innovations or orthogonal process, i.e., (Equation 71) holds.*

*(c) In subsequent parts of the paper, we derive an equivalent sequential characterization of the Cover and Pombra n–FTFI capacity (Equation 34), which is simplified further with the use of a sufficient statistic (that satisfies a Markov recursion).*



To characterize $C_{n}^{fb}\left(\kappa \right)$ using Theorem 1, part (d), we need to compute the (differential) entropy $H(V^{n})$ of $V^{n}$. The following lemma is useful in this respect:

**Lemma 1.** 
*Entropy $H(V^{n})$ calculation from generalized Kalman filter of the PO-SS noise realization.*

*Consider the PO-SS realization of $V^{n}$ of Definition 2. Define the conditional covariance and conditional mean of $S_{t}$ given $V^{t-1}$ by*

(85)
$$\begin{array}{cc} \Sigma_{t}\overset{▵}{=} & cov(S_{t},S_{t}|V^{t-1})=E\left\{\left(S_{t}-{\hat{S}}_{t}\right){\left(S_{t}-{\hat{S}}_{t}\right)}^{T}|V^{t-1}\right\},\;{\hat{S}}_{t}\overset{▵}{=}E\left\{S_{t}|V^{t-1}\right\},\;t=2,\ldots ,n, \end{array}$$


(86)
$$\begin{array}{cc} \Sigma_{1}\overset{▵}{=} & cov(S_{1},S_{1})=K_{S_{1}},\;{\hat{S}}_{1}\overset{▵}{=}\mu_{S_{1}}. \end{array}$$

*Then, the following hold:*

*(a) The conditional distribution of $V_{t}$ conditioned on $V^{t-1}$ is Gaussian, i.e.,*

(87)
$$\begin{array}{c} P_{V_{t}|V^{t-1}}\in N\left(\mu_{V_{t}|V^{t-1}},K_{V_{t}|V^{t-1}}\right),\;t=1,\ldots ,n\; \end{array}$$

*where $\mu_{V_{t}|V^{t-1}}\overset{▵}{=}E\left\{V_{t}|V^{t-1}\right\},K_{V_{t}|V^{t-1}}\overset{▵}{=}cov(V_{t},V_{t}|V^{t-1})$.*

*(b) The conditional mean and covariance $\mu_{V_{t}|V^{t-1}},K_{V_{t}|V^{t-1}}$ are given by the generalized Kalman filter recursions, as follows:*

*(i) The optimal mean square error estimate ${\hat{S}}_{t}$ satisfies the generalized Kalman filter recursion*

(88)
$$\begin{array}{cc} \; & {\hat{S}}_{t+1}=A_{t}{\hat{S}}_{t}+M_{t}\left(\Sigma_{t}\right){\hat{I}}_{t},\;{\hat{S}}_{1}=\mu_{S_{1}}, \end{array}$$


(89)
$$\begin{array}{cc} \; & M_{t}\left(\Sigma_{t}\right)\overset{▵}{=}\left(A_{t}\Sigma_{t}C_{t}^{T}+B_{t}K_{W_{t}}N_{t}^{T}\right){\left(N_{t}K_{W_{t}}N_{t}^{T}+C_{t}\Sigma_{t}C_{t}^{T}\right)}^{-1}, \end{array}$$


(90)
$$\begin{array}{cc} \; & {\hat{I}}_{t}\overset{▵}{=}V_{t}-E\left\{V_{t}|V^{t-1}\right\}=V_{t}-C_{t}{\hat{S}}_{t}=C_{t}\left(S_{t}-{\hat{S}}_{t}\right)+N_{t}W_{t},\;t=1,\ldots ,n, \end{array}$$


(91)
$$\begin{array}{cc} \; & {\hat{I}}_{t}\in N\left(0,K_{{\hat{I}}_{t}}\right),\;t=1,\ldots ,n\;is\;an\;orthogonal\;innovations\;process,\;i.e.,\;{\hat{I}}_{t}\;is\;independent\;of\\ \; & {\hat{I}}_{s},\;for\;all\;t\neq s,\;and\;{\hat{I}}_{t}\;is\;independent\;ofV^{t-1}, \end{array}$$


(92)
$$\begin{array}{cc} \; & K_{{\hat{I}}_{t}}\overset{▵}{=}cov\left({\hat{I}}_{t},{\hat{I}}_{t}\right)=C_{t}\Sigma_{t}C_{t}^{T}+N_{t}K_{W_{t}}N_{t}^{T}. \end{array}$$

*(ii) The error $E_{t}\overset{▵}{=}S_{t}-{\hat{S}}_{t}$ satisfies the recursion*

(93)
$$\begin{array}{cc} E_{t+1}= & M_{t}^{CL}\left(\Sigma_{t}\right)E_{t}+\left(B_{t}-M\left(\Sigma_{t}\right)N_{t}\right)W_{t},\;E_{1}=S_{1}-{\hat{S}}_{1},\;t=1,\ldots ,n, \end{array}$$


(94)
$$\begin{array}{cc} M_{t}^{CL}\left(\Sigma_{t}\right)\overset{▵}{=} & A_{t}-M_{t}\left(\Sigma_{t}\right)C_{t}. \end{array}$$

*(iii) The covariance of the error is such that $E\left\{E_{t}E_{t}^{T}\right\}=\Sigma_{t}$ and satisfies the generalized matrix DRE*

(95)
$$\begin{array}{cc} \Sigma_{t+1}= & A_{t}\Sigma_{t}A_{t}^{T}+B_{t}K_{W_{t}}B_{t}^{T}-\left(A_{t}\Sigma_{t}C_{t}^{T}+B_{t}K_{W_{t}}N_{t}^{T}\right){\left(N_{t}K_{W_{t}}N_{t}^{T}+C_{t}\Sigma_{t}C_{t}^{T}\right)}^{-1}\\ \; & \;.{\left(A_{t}\Sigma_{t}C_{t}^{T}+B_{t}K_{W_{t}}N_{t}^{T}\right)}^{T},\;t=1,\ldots ,n,\;\Sigma_{1}=K_{S_{1}}⪰0,\;\Sigma_{t}⪰0. \end{array}$$

*(iv) The conditional mean and covariance $\mu_{V_{t}|V^{t-1}},K_{V_{t}|V^{t-1}}$ are given by*

(96)
$$\begin{array}{cc} \mu_{V_{t}|V^{t-1}}= & C_{t}{\hat{S}}_{t},\;t=1,\ldots ,n, \end{array}$$


(97)
$$\begin{array}{cc} K_{V_{t}|V^{t-1}}= & K_{{\hat{I}}_{t}}=C_{t}\Sigma_{t}C_{t}^{T}+N_{t}K_{W_{t}}N_{t}^{T},\;t=1,\ldots ,n. \end{array}$$

*(v) The entropy of $V^{n}$ is given by*

(98)
$$\begin{array}{c} H\left(V^{n}\right)=\sum_{t=1}^{n}H\left({\hat{I}}_{t}\right)=\frac{1}{2}\sum_{t=1}^{n}\log\left(2\pi e\left[C_{t}\Sigma_{t}C_{t}^{T}+N_{t}K_{W_{t}}N_{t}^{T}\right]\right) \end{array}$$




**Proof.** (a,b).(i–iv). The generalized Kalman filter of the PO-SS realization of $V^{n}$ and accompanied statements can be found in many textbooks, i.e., [16]. However, it is noted that ${\hat{I}}_{t},t=2,\ldots ,n$, ${\hat{I}}_{1}=V_{1}$ are all independent Gaussian. For example, to show (Equation 93), we write the recursion for $E_{t}=S_{t}-{\hat{S}}_{t}$ using part (i) and the realization of $S_{t}$, part (b). (v) By the chain rule of joint entropy, we have(99)$$\begin{array}{cc} H(V^{n})= & H\left(V_{1}\right)+\sum_{t=2}^{n}H\left(V_{t}|V^{t-1}\right) \end{array}$$(100)$$\begin{array}{cc} = & H\left(V_{1}\right)+\sum_{t=2}^{n}H\left(V_{t}-E\left\{V_{t}|V^{t-1}\right\}|V^{t-1}\right) \end{array}$$(101)$$\begin{array}{cc} = & H\left(V_{1}\right)+\sum_{t=2}^{n}H\left({\hat{I}}_{t}\right),\;\text{by orthogonality of}\;{\hat{I}}_{t}\overset{▵}{=}V_{t}-E\left\{V_{t}|V^{t-1}\right\}\mathrm{and}V^{t-1} \end{array}$$
From (Equation 101) and (Equation 97), we have (Equation 98) from the entropy formula of Gaussian RVs.  □

The next corollary of the entropy $H(V^{n}|s)$ follows directly from Lemma 1 when $S_{1}=s$ is fixed.

**Corollary 1.** 
*Conditional entropy $H\left(V^{n}|s\right),S_{1}=s$ of the PO-SS noise realization.*

*Consider the PO-SS realization of $V^{n}$ of Definition 2, for fixed $S_{1}=s$, and denote the state process generated by recursion (Equation 19) via (we often use the notation $S_{t}=S_{t}^{s}$ to emphasize that the $S_{t}$ process is generated for $S_{1}=S_{1}^{s}=s$ fixed) $S_{t}=S_{t}^{s},t=2,\ldots ,n,S_{1}=S_{1}^{s}=s$. Replace the conditional covariance and conditional mean (Equation 85) and (Equation 86) by*

(102)
$$\begin{array}{cc} \Sigma_{t}^{s}\overset{▵}{=} & cov(S_{t}^{s},S_{t}^{s}|V^{t-1},S_{1}^{s})=E\left\{\left(S_{t}^{s}-{\hat{S}}_{t}^{s}\right){\left(S_{t}^{s}-{\hat{S}}_{t}^{s}\right)}^{T}|V^{t-1},S_{1}^{s}\right\}, \end{array}$$


(103)
$$\begin{array}{cc} {\hat{S}}_{t}^{s}\overset{▵}{=} & E\left\{S_{t}^{s}|V^{t-1},S_{1}^{s}\right\},\;t=2,\ldots ,n,\;\;S_{1}^{s}=s,\;{\hat{S}}_{1}^{s}\overset{▵}{=}s,\;\Sigma_{1}^{s}\overset{▵}{=}cov(S_{1}^{s},S_{1}^{s}\left|S_{1}^{s}\right)=0. \end{array}$$

*Then, all statements of Lemma 1 hold, with the following changes:*

(104)
$$\begin{array}{c} \Sigma_{t}⟼\Sigma_{t}^{s},\;\Sigma_{1}^{s}=0,\;P_{V_{t}|V^{t-1}}⟼P_{V_{t}|V^{t-1},S_{1}^{s}},\;{\hat{S}}_{t}⟼{\hat{S}}_{t}^{s},\;{\hat{S}}_{1}^{s}=s,\;etc,\;t=1,\ldots ,n. \end{array}$$

*In particular, the conditional entropy of the $V^{n}$ conditioned on $S_{1}=S_{1}^{s}=s$ is given by*

(105)
$$\begin{array}{c} H\left(V^{n}|s\right)=\frac{1}{2}\sum_{t=1}^{n}\log\left(2\pi e\left[C_{t}\Sigma_{t}^{s}C_{t}^{T}+N_{t}K_{W_{t}}N_{t}^{T}\right]\right) \end{array}$$

*where $\Sigma_{t}^{s},t=2,\ldots ,n$ satisfies the generalized DRE (Equation 95) with initial condition $\Sigma_{1}^{s}=0$.*


**Proof.** We continue directly from Lemma 1 and (Equation 102), (Equation 103).  □

Next, we introduce an example of a PO-SS realization of the noise that we often use in this paper.

**Example 1.** 
*A time-varying PO-SS$\left(a_{t},c_{t},b_{t}^{1},b_{t}^{2},d_{t}^{1},d_{t}^{2}\right)$ noise realization is defined by*

(106)
$$\begin{array}{cc} \; & S_{t+1}=a_{t}S_{t}+b_{t}^{1}W_{t}^{1}+b_{t}^{2}W_{t}^{2},\;t=1,2,\ldots ,n-1 \end{array}$$


(107)
$$\begin{array}{cc} \; & V_{t}=c_{t}S_{t}+d_{t}^{1}W_{t}^{1}+d_{t}^{2}W_{t}^{2},\;t=1,\ldots ,n, \end{array}$$


(108)
$$\begin{array}{cc} \; & S_{1}\in N\left(\mu_{S_{1}},K_{S_{1}}\right),\;K_{S_{1}}\geq 0,\;W_{t}^{i}\in N\left(0,K_{W_{t}^{i}}\right),\;K_{W_{t}^{i}}\geq 0,\;i=1,2,\;t=1,\ldots ,n, \end{array}$$


(109)
$$\begin{array}{cc} \; & W^{1,n}\;and\;W^{2,n}indep.\;seq.\;and\;indep.\;of\;S_{1}, \end{array}$$


(110)
$$\begin{array}{cc} \; & a_{t}\in R,\;c_{t}\in R,\;b_{t}^{i}\in R,\;d_{t}^{i}\in R,\;i=1,2,\forall t\;are\;nonrandom, \end{array}$$


(111)
$$\begin{array}{cc} \; & b_{t}\circ b_{t}\overset{▵}{=}{\left(b_{t}^{1}\right)}^{2}K_{W_{t}^{1}}+{\left(b_{t}^{2}\right)}^{2}K_{W_{t}^{2}},\;b_{t}\circ d_{t}\overset{▵}{=}b_{t}^{1}K_{W_{t}^{1}}d_{t}^{1}+b_{t}^{2}K_{W_{t}^{2}}d_{t}^{2}, \end{array}$$


(112)
$$\begin{array}{cc} \; & d_{t}\circ d_{t}\overset{▵}{=}{\left(d_{t}^{1}\right)}^{2}K_{W_{t}^{1}}+{\left(d_{t}^{2}\right)}^{2}K_{W_{t}^{2}}>0,\;\forall t. \end{array}$$



The next corollary is an application of Lemma 1 to the time-varying PO-SS noise of Example 1.

**Corollary 2.** 
*The entropy $H(V^{n})$ of the PO-SS$\left(a_{t},c_{t},b_{t}^{1},b_{t}^{2},d_{t}^{1},d_{t}^{2}\right)$ noise of Example 1 is computed from Lemma 1 with the following changes:*

(113)
$$\begin{array}{cc} \; & C_{t}⟼c_{t},\;A_{t}⟼a_{t},\;B_{t}K_{W_{t}}N_{t}^{T}⟼b_{t}\circ d_{t},\\ \; & B_{t}K_{W_{t}}B_{t}^{T}⟼b_{t}\circ b_{t},\;N_{t}K_{W_{t}}N_{t}^{T}⟼d_{t}\circ d_{t},\;t=1,\ldots ,n. \end{array}$$



**Proof.** This is easily verified.  □

From Corollary 2, we have the following observations:

**Remark 4.** 
*Consider the PO-SS$\left(a_{t},c_{t},b_{t}^{1},b_{t}^{2},d_{t}^{1},d_{t}^{2}\right)$ noise of Example 1. Then, the following hold.*

*(a) Consider the code of Definition 2. At each time t, the optimal channel input process $X^{n}$ is either realized by (Equation 35), or equivalently by (Equation 68), i.e.,*

(114)
$$\begin{array}{c} X_{t}=\sum_{j=1}^{t-1}B_{t,j}V_{t,j}+{\bar{Z}}_{t}=\sum_{j=1}^{t-1}\Gamma_{t,j}^{1}V_{j}+\sum_{j=1}^{t-1}\Gamma_{t,j}^{2}Y_{j}+Z_{t},\;t=1,2,\ldots ,n. \end{array}$$

*Moreover, $X_{t}$ cannot be expressed in terms of the state $S^{t}$ because, by (Equation 106) and (Equation 107), the noise sequence $V^{t-1}$ does not specify $S^{t}$ for t=1,…,n.*

*(b) Consider the code of Definition 3, i.e., with a fixed initial state $S_{1}=S_{1}^{s}=s$. With Corollary 1 using (Equation 113) $H(V^{n}|s)$ is computed from Lemma 1, with $\Sigma_{1}=\Sigma_{1}^{s}=0$, and (Equation 105) reduces to*

(115)
$$\begin{array}{c} H\left(V^{n}|s\right)=\frac{1}{2}\sum_{t=1}^{n}\log\left(2\pi e\left[{\left(c_{t}\right)}^{2}\Sigma_{t}^{s}+d_{t}\circ d_{t}\right]\right) \end{array}$$

*where $\Sigma_{t}^{s}$ is the solution of (Equation 95) with $\Sigma_{1}=\Sigma_{1}^{s}=0$ (using (Equation 113)).*



We also apply our results to various versions of the autoregressive moving average (ARMA) noise model, such as the double-side and single-sided, stationary version of the ARMA noise, previously analyzed in [4] and in many other papers, to illustrate fundamental differences of Case (I) and Case (II) formulations.

**Example 2.** 
*The time-invariant ARMA(a,c) noise*

*(a) The time-invariant one-sided, stable or unstable, autoregressive moving average (ARMA(a,c),a∈(−∞,∞),c∈(−∞,∞)) noise is defined by*

(116)
$$\begin{array}{cc} \; & V_{t}=cV_{t-1}+W_{t}-aW_{t-1},\;\forall t\in Z_{+}\overset{▵}{=}\left\{1,2,\ldots \right\}, \end{array}$$


(117)
$$\begin{array}{cc} \; & V_{0}\in N\left(0,K_{V_{0}}\right),\;K_{V_{0}}\geq 0,\;W_{0}\in N\left(0,K_{W_{0}}\right),\;K_{W_{0}}\geq 0, \end{array}$$


(118)
$$\begin{array}{cc} \; & W_{t}\in N\left(0,K_{W}\right),\;K_{W}>0,\;\forall t\in Z_{+},\;\left\{W_{0},W_{1},\ldots ,W_{n}\right\}\;indep.\;seq.\;and\;indep.\;of\;V_{0}, \end{array}$$


(119)
$$\begin{array}{cc} \; & c\in (-\infty ,\infty ),\;a\in (-\infty ,\infty ),\;c\neq a. \end{array}$$

*To express the AR(a,c) in state space form, we define the state variable of the noise by*

(120)
$$\begin{array}{c} S_{t}\overset{▵}{=}\frac{cV_{t-1}-aW_{t-1}}{c-a},\;\forall t\in Z_{+} \end{array}$$

*Then, the state space realization of $V^{n}$ is*

(121)
$$\begin{array}{cc} \; & S_{t+1}=cS_{t}+W_{t},\;\forall t\in Z_{+}, \end{array}$$


(122)
$$\begin{array}{cc} \; & V_{t}=\left(c-a\right)S_{t}+W_{t},\;\forall t\in Z_{+}, \end{array}$$


(123)
$$\begin{array}{cc} \; & K_{S_{1}}=\frac{{\left(c\right)}^{2}K_{V_{0}}+{\left(a\right)}^{2}K_{W_{0}}}{{\left(c-a\right)}^{2}},\;K_{V_{0}}\geq 0,\;K_{W_{0}}\geq 0\;both\;given. \end{array}$$

*We note that the AR(a,c) is not necessarily stationary or asymptotically stationary.*

*A special case of the AR(a,c) is the AR(c) noise (i.e., with a=0) defined by*

(124)
$$\begin{array}{cc} \; & V_{t}=cV_{t-1}+W_{t},\;t=1,2,\ldots ,\;K_{V_{0}}\geq 0,\;K_{W}>0. \end{array}$$

*(b) Double-sided wide-sense stationary ARMA(a,c),a∈[−1,1],c∈(−1,1) noise. A double-sided wide-sense stationary ARMA(a,c) noise is defined by*

(125)
$$\begin{array}{c} V_{t}=cV_{t-1}+W_{t}-aW_{t-1},\;\forall t\in Z\overset{▵}{=}\left\{\ldots ,-1,0,1,\ldots \right\},\;|a|\leq 1,\;|c|<1. \end{array}$$

*where $W_{t},\forall t\in Z$ is an independent and identically distributed Gaussian sequence, i.e., $W_{t}\in N\left(0,K_{W}\right)$, ∀t. The power spectral density (PSD) of the wide-sense stationary noise (this corresponds to [4] (Equation (43) with L=1)) is given by*

(126)
$$\begin{array}{cc} S_{V}\left(e^{j\theta }\right)\overset{▵}{=} & K_{W}\frac{\left(1-ae^{i\theta }\right)\left(1-ae^{-i\theta }\right)}{\left(1-ce^{i\theta }\right)\left(1-ce^{-i\theta }\right)},\;|c|<1,\;|a|\leq 1,\;c\neq a,\;K_{W}>0. \end{array}$$

*We define the state process by*

(127)
$$\begin{array}{c} S_{t}\overset{▵}{=}\frac{cV_{t-1}-aW_{t-1}}{c-a},\;\forall t\in Z. \end{array}$$

*Then, the stationary state space realization of $V_{t},\forall t\in Z$ is*

(128)
$$\begin{array}{cc} \; & S_{t+1}=cS_{t}+W_{t},\;\forall t\in Z, \end{array}$$


(129)
$$\begin{array}{cc} \; & V_{t}=\left(c-a\right)S_{t}+W_{t},\;\forall t\in Z \end{array}$$

*provided that the initial covariances $cov\left(S_{t},S_{t}\right),cov\left(S_{t},V_{t}\right),cov\left(V_{t},V_{t}\right)$ are chosen appropriately to ensure stationarity (see Proposition 1).*

*(c) One-sided wide-sense stationary ARMA(a,c),a∈[−1,1],c∈(−1,1). The one-sided wide-sense stationary ARMA(a,c) noise is defined as in part (b) with $\forall t\in Z\overset{▵}{=}\left\{\ldots ,-1,0,1,\ldots ,\right\}$ replaced by $\forall t\in Z_{+}\overset{▵}{=}\left\{1,2,\ldots ,\right\}$ and (Equation 127)–(Equation 129), which hold, $\forall t\in Z_{+}$, provided that the initial covariances are chosen appropriately (see Proposition 1).*



For the AR(a,c) noise, we clarify differences of the feedback codes of Definitions 1 and 3 in the next remark, as well as of the Case (I) formulation versus the Case (II) formulation (and we discuss implication to results in [4,7,9,11,12]).

**Remark 5.** 
*ARMA(a,c) noise of Example 2*

*(a) Consider any of the AR(a,c) of Example 2. For the code of Definition 2, the channel input process $X^{n}$ cannot be expressed in terms of the state $S^{n}$ (also see Remark 4(a)).*

*(b) Consider the non-stationary AR(a,c),a∈(−∞,∞),c∈(−∞,∞) of Example 2(a).*

*(i) Assume the code of Definition 3, with initial state $V_{0}=v_{0}$ known to the encoder. By (Equation 120),*

(130)
$$\begin{array}{c} S_{1}=S_{1}^{v_{0}}=\frac{cv_{0}-aW_{0}}{c-a},\;V_{0}=v_{0}; \end{array}$$

*hence, knowledge of $V_{0}=v_{0}$ at the encoder does not determine $S_{1}^{v_{0}}$ because the encoder requires knowledge of $W_{0}$ for this to hold. It then follows that $H(V^{n}|v_{0})$ is computed from Corollary 1 as follows:*

(131)
$$\begin{array}{c} \Sigma_{1}=\Sigma_{1}^{v_{0}}=\frac{{\left(a\right)}^{2}K_{W_{0}}}{{\left(c-a\right)}^{2}},\;\text{(105)}\;reduces\;to\;H\left(V^{n}|v_{0}\right)=\frac{1}{2}\sum_{t=1}^{n}\log\left(2\pi e\left[{\left(c\right)}^{2}\Sigma_{t}^{v_{0}}+K_{W}\right]\right) \end{array}$$

*where $\Sigma_{t}^{v_{0}}$ is the solution of (Equation 95) with initial data $\Sigma_{1}=K_{S_{1}}=\Sigma_{1}^{v_{0}},K_{W_{0}}\geq 0$.*

*(ii) Assume the code of Definition 3, with initial state $S_{1}=s$ or $\left(V_{0},W_{0}\right)=\left(v_{0},w_{0}\right)$, known to the encoder. Then, by Corollary 1,*

(132)
$$\begin{array}{c} H\left(V^{n}|v_{0},w_{0}\right)=\frac{1}{2}\sum_{t=1}^{n}\log\left(2\pi eK_{W}\right). \end{array}$$

*By (Equation 120), $S_{1}\overset{▵}{=}\frac{cV_{0}-aW_{0}}{c-a}$, and a necessary condition for Conditions 1 of Section 1.1 to hold is the following: both $\left(V_{0},W_{0}\right)=\left(v_{0},w_{0}\right)$ are known to the encoder and the decoder.*

*(c) The statements of parts (a) and (b) also hold for the double-sided and the one-sided wide-sense stationary AR(a,c),a∈[−1,1],c∈(−1,1) of Example 2(b,c).*

*(d) The Case (II) formulation discussed in Section 1.1 requires Conditions 1 and 2 to hold. For any of the AR(a.c) noise models, Conditions 1 and 2 hold if and only if $S_{1}=s_{1}$ or $\left(V_{0},W_{0}\right)=\left(v_{0},w_{0}\right)$ are known to the encoder. Clearly, the values of $H(V^{n})$ under the Case (I) formulation are fundamentally different from the values of $H\left(V^{n}|s\right),S_{1}=s$ under the Case (II) formulation. Consequently, in general, $C_{n}^{fb}\left(\kappa \right)$ given by (Equation 75) is fundamentally different from $C_{n}^{fb}\left(\kappa ,s\right)$, i.e., it corresponds to a fixed initial state $S_{1}=s$, known to the encoder and the decoder, and to the channel input distribution.*

*(e) From parts (a–d), the characterization of feedback capacity for the stationary ARMA(a,c), a∈[−1,1],c∈(−1,1) given in [4] (Theorem 6.1, $C_{FB}$) (which is derived based on [4] (Lemma 6.1)) presupposed the encoder and the decoder assumed knowledge of $S_{1}=S_{1}^{s}=s$,*


*In fact, the formulas of capacity in [4,7,8] use $H\left(V^{n}|v_{0},w_{0}\right)=\frac{1}{2}\sum_{t=1}^{n}\log\left(2\pi eK_{W}\right)$.*


In the next proposition, we state conditions for the stable realizations of Example 2(a), i.e., AR(a,c),a∈[−1,1],c∈(−1,1) to be asymptotically stationary, and for the realizations of Example 2(b,c) to be stationary. We should emphasize that for stationary noise, we need to determine the initial conditions of the generalized Kalman filter of Lemma 1 to correspond to the stationary noise.

**Proposition 1.** 
*Asymptotically stationary and stationary ARMA(a,c) noises of Example 2*

*(a) The realization of the double-sided ARMA(a,c),a∈[−1,1],c∈(−1,1) noise of Example 2(b) is stationary if the following conditions hold:*

(133)
$$\begin{array}{cc} \; & d_{11}\overset{▵}{=}cov\left(S_{t},S_{t}\right)=K_{S_{t}},\;d_{12}\overset{▵}{=}cov\left(S_{t},V_{t}\right)=K_{S_{t},V_{t}},\\ \; & d_{22}\overset{▵}{=}cov\left(V_{t},V_{t}\right)=K_{V_{t}},\;are\;constant\;\forall t\in Z. \end{array}$$

*where the constants $\left(d_{11},d_{12},d_{22}\right)$ are given by*

(134)
$$\begin{array}{cc} \; & d_{11}=\frac{K_{W}}{1-c^{2}},\;d_{12}=\frac{\left(c-a\right)K_{W}}{1-c^{2}},\;d_{22}=\frac{{\left(c-a\right)}^{2}K_{W}}{1-c^{2}}+K_{W}. \end{array}$$

*Similarly, the one-sided ARMA(a,c),a∈[−1,1],c∈(−1,1) noise of Example 2(c) is stationary if the above equations hold $\forall t\in Z_{+}\overset{▵}{=}\left\{1,2,\ldots \right\}$.*

*(b) The realization of the ARMA(a,c) noise of Example 2(a) is asymptotically stationary if a∈[−1,1],c∈(−1,1).*

*(c) For the stationary realization of part (a), the optimal conditional variance and conditional mean of $S_{t}$ from $\left(V_{0},V_{1},V_{2},\ldots ,V_{t-1}\right)$, i.e., $\Sigma_{t}\overset{▵}{=}cov(S_{t},S_{t}|V^{t-1},V_{0}),{\hat{S}}_{t}\overset{▵}{=}E\left\{S_{t}|V^{t-1},V_{0}\right\}$ are defined by the generalized Kalman filter given by*

(135)
$$\begin{array}{cc} \; & {\hat{S}}_{t+1}=c{\hat{S}}_{t}+\left({\left(c\right)}^{2}\Sigma_{t}+K_{W}\right){\left(K_{W}+{\left(c-a\right)}^{2}\Sigma_{t}\right)}^{-1}\left(V_{t}-\left(c-a\right){\hat{S}}_{t}\right),\;t=1,2,\ldots , \end{array}$$


(136)
$$\begin{array}{cc} \; & \Sigma_{t+1}={\left(c\right)}^{2}\Sigma_{t}+K_{W_{t}}-{\left(c\left(a-c\right)\Sigma_{t}+K_{W}\right)}^{2}{\left(K_{W}+{\left(c(a-c)\right)}^{2}\Sigma_{t}\right)}^{-1} \end{array}$$

*initialized at the initial data*

(137)
$$\begin{array}{cc} \; & {\hat{S}}_{1}\overset{▵}{=}E\left\{S_{1}|V_{0}\right\}=\frac{cd_{12}+K_{W}}{d_{22}}V_{0}, \end{array}$$


(138)
$$\begin{array}{cc} \; & \Sigma_{1}\overset{▵}{=}cov\left(S_{1},S_{1}|V_{0}\right)=d_{11}-\frac{{\left(d_{12}\right)}^{2}}{d_{22}}. \end{array}$$

*(i) If the conditioning information is $\left(V_{-N},\ldots ,V_{0},V_{1},V_{2},\ldots ,V_{t-1}\right)$, then the generalized Kalman filters (Equation 135) and (Equation 136) still hold and are initialized at the initial data*

(139)
$$\begin{array}{cc} \; & {\hat{S}}_{-N+1}=\frac{cd_{12}+K_{W}}{d_{22}}V_{-N}, \end{array}$$


(140)
$$\begin{array}{cc} \; & \Sigma_{-N+1}=d_{11}-\frac{{\left(d_{12}\right)}^{2}}{d_{22}}. \end{array}$$

*(ii) If the inital data $V_{0}$ are not available, then the generalized Kalman filter is initialized at initial data ${\hat{S}}_{1}=0$, $\Sigma_{1}=cov\left(S_{1},S_{1}\right)=d_{11}$.*



**Proof.** See Section A.2.  □

**Remark 6.** 
*Consider the stationary double-sided or one-sided ARMA(a,c),a∈[−1,1],c∈(−1,1) of Example 2. From Proposition 1 and, in particular, the initial data ${\hat{S}}_{1},\Sigma_{1}$ stated in (Equation 137) and (Equation 138), it is clear that even if the encoder and the decoder know the initial state $V_{0}$. Thus, $H\left(V^{n}|v_{0}\right)\neq \frac{1}{2}\sum_{t=1}^{n}\log\left(2\pi eK_{W}\right)$. In this case, the value of $C_{n}^{fb}\left(\kappa ,v_{0}\right)$ defined by (Equation 81) is fundamentally different from the formulation in [4,7,8], leading to the characterization of feedback capacity [4] (Theorem 6.1).*


In the next corollary, we further clarify the difference between Case (I) formulation and Case (II) formulation, by stating the analog of Theorem 1 for the code of Definition 3, i.e., when $S_{1}=S_{1}^{s}=s$ is fixed.

**Corollary 3.** 
*n–FTFI capacity for feedback code of Definition 3*

*Consider the time-varying AGN channel defined by (Equation 1), driven by a noise with the PO-SS realization of Definition 2, and the code of Definition 3, with initial state $S_{1}=S_{1}^{s}=s$ being fixed.*

*Then, the following hold.*

*(a) The n–FTFI capacity $C_{n}^{fb}\left(\kappa ,s\right)$ is given by*

(141)
$$\begin{array}{cc} C_{n}^{fb}\left(\kappa ,s\right)\overset{▵}{=} & \underset{\frac{1}{n}E_{s}^{\bar{P}}\left\{\sum_{t=1}^{n}{\left(X_{t}\right)}^{2}|S_{1}\right\}\leq \kappa }{\sup}H^{\bar{P}}\left(Y^{n}|s\right)-H\left(V^{n}|s\right). \end{array}$$


(142)
$$\begin{array}{cc} X_{t}= & \Gamma^{0}s+\sum_{j=1}^{t-1}\Gamma_{t,j}^{1}V_{j}+\sum_{j=1}^{t-1}\Gamma_{t,j}^{2}Y_{j}+Z_{t},\;t=1,\ldots ,n. \end{array}$$

*where the supremum is over all $(\Gamma^{0},\Gamma_{t,j}^{1},\Gamma_{t,j}^{2},K_{Z_{t}}),j=1,\ldots ,t-1,t=1,\ldots ,n$ of the realization of $X^{n}$, which induces the distribution ${\bar{P}}_{t}\left(dx_{t}|v^{t-1},y^{t-1},s\right),t=1,\ldots ,n$, and all statements of Theorem 1 and Lemma 1 hold, with the conditional distributions, expectations, and entropies replaced by the corresponding expressions with a fixed $S_{1}=S_{1}^{s}=s$.*

*(b) A necessary condition for Condition 2 of Section 1.1 to hold is as follows:*

*(i) $N_{t}W_{t}$ uniquely defines $C_{t+1}B_{t}W_{t},\forall t$.*

*Moreover, if (i) holds, then the entropy $H(V^{n}|s)$ of part (a) is given by*

(143)
$$\begin{array}{c} H\left(V^{n}|s\right)=\frac{1}{2}\sum_{t=1}^{n}\log\left(2\pi eN_{t}K_{W_{t}}N_{t}^{T}\right). \end{array}$$

*The stable, time-invariant PO-SS realization of Definition 2, which is considered in [4,7], satisfies $W_{t}:\Omega \to R,N_{t}=1,\forall t$, i.e., $n_{w}=1$. Moreover, for this realization, (i) always holds.*



**Proof.** See Section A.3.  □

In the next remark, we illustrate that $H(V^{n}|s)$ given by (Equation 143) follows directly from Lemma 1 by fixing $S_{1}=S_{1}^{s}=s$ and assuming that $N_{t}W_{t}$ uniquely defines $C_{t+1}B_{t}W_{t},\forall t$.

**Remark 7.** 
*The n–FTFI capacity for code of Definition 1 versus code of Definition 3.*

*Consider the generalized Kalman filter of the PO-SS noise realization, of Lemma 1, and assume the following:*

*(i) The initial state of the noise $S_{1}$ is known, i.e., $S_{1}=S_{1}^{s}=s$ or $S_{1}=S_{1}^{s}=s=0$, and $N_{t}W_{t}$ uniquely defines $C_{t+1}B_{t}W_{t},\forall t$.*

*Then, all statements of Lemma 1 hold, by replacing $\left(\Sigma_{t},{\hat{S}}_{t}\right)$ with $\left(\Sigma_{t}^{s},{\hat{S}}_{t}^{s}\right)$ for t=1,2,…. Since $\Sigma_{t}^{s}$ satisfies the generalized DRE (Equation 95) with initial condition $\Sigma_{1}^{s}=0$, then it is easy to deduce that $\Sigma_{t}^{s}=0$ for t=1,2,…,n is a solution. By substituting $\Sigma_{t}^{s}=0,t=1,2,\ldots ,n$ in (Equation 98), we obtain (Equation 143), as expected, which is precisely the entropy of the noise that appeared in [4,7].*

*On the other hand, for the code of Definition 1, by Theorem 1(d), the right-hand side of the n–FTFI capacity $C_{n}^{fb}\left(\kappa \right)$ involves $H(V^{n})$, which is computed using the generalized Kalman filter of Lemma 1.*



### 2.3. A Sufficient Statistic Approach to the Characterization of *n*–FTFI Capacity of AGN Channels Driven by PO-SS Noise Realizations

The characterization of the *n*–FTFI capacity via (Equation 34) (which is equivalently given in Theorem 1(d)), although compactly represented, is not very practical because the input process $X^{n}$ is not expressed in terms of a *sufficient statistic* that summarizes the information of the channel input strategy [39].
In this section, we wish to identify a *sufficient statistic* for the input process $X_{t}$, given by (Equation 68), called the *state of the input*, which summarizes the information contained in $\left(V^{t-1},Y^{t-1}\right)$. It will then become apparent that the characterization of the *n*–FTFI capacity for the Cover and Pombra formulation and code of Definition 1 can be expressed as a functional of *two generalized matrix DREs*.

First, we invoke Theorem 1 and Lemma 1 to show that for each time *t*, $X_{t}$ is expressed as(144)$$\begin{array}{cc} \; & X_{t}=\Lambda_{t}\left({\hat{S}}_{t}-E\left\{{\hat{S}}_{t}|Y^{t-1}\right\}\right)+Z_{t},\;t=1,\ldots ,n, \end{array}$$(145)$$\begin{array}{cc} \; & {\hat{S}}_{t}\overset{▵}{=}E\left\{S_{t}|V^{t-1}\right\},\;{\hat{\hat{S}}}_{t}\overset{▵}{=}E\left\{{\hat{S}}_{t}|Y^{t-1}\right\} \end{array}$$
which means, at each time *t*, the state of the channel input process $X_{t}$ is $\left({\hat{S}}_{t},{\hat{\hat{S}}}_{t}\right)$. We show that ${\hat{\hat{S}}}_{t}$ satisfies another generalized Kalman filter recursion.

Now, we prepare to prove (Equation 144) and the main theorem. We start with preliminary calculations.(146)$$\begin{array}{cc} P\left\{Y_{t}\in dy|Y^{t-1},X^{t}\right\}= & P_{t}\left(dy|X_{t},V^{t-1}\right),\;\;t=2,\ldots ,n,\;\text{by channel definition} \end{array}$$(147)$$\begin{array}{cc} = & P_{t}\left(dy|X_{t},V^{t-1},{\hat{S}}^{t}\right),\;\mathrm{by}\;{\hat{S}}_{t}=E\left\{S_{t}|V^{t-1}\right\} \end{array}$$(148)$$\begin{array}{cc} = & P_{t}\left(dy|X_{t},V^{t-1},{\hat{S}}_{t},{\hat{I}}^{t-1}\right),\;\mathrm{by}\;\text{(90)},\;i.e.,V_{t}=C_{t}{\hat{S}}_{t}+{\hat{I}}_{t} \end{array}$$(149)$$\begin{array}{cc} = & P_{t}\left(dy|X_{t},{\hat{S}}_{t}\right),\;\mathrm{by}\;Y_{t}=X_{t}+V_{t}=X_{t}+C_{t}{\hat{S}}_{t}+{\hat{I}}_{t},\text{(91)}. \end{array}$$
At t=1, we also have $P\left\{Y_{1}\in dy|X_{1}\right\}=P_{1}\left(dy|X_{1}\right)$. By (Equation 149), it follows that the conditional distribution of $Y_{t}$ given $Y^{t-1}=y^{t-1}$ is(150)$$\begin{array}{cc} P_{t}\left(dy_{t}|y^{t-1}\right)= & \int P_{t}\left(dy|x_{t},{\hat{s}}_{t}\right)P_{t}\left(dx_{t}|{\hat{s}}_{t},y^{t-1}\right)P_{t}\left(d{\hat{s}}_{t}|y^{t-1}\right),\;t=2,\ldots ,n, \end{array}$$(151)$$\begin{array}{cc} P_{1}\left(dy_{1}\right)= & \int P_{1}\left(dy|x_{t},{\hat{s}}_{1}\right)P_{1}\left(dx_{1}|{\hat{s}}_{1}\right)P_{1}\left(d{\hat{s}}_{1}\right). \end{array}$$
From the above distributions, at each time *t*, the distribution of $X_{t}$ conditioned on $\left(V^{t-1},Y^{t-1}\right)$, given in Theorem 1, is also expressed as a linear functional of $\left({\hat{S}}_{t},Y^{t-1}\right)$, for t=1,…,n.

The next theorem further shows that for each *t*, the dependence of $X_{t}$ on $Y^{t-1}$ is expressed in terms of $E\left\{{\hat{S}}_{t}|Y^{t-1}\right\}$ for t=1,…,n, and this dependence gives rise to an equivalent sequential characterization of the Cover and Pombra *n*–FTFI capacity, $C_{n}^{fb}\left(\kappa \right)$.

**Theorem 3.** 
*Equivalent characterization of n–FTFI capacity $C_{n}^{fb}\left(\kappa \right)$ for PO-SS noise realizations*

*Consider the time-varying AGN channel defined by (Equation 1), driven by a noise with the PO-SS realization of Definition 2, and the code of Definition 1. Also consider the generalized Kalman filter of Lemma 1.*

*Define the conditional covariance and conditional mean of ${\hat{S}}_{t}$, given $Y^{t-1}$, by*

(152)
$$\begin{array}{cc} K_{t}\overset{▵}{=} & cov\left({\hat{S}}_{t},{\hat{S}}_{t}|Y^{t-1}\right)=E\left\{\left({\hat{S}}_{t}-{\hat{\hat{S}}}_{t}\right){\left({\hat{S}}_{t}-{\hat{\hat{S}}}_{t}\right)}^{T}|Y^{t-1}\right\},\;\;t=2,\ldots ,n, \end{array}$$


(153)
$$\begin{array}{cc} {\hat{\hat{S}}}_{t}\overset{▵}{=} & E\left\{{\hat{S}}_{t}|Y^{t-1}\right\},\;{\hat{\hat{S}}}_{1}\overset{▵}{=}\mu_{S_{1}},\;K_{1}\overset{▵}{=}0. \end{array}$$

*Then, the following hold.*

*(a) An equivalent characterization of the n–FTFI capacity $C_{n}^{fb}\left(\kappa \right)$, defined by (Equation 34) and (Equation 35), is*

(154)
$$\begin{array}{c} C_{n}^{fb}\left(\kappa \right)=\underset{P_{\left[1,n\right]}^{\hat{S}}\left(\kappa \right)}{\sup}\sum_{t=1}^{n}H\left(Y_{t}|Y^{t-1}\right)-H\left(V^{n}\right) \end{array}$$

*where $\left(X^{n},Y^{n}\right)$ is jointly Gaussian, and*

(155)
$$\begin{array}{cc} \; & H\left(V^{n}\right)\;is\;the\;entropy\;of\;V^{n}\;given\;in\;Lemma\;1,i.e.,\text{(98)}, \end{array}$$


(156)
$$\begin{array}{cc} \; & {\hat{I}}^{n}\;is\;the\;innovations\;process\;of\;V^{n}\;given\;in\;Lemma\;1; \end{array}$$


(157)
$$\begin{array}{cc} \; & Y_{t}=X_{t}+V_{t},\;t=1,\ldots ,n; \end{array}$$


(158)
$$\begin{array}{cc} \; & V_{t}=C_{t}{\hat{S}}_{t}+{\hat{I}}_{t}; \end{array}$$


(159)
$$\begin{array}{cc} \; & P_{t}\left(dy_{t}|y^{t-1}\right)=\int P_{t}\left(dy|x_{t},{\hat{s}}_{t}\right)P_{t}\left(dx_{t}|{\hat{s}}_{t},y^{t-1}\right)P_{t}\left(d{\hat{s}}_{t}|y^{t-1}\right),\;t=2,\ldots ,n; \end{array}$$


(160)
$$\begin{array}{cc} \; & P_{1}\left(dy_{1}\right)=\int P_{1}\left(dy|x_{t},{\hat{s}}_{1}\right)P_{1}\left(dx_{1}|{\hat{s}}_{1}\right)P_{1}\left(d{\hat{s}}_{1}\right); \end{array}$$


(161)
$$\begin{array}{cc} \; & P_{t}\left(dy_{t}|y^{t-1}\right)\in N\left(\mu_{Y_{t}|Y^{t-1}},K_{Y_{t}|Y^{t-1}}\right); \end{array}$$


(162)
$$\begin{array}{cc} \; & \mu_{Y_{t}|Y_{t-1}}\;is\;linear\;in\;Y^{t-1}\;and\;K_{Y_{t}|Y^{t-1}}\;is\;nonrandom; \end{array}$$


(163)
$$\begin{array}{cc} \; & P_{t}\left(dx_{t}|{\hat{s}}_{t},y^{t-1}\right)\in N\left(\mu_{X_{t}|{\hat{S}}_{t},Y^{t-1}},K_{X_{t}|{\hat{S}}_{t},Y^{t-1}}\right); \end{array}$$


(164)
$$\begin{array}{cc} \; & \mu_{X_{t}|{\hat{S}}_{t},Y^{t-1}}\;is\;linear\;in\;\left({\hat{S}}_{t},Y^{t-1}\right)\;and\;K_{X_{t}|{\hat{S}}_{t},Y^{t-1}}is\;nonrandom; \end{array}$$


(165)
$$\begin{array}{cc} \; & P_{\left[1,n\right]}^{\hat{S}}\left(\kappa \right)\overset{▵}{=}\left\{P_{t}\left(dx_{t}|{\hat{s}}_{t},y^{t-1}\right),t=1,\ldots ,n:\frac{1}{n}E\left(\sum_{t=1}^{n}{\left(X_{t}\right)}^{2}\right)\leq \kappa \right\}. \end{array}$$


*(b) The optimal jointly Gaussian process $\left(X^{n},Y^{n}\right)$ of part (a) is represented as a function of a sufficient statistic by*

(166)
$$\begin{array}{cc} \; & X_{t}=\Lambda_{t}\left({\hat{S}}_{t}-{\hat{\hat{S}}}_{t}\right)+Z_{t},\;t=1,\ldots ,n, \end{array}$$


(167)
$$\begin{array}{cc} \; & Z_{t}\in N\left(0,K_{Z_{t}}\right)\;independent\;of\;\left(X^{t-1},V^{t-1},{\hat{S}}^{t},\hat{{\hat{S}}^{t}},{\hat{I}}^{t},Y^{t-1}\right),\;t=1,\ldots ,n; \end{array}$$


(168)
$$\begin{array}{cc} \; & {\hat{I}}_{t}\in N\left(0,K_{{\hat{I}}_{t}}\right)\;independent\;of\;\left(X^{t-1},V^{t-1},{\hat{S}}^{t},Y^{t-1},\hat{{\hat{S}}^{t}}\right),\;t=1,\ldots ,n; \end{array}$$


(169)
$$\begin{array}{cc} \; & Y_{t}=\Lambda_{t}\left({\hat{S}}_{t}-{\hat{\hat{S}}}_{t}\right)+Z_{t}+V_{t},\;t=1,\ldots ,n; \end{array}$$


(170)
$$\begin{array}{cc} \; & \;\;=\Lambda_{t}\left({\hat{S}}_{t}-{\hat{\hat{S}}}_{t}\right)+C_{t}{\hat{S}}_{t}+{\hat{I}}_{t}+Z_{t}; \end{array}$$


(171)
$$\begin{array}{cc} \; & \frac{1}{n}E\left\{\sum_{t=1}^{n}{\left(X_{t}\right)}^{2}\right\}=\frac{1}{n}\sum_{t=1}^{n}\left(\Lambda_{t}K_{t}\Lambda_{t}^{T}+K_{Z_{t}}\right). \end{array}$$

*where Λt is nonrandom.*

*The conditional mean and covariance, ${\hat{\hat{S}}}_{t}$ and $K_{t}$, are given by generalized Kalman filter equations, as follows:*

*(i) ${\hat{\hat{S}}}_{t}$ satisfies the Kalman filter recursion*

(172)
$$\begin{array}{cc} \; & {\hat{\hat{S}}}_{t+1}=A_{t}{\hat{\hat{S}}}_{t}+F_{t}\left(\Sigma_{t},K_{t}\right)I_{t},\;{\hat{\hat{S}}}_{1}=\mu_{S_{1}}; \end{array}$$


(173)
$$\begin{array}{cc} \; & F_{t}\left(\Sigma_{t},K_{t}\right)\overset{▵}{=}\left(A_{t}K_{t}{\left(\Lambda_{t}+C_{t}\right)}^{T}+M_{t}\left(\Sigma_{t}\right)K_{{\hat{I}}_{t}}\right){\left\{K_{{\hat{I}}_{t}}+K_{Z_{t}}+\left(\Lambda_{t}+C_{t}\right)K_{t}{\left(\Lambda_{t}+C_{t}\right)}^{T}\right\}}^{-1}; \end{array}$$


(174)
$$\begin{array}{cc} \; & I_{t}\overset{▵}{=}Y_{t}-E\left\{Y_{t}|Y^{t-1}\right\}=Y_{t}-C_{t}{\hat{\hat{S}}}_{t}=\left(\Lambda_{t}+C_{t}\right)\left({\hat{S}}_{t}-{\hat{\hat{S}}}_{t}\right)+{\hat{I}}_{t}+Z_{t},\;t=1,\ldots ,n; \end{array}$$


(175)
$$\begin{array}{cc} \; & I_{t}\in N\left(0,K_{I_{t}}\right),\;t=1,\ldots ,n\;is\;an\;orthogonal\;innovations\;process,\\ \; & \;\;i.e.,\;I_{t}\;is\;independent\;of\;I_{s},\;for\;all\;t\neq s,\;and\;I_{t}\;is\;independent\;of\;V^{t-1}; \end{array}$$


(176)
$$\begin{array}{cc} \; & K_{Y_{t}|Y^{t-1}}=K_{I_{t}}\overset{▵}{=}cov\left(I_{t},I_{t}\right)=\left(\Lambda_{t}+C_{t}\right)K_{t}{\left(\Lambda_{t}+C_{t}\right)}^{T}+K_{{\hat{I}}_{t}}+K_{Z_{t}}; \end{array}$$


(177)
$$\begin{array}{cc} \; & K_{{\hat{I}}_{t}}\;given\;by\;\text{(92)}. \end{array}$$


*(ii) The error ${\hat{E}}_{t}\overset{▵}{=}{\hat{S}}_{t}-{\hat{\hat{S}}}_{t}$ satisfies the recursion*

$$\begin{array}{cc} {\hat{E}}_{t+1}= & F_{t}^{CL}\left(\Sigma_{t},K_{t}\right){\hat{E}}_{t}+\left(M_{t}\left(\Sigma_{t}\right)-F_{t}\left(\Sigma_{t},K_{t}\right)\right){\hat{I}}_{t} \end{array}$$


(178)
$$\begin{array}{cc} \; & -F_{t}\left(\Sigma_{t}.K_{t}\right)Z_{t},\;{\hat{E}}_{1}={\hat{S}}_{1}-{\hat{\hat{S}}}_{1}=0,\;t=1,\ldots ,n; \end{array}$$


(179)
$$\begin{array}{cc} F_{t}^{CL}(\Sigma_{t}, & K_{t})\overset{▵}{=}A_{t}-F_{t}\left(\Sigma_{t},K_{t}\right)\left(\Lambda_{t}+C_{t}\right). \end{array}$$

*(iii) $K_{t}=E\left\{{\hat{E}}_{t}{\hat{E}}_{t}^{T}\right\}$ satisfies the generalized DRE*

(180)
$$\begin{array}{cc} \; & K_{t+1}=A_{t}K_{t}A_{t}^{T}-\left(A_{t}K_{t}{\left(\Lambda_{t}+C_{t}\right)}^{T}+M_{t}\left(\Sigma_{t}\right)K_{{\hat{I}}_{t}}\right)(K_{{\hat{I}}_{t}}+K_{Z_{t}}\\ \; & +\left(\Lambda_{t}+C_{t}\right)K_{t}{\left(\Lambda_{t}+C_{t}\right)}^{T}{)}^{-1}{\left(A_{t}K_{t}{\left(\Lambda_{t}+C_{t}\right)}^{T}+M_{t}\left(\Sigma_{t}\right)K_{{\hat{I}}_{t}}\right)}^{T}\\ \; & +M_{t}\left(\Sigma_{t}\right)K_{{\hat{I}}_{t}}{\left(M_{t}\left(\Sigma_{t}\right)\right)}^{T},\;K_{t}⪰0,\;t=1,\ldots ,n,\;K_{1}=0. \end{array}$$

*(c) An equivalent characterization of the n–FTFI capacity $C_{n}^{fb}\left(\kappa \right)$, defined by (Equation 33) and (Equation 34), using the sufficient statistics of part (b) is*

(181)
$$\begin{array}{cc} C_{n}^{fb}\left(\kappa \right)= & \underset{\left(\Lambda_{t},K_{Z_{t}}\right),t=1,\ldots ,n:\;\frac{1}{n}E\left\{\sum_{t=1}^{n}{\left(X_{t}\right)}^{2}\right\}\leq \kappa }{\sup}\frac{1}{2}\sum_{t=1}^{n}\log\frac{K_{Y_{t}|Y^{t-1}}}{K_{V_{t}|V^{t-1}}} \end{array}$$


(182)
$$\begin{array}{cc} = & \underset{\left(\Lambda_{t},K_{Z_{t}}\right),t=1,\ldots ,n:\;\frac{1}{n}E\left\{\sum_{t=1}^{n}{\left(X_{t}\right)}^{2}\right\}\leq \kappa }{\sup}\frac{1}{2}\sum_{t=1}^{n}\log\frac{K_{I_{t}}}{K_{{\hat{I}}_{t}}}\\ = & \underset{\left(\Lambda_{t},K_{Z_{t}}\right),t=1,\ldots ,n:\;\frac{1}{n}\sum_{t=1}^{n}\left(\Lambda_{t}K_{t}\Lambda_{t}^{T}+K_{Z_{t}}\right)\leq \kappa }{\sup}\frac{1}{2}\{ \end{array}$$


(183)
$$\begin{array}{cc} \; & \sum_{t=1}^{n}\log\left(\frac{\left(\Lambda_{t}+C_{t}\right)K_{t}{\left(\Lambda_{t}+C_{t}\right)}^{T}+K_{{\hat{I}}_{t}}+K_{Z_{t}}}{K_{{\hat{I}}_{t}}}\right)\}. \end{array}$$




**Proof.** See Section A.4.  □

**Remark 8.** 
*On the characterization of n–FTFI capacity of Theorem 3*

*The characterization of the n–FTFI capacity $C_{n}^{fb}\left(\kappa \right)$ given by (Equation 183) involves the generalized matrix DRE $K_{t}$, which is also a functional of the generalized matrix DRE $\Sigma_{t}$ of the error covariance of the state $S^{n}$ from the noise output $V^{n}$. This feature is not part of the analysis in [4] and recent studies [4,7,9,11,12].*



The next corollary follows directly from Theorem 3 as a degenerate case.

**Corollary 4.** 
*Equivalent characterization of n–FTFI capacity $C_{n}^{fb}\left(\kappa ,s\right)$ for PO-SS noise realizations Consider the time-varying AGN channel defined by (Equation 1), driven by a noise with the PO-SS realization of Definition 2, and the code of Definition 3, with initial state $S_{1}=S_{1}^{s}=s$ fixed, and replace (Equation 152) and (Equation 153) by*

(184)
$$\begin{array}{cc} K_{t}=K_{t}^{s}\overset{▵}{=} & cov\left({\hat{S}}_{t}^{s},{\hat{S}}_{t}^{s}|Y^{t-1},S_{1}=s\right)=E\left\{\left({\hat{S}}_{t}^{s}-\hat{{\hat{S}}_{t}^{s}}\right){\left({\hat{S}}_{t}^{s}-\hat{{\hat{S}}_{t}^{s}}\right)}^{T}\right\}, \end{array}$$


(185)
$$\begin{array}{cc} {\hat{\hat{S}}}_{t}= & \hat{{\hat{S}}_{t}^{s}}\overset{▵}{=}E\left\{{\hat{S}}_{t}^{s}|Y^{t-1},S_{1}=s\right\},\;t=2,\ldots ,n,\;\hat{{\hat{S}}_{1}}=\hat{{\hat{S}}_{1}^{s}}\overset{▵}{=}s,\;K_{1}=K_{1}^{s}=0. \end{array}$$

*Then, the characterization of n–FTFI capacity, (Equation 3), is*

(186)
$$\begin{array}{cc} \; & C_{n}^{fb}\left(\kappa ,s\right)=\underset{P_{\left[1,n\right]}^{\hat{{\hat{S}}^{s}}}\left(\kappa \right)}{\sup}\sum_{t=1}^{n}H\left(Y_{t}|Y^{t-1},s\right)-H\left(V^{n}|s\right), \end{array}$$


(187)
$$\begin{array}{cc} \; & P_{\left[1,n\right]}^{\hat{{\hat{S}}^{s}}}\left(\kappa \right)\overset{▵}{=}\left\{P_{t}\left(dx_{t}|\hat{{\hat{s}}_{t}^{s}},y^{t-1},s\right),t=1,\ldots ,n:\frac{1}{n}E\left(\sum_{t=1}^{n}{\left(X_{t}\right)}^{2}|S_{1}^{s}=s\right)\leq \kappa \right\} \end{array}$$

*where $H(V^{n}|s)$ is given by Corollary 1, and the statements of Theorem 3 hold with the above changes, i.e., (Equation 184), (Equation 185), and all conditional entropies, distributions, expectations, etc., are defined for fixed $S_{1}=S_{1}^{s}=s$.*


**Proof.** It is easily verified from the derivation of Theorem 3 by fixing $S_{1}=S_{1}^{s}=s$.  □

**Remark 9.** 
*On the characterization of n–FTFI capacity of Corollary 4*

*The characterization of the n–FTFI capacity $C_{n}^{fb}\left(\kappa ,s\right)$ given in Corollary 4 (similar to Theorem 3) involves two generalized matrix DREs, because it does not assume that Conditions 1 and 2 hold. This distinction is not part of the analysis in [4,7,9,11,12].*



### 2.4. Application Examples

In this section, we apply Theorem 3 to specific examples.

First, we consider the application example of the AGN channel driven by the PO-SS$\left(a_{t},c_{t},b_{t}^{1},b_{t}^{2},d_{t}^{1},d_{t}^{2}\right)$ noise.

**Corollary 5.** 
*The n–FTFI capacity $C_{n}^{fb}\left(\kappa \right)$ of the AGN channel driven by the PO-SS$(a_{t},c_{t},b_{t}^{1},b_{t}^{2}$, $d_{t}^{1},d_{t}^{2})$ noise is obtained from Lemma 1 and Theorem 3 by using (Equation 113).*


**Proof.** This is easily verified, as in Corollary 2.  □

In the next corollary, we apply Theorem 3 to the stable and unstable ARMA(a,c) noise to obtain the characterization of the *n*–FTFI capacity $C_{n}^{fb}\left(\kappa \right)$ and $C_{n}^{fb}\left(\kappa ,s\right)$. It is then obvious that for the stable ARMA(a,c),a∈[−1,1],c∈(−1,1) noise, the characterization of $C_{n}^{fb}\left(\kappa \right)$ involves two generalized DREs, contrary to the analysis in [4,7,9,11,12], for the same noise model.

**Corollary 6.** 
*Characterization of n–FTFI capacity $C_{n}^{fb}\left(\kappa \right)$ for the ARMA(a,c),a∈(−∞,∞),c∈(−∞,∞)*

*Consider the time-varying AGN channel defined by (Equation 1) and the code of Definition 1.*

*(a) For the non-stationary ARMA(a,c),a∈(−∞,∞),c∈(−∞,∞) noise of Example 2(a), the characterization of the n–FTFI capacity, $C_{n}^{fb}\left(\kappa \right)$, is*

(188)
$$\begin{array}{cc} \; & C_{n}^{fb}\left(\kappa \right)=\underset{\left(\Lambda_{t},K_{Z_{t}}\right),t=1,\ldots ,n:\;\frac{1}{n}\sum_{t=1}^{n}\left({\left(\Lambda_{t}\right)}^{2}K_{t}+K_{Z_{t}}\right)\leq \kappa }{\sup}\frac{1}{2}\sum_{t=1}^{n}\log\left(\frac{{\left(\Lambda_{t}+c-a\right)}^{2}K_{t}+K_{{\hat{I}}_{t}}+K_{Z_{t}}}{K_{{\hat{I}}_{t}}}\right) \end{array}$$

*subject to the constraints*

$$\begin{array}{cc} K_{t+1}= & {\left(c\right)}^{2}K_{t}+{\left(M_{t}\left(\Sigma_{t}\right)\right)}^{2}K_{{\hat{I}}_{t}}-{\left(cK_{t}\left(\Lambda_{t}+c-a\right)+M_{t}\left(\Sigma_{t}\right)K_{{\hat{I}}_{t}}\right)}^{2} \end{array}$$


(189)
$$\begin{array}{cc} \; & \;.{\left(K_{{\hat{I}}_{t}}+K_{Z_{t}}+{\left(\Lambda_{t}+c-a\right)}^{2}K_{t}\right)}^{-1},\;K_{1}=0,\;t=1,\ldots ,n, \end{array}$$


(190)
$$\begin{array}{cc} K_{Z_{t}}\geq & 0,\;K_{t}\geq 0,\;c\neq a,\;K_{W}>0,\;t=1,\ldots ,n \end{array}$$

*and where*

(191)
$$\begin{array}{cc} \; & M_{t}\left(\Sigma_{t}\right)\overset{▵}{=}\left(c\Sigma_{t}\left(c-a\right)+K_{W}\right){\left(K_{W}+{\left(c-a\right)}^{2}\Sigma_{t}\right)}^{-1}, \end{array}$$


(192)
$$\begin{array}{cc} \; & K_{{\hat{I}}_{t}}={\left(c-a\right)}^{2}\Sigma_{t}+K_{W},\;t=1,\ldots ,n, \end{array}$$


(193)
$$\begin{array}{cc} \; & \Sigma_{t+1}={\left(c\right)}^{2}\Sigma_{t}+K_{W}-{\left(c\Sigma_{t}\left(c-a\right)+K_{W}\right)}^{2}{\left(K_{W}+{\left(c-a\right)}^{2}\Sigma_{t}\right)}^{-1},\;t=1,\ldots ,n, \end{array}$$


(194)
$$\begin{array}{cc} \; & \Sigma_{1}=K_{S_{1}}=\frac{{\left(c_{0}\right)}^{2}K_{S_{0}}+{\left(a_{0}\right)}^{2}K_{W_{0}}}{{\left(c_{0}-a_{0}\right)}^{2}}. \end{array}$$

*The optimal jointly Gaussian process $\left(X^{n},Y^{n}\right)$ is obtained from Theorem 3(b) by invoking*

(195)
$$\begin{array}{c} A_{t}⟼c,\;C_{t}⟼c-a,\;B_{t}⟼1,\;N_{t}⟼1,\;t=1,2,\ldots ,n. \end{array}$$

*Special Case. If $\Sigma_{1}=0$ or the initial state is fixed, $S_{1}=S_{1}^{s}=s$, then*

(196)
$$\begin{array}{c} \Sigma_{t}=\Sigma_{t}^{s}=0,\;K_{{\hat{I}}_{t}}=K_{W},\;M_{t}\left(\Sigma_{t}\right)=M_{t}\left(\Sigma_{t}^{s}\right)=1,\;t=1,2,\ldots \end{array}$$

*and $C_{n}^{fb}\left(\kappa \right)$ reduces to*

(197)
$$\begin{array}{cc} C_{n}^{fb}\left(\kappa \right)= & C_{n}^{fb}\left(\kappa ,s\right)=\underset{\left(\Lambda_{t},K_{Z_{t}}\right),t=1,\ldots ,n:\;\frac{1}{n}\sum_{t=1}^{n}\left({\left(\Lambda_{t}\right)}^{2}K_{t}^{s}+K_{Z_{t}}\right)\leq \kappa }{\sup}\{\\ \; & \frac{1}{2}\sum_{t=1}^{n}\log\left(\frac{{\left(\Lambda_{t}+c-a\right)}^{2}K_{t}^{s}+K_{W}+K_{Z_{t}}}{K_{W}}\right)\} \end{array}$$

*subject to the constraints*

(198)
$$\begin{array}{cc} \; & K_{t+1}^{s}={\left(c\right)}^{2}K_{t}^{s}+K_{W}-{\left(cK_{t}^{s}\left(\Lambda_{t}+c-a\right)+K_{W}\right)}^{2}\\ \; & .{\left(K_{Z_{t}}+{\left(\Lambda_{t}+c-a\right)}^{2}K_{t}^{s}+K_{W}\right)}^{-1},\;K_{1}^{s}=0,\;\;K_{t}^{s}\geq 0,\;K_{Z_{t}}\geq 0,\;t=1,\ldots ,n. \end{array}$$

*(This special case is precisely the application example analyzed in [4,7,8]).*


*(b) For the non-stationary AR(c),c∈(−∞,∞) noise of Example 2(c), the characterization of the n–FTFI capacity $C_{n}^{fb}\left(\kappa \right)$ is obtained from part (a) by setting a=0, i.e.,*

(199)
$$\begin{array}{cc} \; & C_{n}^{fb}\left(\kappa \right)=\underset{\left(\Lambda_{t},K_{Z_{t}}\right),t=1,\ldots ,n:\;\frac{1}{n}\sum_{t=1}^{n}\left({\left(\Lambda_{t}\right)}^{2}K_{t}+K_{Z_{t}}\right)\leq \kappa }{\sup}\frac{1}{2}\sum_{t=1}^{n}\log\left(\frac{{\left(\Lambda_{t}+c\right)}^{2}K_{t}+{\left(c\right)}^{2}\Sigma_{t}+K_{W}+K_{Z_{t}}}{{\left(c\right)}^{2}\Sigma_{t}+K_{W}}\right) \end{array}$$

*subject to the constraints of $K_{t},\Sigma_{t}$ are the non-negative solutions of the generalized RDEs:*

$$\begin{array}{cc} K_{t+1}= & {\left(c\right)}^{2}K_{t}+{\left(c\right)}^{2}\Sigma_{t}+K_{W}-{\left(cK_{t}\left(\Lambda_{t}+c\right)+{\left(c\right)}^{2}\Sigma_{t}+K_{W}\right)}^{2} \end{array}$$


(200)
$$\begin{array}{cc} \; & .{\left({\left(c\right)}^{2}\Sigma_{t}+K_{W}+K_{Z_{t}}+{\left(\Lambda_{t}+c\right)}^{2}K_{t}\right)}^{-1},\;K_{1}=0,\;t=1,\ldots ,n, \end{array}$$


(201)
$$\begin{array}{cc} \Sigma_{t+1}= & {\left(c\right)}^{2}\Sigma_{t}+K_{W}-{\left({\left(c\right)}^{2}\Sigma_{t}+K_{W}\right)}^{2}{\left(K_{W}+{\left(c\right)}^{2}\Sigma_{t}\right)}^{-1},\;\Sigma_{1}=K_{S_{1}}=K_{S_{0}}\geq 0. \end{array}$$

*(c) For the non-stationary AR(c),c∈(−∞,∞) noise of Example 2(c), with $\Sigma_{1}=0$ or a fixed initial state $S_{1}=S_{1}^{s}=s$, then (Equation 196) holds, i.e., $\Sigma_{t}=\Sigma_{t}^{s}=0,K_{{\hat{I}}_{t}}=K_{W},M_{t}\left(\Sigma_{t}\right)=M_{t}\left(\Sigma_{t}^{s}\right)=1,t=1,2,\ldots$, and $C_{n}^{fb}\left(\kappa \right)$ reduces to*

(202)
$$\begin{array}{cc} \; & C_{n}^{fb}\left(\kappa \right)=C_{n}^{fb}\left(\kappa ,s\right)\\ \; & =\underset{\left(\Lambda_{t},K_{Z_{t}}\right),t=1,\ldots ,n:\;\frac{1}{n}\sum_{t=1}^{n}\left({\left(\Lambda_{t}\right)}^{2}K_{t}^{s}+K_{Z_{t}}\right)\leq \kappa }{\sup}\frac{1}{2}\sum_{t=1}^{n}\log\left(\frac{{\left(\Lambda_{t}+c\right)}^{2}K_{t}^{s}+K_{W}+K_{Z_{t}}}{K_{W}}\right) \end{array}$$

*subject to the constraint*

(203)
$$\begin{array}{cc} \; & K_{t+1}^{s}={\left(c\right)}^{2}K_{t}^{s}+K_{W}\\ \; & -{\left(cK_{t}^{s}\left(\Lambda_{t}+c\right)+K_{W}\right)}^{2}{\left(K_{W}+K_{Z_{t}}+{\left(\Lambda_{t}+c\right)}^{2}K_{t}^{s}\right)}^{-1},\;K_{1}^{s}=0,\;t=1,\ldots ,n. \end{array}$$



**Proof.** (a) The first part follows directly from Theorem 3 by using (Equation 195). The last part is obtained as follows. If $\Sigma_{1}=0$ or $S_{1}=S_{1}^{s}=s$ is fixed, then, by (Equation 193), it follows that $\Sigma_{t}=\Sigma_{t}^{s}=0,\forall t=2,\ldots$. Moreover, by (Equation 191) and (Equation 192), it follows that $M_{t}\left(\Sigma_{t}\right)=M_{t}\left(\Sigma_{t}^{s}\right)=1,K_{{\hat{I}}_{t}}={\left(c-a\right)}^{2}\Sigma_{t}^{s}+K_{W}=K_{W},t=1,2,\ldots$. By substituting into (Equation 188) and (Equation 189), we obtain (Equation 197) and (Equation 180). (b) From part (a), let a=0. Then,(204)$$\begin{array}{cc} \; & M_{t}\left(\Sigma_{t}\right)=\left({\left(c\right)}^{2}\Sigma_{t}+K_{W}\right){\left(K_{W}+{\left(c\right)}^{2}\Sigma_{t}\right)}^{-1},\;K_{{\hat{I}}_{t}}={\left(c\right)}^{2}\Sigma_{t}+K_{W},\;t=1,\ldots ,n. \end{array}$$
By substituting into the equations of part (a), we obtain (Equation 200) and (Equation 201). (c) This is a special case of parts (a) and (b).  □

**Remark 10.** 
*By Corollary 6(a), it is obvious that if $\Sigma_{1}=0$, i.e., $K_{S_{0}}=K_{W_{0}}=0$, which means $S_{1}=S_{1}^{s}=s$ is fixed, $\left(V_{0},W_{0}\right)=\left(v_{0},w_{0}\right)$ is fixed (and known to the encoder and the decoder)—see (Equation 120). Then, $\Sigma_{1}=\Sigma_{1}^{s}=0$ and $C_{n}^{fb}\left(\kappa \right)=C_{n}^{fb}\left(\kappa ,s\right)$, which depends on the initial state $S_{1}=S_{1}^{s}=s$. To ensure that we obtain a large enough n, the rate $\frac{1}{n}C_{n}\left(\kappa ,s\right)$ is independent of s. Moreover, it is necessary to identify conditions for convergence of solutions $K_{t}^{s},t=1,2,\ldots$ of the generalized DRE (Equation 180) to a unique limit, $\lim_{n⟶\infty }K_{n}^{s}=K^{\infty }\geq 0$, which does not depend on the initial data $K_{1}^{s}=0$. We address this problem in Section 3.*


### 2.5. Case (II) Formulation: A Degenerate of Case (I) Formulation

Theorem 3 gives the *n*–FTFI capacity for the Case (I) formulation. However, since the Case (II) formulation is a special case of the Case (I) formulation, we expect that we can recover the characterization of the *n*–FTFI capacity for the Case (II) formulation from Theorem 3, i.e., when the code is $\left(s,2^{nR},n\right)$, n=1,2,…, and Conditions 1 and 2 of Section 1.1 hold. We show this in the next corollary.

**Corollary 7.** 
*The degenerate n–FTFI capacity $C_{n}^{fb}\left(\kappa \right)$ of Theorem 3 for the Case (II) formulation*

*Consider the time-varying AGN channel defined by (Equation 1), driven by a noise with PO-SS realization of Definition 2, and suppose that the following hold:*

*(1) The code is $\left(s,2^{nR},n\right)$, n=1,2,…;*

*(2) Conditions 1 and 2 of Section 1.1 hold.*




*Then, the following hold:*

*(a) Corollary 1 holds, i.e., all statements of Lemma 1 hold with $\left(\Sigma_{t},{\hat{S}}_{t}\right)$ replaced by $\left(\Sigma_{t}^{s},{\hat{S}}_{t}^{s}\right)$ as defined by (Equation 102) and (Equation 103). In particular, $\left(\Sigma_{t}^{s},{\hat{S}}_{t}^{s}\right)=\left(0,S_{t}^{s}\right)$ for t=1,2,…, and $H\left(V^{n}\right)=H\left(V^{n}|s\right)$ is given by (Equation 143).*

*(b) All statements of Theorem 3 hold with $\left(\Sigma_{t},{\hat{S}}_{t}\right)$ replaced by $\left(\Sigma_{t}^{s},{\hat{S}}_{t}^{s}\right)$, as in part (a), and $\left(K_{t},{\hat{\hat{S}}}_{t}\right)$ defined by (Equation 152) and (Equation 153) reduces to*

(205)
$$\begin{array}{ccc} & & K_{t}=K_{t}^{s}=cov\left(S_{t}^{s},S_{t}^{s}|Y^{t-1},S_{1}^{s}=s\right),\; \end{array}$$


(206)
$$\begin{array}{ccc} & & {\hat{\hat{S}}}_{t}={\hat{S}}_{t}^{s}=E\left\{S_{t}^{s}|Y^{t-1},S_{1}^{s}=s\right\},\;K_{1}^{s}=0,{\hat{S}}_{1}^{s}=s,\;t=2,\ldots ,n \end{array}$$

*In particular, the optimal input process $X^{n}$ of Theorem 3(c) degenerates to*

(207)
$$\begin{array}{cc} \; & X_{t}=\Lambda_{t}\left(S_{t}^{s}-{\hat{S}}_{t}^{s}\right)+Z_{t},\;X_{1}=Z_{t},\;t=2,\ldots ,n. \end{array}$$

*(c) The characterization of the n–FTFI capacity, $C_{n}^{fb}\left(\kappa \right)$ of Theorem 3, degenerates to $C_{n}^{fb,S}\left(\kappa ,s\right)$. It is defined by*

(208)
$$\begin{array}{cc} C_{n}^{fb}\left(\kappa \right)= & C_{n}^{fb,S}\left(\kappa ,s\right)\overset{▵}{=}\underset{\left(\Lambda_{t},K_{Z_{t}}\right),t=1,\ldots ,n:\;\frac{1}{n}E_{s}\left\{\sum_{t=1}^{n}{\left(X_{t}\right)}^{2}\right\}\leq \kappa }{\sup}\sum_{t=1}^{n}\log\frac{K_{Y_{t}|Y^{t-1},s}}{K_{V_{t}|V^{t-1},s}} \end{array}$$


(209)
$$\begin{array}{cc} = & \underset{\left(\Lambda_{t},K_{Z_{t}}\right),t=1,\ldots ,n:\;\frac{1}{n}\sum_{t=1}^{n}\left(\Lambda_{t}K_{t}^{s}\Lambda_{t}^{T}+K_{Z_{t}}\right)\leq \kappa }{\sup}\{\\ \; & \frac{1}{2}\sum_{t=1}^{n}\log\left(\frac{\left(\Lambda_{t}+C_{t}\right)K_{t}^{s}{\left(\Lambda_{t}+C_{t}\right)}^{T}+N_{t}K_{W_{t}}N_{t}^{T}+K_{Z_{t}}}{N_{t}K_{W_{t}}N_{t}^{T}}\right)\}. \end{array}$$

*$K_{t}=K_{t}^{s}=E_{s}\left\{E_{t}^{s}{\left(E_{t}^{s}\right)}^{T}\right\}$ satisfies the generalized DRE*

(210)
$$\begin{array}{cc} \; & K_{t+1}^{s}=A_{t}K_{t}A_{t}^{T}+B_{t}K_{W_{t}}B_{t}^{T}-\left(B_{t}K_{W_{t}}N_{t}^{T}+A_{t}K_{t}^{s}{\left(\Lambda_{t}+C_{t}\right)}^{T}\right)\{N_{t}K_{W_{t}}N_{t}^{T}+K_{Z_{t}}\\ \; & +\left(\Lambda_{t}+C_{t}\right)K_{t}^{s}{\left(\Lambda_{t}+C_{t}\right)}^{T}{\}}^{-1}{\left(B_{t}K_{W_{t}}N_{t}^{T}+A_{t}K_{t}^{s}\left(\Lambda_{t}+C_{t}\right)\right)}^{T},\;K_{t}^{s}⪰0,\;K_{1}^{s}=0,\;t=1,\ldots ,n. \end{array}$$

*and the statements of parts (a) and (b) hold.*



**Proof.** (a) The statements about Lemma 1 follow from Remark 7. (b) The statements about Theorem 3 are easily verified by replacing all conditional expectations, distributions, etc., for a fixed initial state $S_{1}=S_{1}^{s}=s$, and by using part (a), i.e., $\left(\Sigma_{t}^{s},{\hat{S}}_{t}^{s}\right)=\left(0,S_{t}^{s}\right)$, t=1,2,…. Part (c) follows from parts (a) and (b).  □

### 2.6. Comments on Past Studies

It is easily verified that Yang, Kavcic and Tatikonda [8] analyzed $C_{n}^{fb}\left(\kappa ,s\right)$, which is defined by (Equation 81), under the Case (II) formulation, i.e., Conditions 1 and 2 of Section 1.1 hold, as discussed in the next remark.

**Remark 11.** 
*Prior studies on the time-invariant stationary noise of PSD (Equation 41)*

*Yang, Kavcic and Tatikonda [8] analyzed the AGN channel driven by a stationary noise with PSD defined by (Equation 41) (see [8] (Theorem 1)). The special case of (Equation 126) is found in [8] (Section VI.B, Theorem 7).*

*The analysis in [8] presupposed the following formulation:*

*(i) The code is $\left(s,2^{nR},n\right)$, n=1,2,…, where $S_{1}=S_{1}^{s}=s$ is the initial state of the noise, known to the encoder and the decoder, as discussed in Definition 3;*

*(ii) Conditions 1 and 2 of Section 1.1 hold;*

*(iii) The n–FTFI capacity formula is $C_{n}^{fb}\left(\kappa ,s\right)$, defined by (Equation 81).*

*We emphasize that in [8] (Section II.C), a specific realization of the PSD is considered to ensure that Conditions 1 and 2 hold, i.e., the analysis in [8] presupposed a stationary noise and the Case (II) formulation.*



Now, we ask the following: Given the PSD of the noise defined by (Equation 41), and the double-sided realization [4] (Equation (58)), i.e., the analog of time-invariant version of the PO-SS realization of Definition 2, or its analogous one-sided realization, what are the necessary conditions for the feedback capacity of [4] (Theorem 6.1) to be valid?
The answer to this question is as follows: Conditions 1 and 2 of Section 1.1 are necessary conditions. We show this in the next proposition.

**Proposition 2.** 
*Conditions for validity of the feedback capacity characterization of [4] (Theorem 6.1)*

*Consider the AGN channel (Equation 1) driven by a stationary noise with PSD defined by (Equation 41) with the double-sided or one-sided realization [4] (Equation (58)), (i.e., analog of time invariant of Definition 2).*

*Then, a necessary condition for [4] (Theorem 6.1) to hold is*

(211)
$$\begin{array}{c} P_{X_{t}|X^{t-1},Y_{-\infty }^{t-1}}=P_{X_{t}|S^{t},Y_{-\infty }^{t-1}},\;t=1,\ldots , \end{array}$$

*Further, Conditions 1 and 2 of Section 1.1 are necessary and sufficient for Equality (Equation 211) to hold.*



**Proof.** See Section A.5.  □

The next remark is our final observation on prior studies.

**Remark 12.** 
*Comparison of Cover and Pombra Characterization and current literature*

*From Corollary 7 and Proposition 2, we have the following:*

*The characterization of feedback capacity given in [4] (Theorem 6.1, $C_{FB}$) corresponds to the Case (II) formulation and not to the Case (I) formulation. Further, the optimization problem of [4] (Theorem 6.1, $C_{FB}$) is precisely the optimization problem investigated in [8] (Section VI), with the additional restriction that the innovations’ part of the channel input is taken to be asymptotically zero in [4] (Theorem 6.1, $C_{FB}$), i.e., see [4] (Lemma 6.1 and comments above it). Recent studies [7,9,11,12] should be read with caution because the results therein often build on [4] (Theorems 4.1 and 6.1).*



## 3. Asymptotic Analysis for Case (I) Formulation

In this section, we address the asymptotic per unit time limit of the *n*–FTFI capacity. Our analysis includes the following:

(1) Fundamental differences of entropy rates of jointly Gaussian stable versus unstable noise processes.

(2) Necessary and/or sufficient conditions for existence of entropy rates of unstable (and stable) $Y^{n},n=1,2,\ldots$, and $V^{n},n=1,2,\ldots$, expressed in terms of detectability and stabilizability or unit circle controllability conditions of generalized DREs [16,17], and asymptotic stationarity of the optimal input process $X^{n},n=1,2,\ldots$ (and output process $Y^{n},n=1,2,\ldots$, if the noise is stable).

This section also reconfirms that, in general, the asymptotic analysis of the *n*–FTFI capacity of a feedback code that depends on the initial state of the channel, i.e., $S_{1}=S_{1}^{s}=s$ is fundamentally different from code that does does not depend on the initial state.

Closed-form expressions of the asymptotic per unit time limit of $C_{n}^{fb}\left(\kappa ,s\right)$ of AGN channels driven by AR(c),c∈(−∞,∞) noise, i.e., stable and unstable, are found in [1].

Closed-form expressions of the asymptotic per unit time limit of $C_{n}^{fb}\left(\kappa \right)$ of AGN channels driven by ARMA(a,c),a∈(−∞,∞),c∈(−∞,∞) noise are found in [3].

We consider the following definition of rate, often used for nonfeedback capacity of stationary processes; however, our formulations does not assume stationarity.

**Definition 4.** 
*Per unit time limit of $C_{n}^{fb,o}\left(\kappa \right)$ and $C_{n}^{fb,o}\left(\kappa ,s\right)$*

*Consider the AGN channel defined by (Equation 1), driven by the time-invariant PO-SS realization of Definition 2.*

*(a) For the code of Definition 1, define the per unit time limit*

(212)
$$\begin{array}{cc} C^{fb,o}\left(\kappa \right) & \overset{▵}{=}\underset{\lim_{n⟶\infty }\frac{1}{n}E\left\{\sum_{t=1}^{n}{\left(X_{t}\right)}^{2}\right\}\leq \kappa }{\sup}\lim_{n⟶\infty }\frac{1}{n}\left\{H\left(Y^{n}\right)-H\left(V^{n}\right)\right\} \end{array}$$


(213)
$$\begin{array}{cc} \; & \leq C^{fb}\left(\kappa \right)\overset{▵}{=}\lim_{n⟶\infty }\frac{1}{n}C_{n}^{fb}\left(\kappa \right) \end{array}$$

*where, for problem $C^{fb,o}\left(\kappa \right)$, the supremum is taken over all asymptotically time-invarinat distributions with feedback $P_{X_{t}|X^{t-1},Y^{t-1}}^{o}=P_{X_{t}|V^{t-1},Y^{t-1}}^{o},t=1,2,\ldots$, such that the limits exists and the supremum exists and it is finite.*

*(b) For the code of Definition 3, i.e., $\left(s,2^{nR},n\right)$, n=1,2,…, with initial state $S_{1}=S_{1}^{s}=s$, $C^{fb,o}\left(\kappa \right)$ is replaced by $C^{fb,o}\left(\kappa ,s\right)$, defined by (Equation 212), with differential entropies, conditional expectations, and conditional distributions, defined for fixed $S_{1}^{s}=s$.*



The rate definition, $C^{fb,o}\left(\kappa \right)$, i.e., the interchange of limit and supremum, is consistent with the definition of rates considered in [4,7,9,11,12]. However, unlike [4,7,9,11,12], we treat the general time-invariant, stable and unstable PO-SS noise realization of Definition 2, which is not necessarily stationary or asymptotically stationary.
We should emphasize that, in general, and irrespective of whether the noise is stable or unstable, the entropy rates that appear in (Equation 212) and (Equation 213) may not exist. To show existence of the limits $C^{fb,o}\left(\kappa \right),C^{fb}\left(\kappa \right)$ and $C^{fb,o}\left(\kappa ,s\right)$, we identify necessary and/or sufficient conditions, using the characterization of Theorem 3, when the channel input strategies are restricted to asymptotically time-invariant strategies $\lim_{n⟶\infty }\Lambda_{n}=\Lambda^{\infty },\lim_{n⟶\infty }K_{Z_{n}}=K_{Z}^{\infty }$. Clearly, by (Equation 212), whether the limit as, n⟶∞ and the supremum over channel input distributions exist, depend on the convergence properties of the coupled generalized matrix DREs, $\Sigma_{n},K_{n}$, as n⟶∞.

### 3.1. Entropy Rates of Gaussian Processes

First, we recall the following definition, which is standard and found in many textbooks:

**Definition 5.** 
*Entropy rate of continuous-valued random processes*

*Let $X_{t}:\Omega \to R^{n_{z}},n_{x}\in Z_{+}$ be a random process defined on some probability space (Ω,F,P). The entropy rate (differential) is defined by*

(214)
$$\begin{array}{c} H_{R}\left(X^{\infty }\right)\overset{▵}{=}\lim_{n⟶\infty }\frac{1}{n}H\left(X_{1},X_{2},\ldots ,X_{n}\right) \end{array}$$

*when the limit exists.*



The next theorem quantifies the existence of entropy rates of stationary Gaussian processes [16].

**Theorem 4.** 
*The entropy rate of stationary zero-mean full-rank Gaussian process [16]*

*Let $X_{t}:\Omega \to R^{n_{x}},n_{x}\in Z_{+},\forall t\in Z_{+}$ be a stationary Gaussian process, with a zero mean, and full-rank covariance of $X^{n}$. Let $H_{t}^{X}$ denote the Hilbert space of RVs generated by $\left\{X_{t}:s\leq t,s,t\in Z_{+}\right\}$, and define the innovations process by*

(215)
$$\begin{array}{c} \Sigma_{t}\overset{▵}{=}E\left\{\left(X_{t}-E\left\{X_{t}|H_{t-1}^{X}\right\}\right){\left(X_{t}-E\left\{X_{t}|H_{t-1}^{X}\right\}\right)}^{T}\right\}≻0 \end{array}$$

*and its limit*

(216)
$$\begin{array}{c} \Sigma \overset{▵}{=}\lim_{n⟶\infty }\Sigma_{n} \end{array}$$

*Then, the entropy rate is given by*

(217)
$$\begin{array}{cc} H_{R}\left(X^{\infty }\right)= & \frac{n_{x}}{2}\log\left(2\pi e\right)+\frac{1}{2}\lim_{n⟶\infty }\frac{1}{n}\sum_{t=1}^{n}\log\left|\Sigma_{t}\right| \end{array}$$


(218)
$$\begin{array}{cc} = & \frac{n_{x}}{2\pi }\log\left(2\pi e\right)+\frac{1}{2}\log\left|\Sigma \right| \end{array}$$

*when it exists.*



An application of Theorem 4 is given in the next proposition [15].

**Proposition 3.** 
*Entropy rate of Gaussian process described by PSD (Equation 41)*

*Let $V_{t},\forall t\in Z_{+}$ be a real, scalar-valued, stationary Gaussian noise with PSD (Equation 41), with a corresponding time-invariant stationary realization (similar to Definition 2). Then, the entropy rate is given by*

(219)
$$\begin{array}{c} H_{R}\left(V^{\infty }\right)=\frac{1}{2}\log\left(2\pi eK_{W}\right). \end{array}$$




**Proof.** This is shown in [15] by using the Szego formula and Poisson’s integral formula.  □

The next remark is trivial; it is introduced for a subsequent comparison.

**Remark 13.** 
*Let $V_{t},\forall t\in Z_{+}$ be the non-stationary ARMA(a,c),a∈(−∞,∞),c∈(−∞,∞) noise of Example 2. Then, the conditional entropy of $V^{n}$ for fixed initial state $S_{1}=S_{1}^{s}=s$ is given by*

(220)
$$\begin{array}{c} H_{R}\left(V^{\infty }|s\right)\overset{▵}{=}\lim_{n⟶\infty }\frac{1}{n}H\left(V^{n}|s\right)=\lim_{n⟶\infty }\frac{1}{n}\sum_{t=1}^{n}\frac{1}{2}\log\left(2\pi eK_{W}\right)=\frac{1}{2}\log\left(2\pi eK_{W}\right). \end{array}$$



The next lemma identifies fundamental conditions for the existence of the entropy rate of the time-varying PO-SS noise realization of Definition 2 (if $S_{1}=S_{1}^{s}=s$ is not fixed) and includes the entropy rate $H_{R}\left(V^{\infty }\right)$ of the non-stationary ARMA(a,c),a∈(−∞,∞),c∈(−∞,∞) noise of Remark 13.

**Lemma 2.** 
*Entropy rate of the time-varying PO-SS noise realization of Definition 2*

*Consider the time-varying PO-SS noise realization of Definition 2. Then, the following hold:*

*(a) The joint entropy of $V^{n}$, when it exists, is given by*

(221)
$$\begin{array}{c} H\left(V^{n}\right)=\sum_{t=1}^{n}H\left({\hat{I}}_{t}\right)=\frac{1}{2}\sum_{t=1}^{n}\log\left(2\pi eK_{{\hat{I}}_{t}}\right) \end{array}$$

*where ${\hat{I}}_{t},t=1,\ldots ,n$ is a zero-mean covariance $K_{{\hat{I}}_{t}}\overset{▵}{=}cov\left({\hat{I}}_{t},{\hat{I}}_{t}\right)$, Gaussian orthogonal innovation process of $V^{n}$ that is defined by*

(222)
$$\begin{array}{cc} {\hat{I}}_{t}\overset{▵}{=} & \;V_{t}-E\left\{V_{t}|V^{t-1}\right\},\;t=1,\ldots ,n \end{array}$$

*that is, ${\hat{I}}_{t}$ is independent of ${\hat{I}}_{k},\forall k\neq t$.*

*(b) Suppose that the sequence $K_{{\hat{I}}_{t}},t=1,2,\ldots ,n$ is such that*

(223)
$$\begin{array}{c} \lim_{n⟶\infty }K_{{\hat{I}}_{n}}=K_{\hat{I}}^{\infty }≻0. \end{array}$$

*Then, the entropy rate of $V_{t},\forall t\in Z_{+}$ is given by*

(224)
$$\begin{array}{c} H_{R}\left(V^{\infty }\right)=\lim_{n⟶\infty }\frac{1}{n}\sum_{t=1}^{n}H\left({\hat{I}}_{t}\right)=\frac{1}{2}\log\left(2\pi eK_{\hat{I}}^{\infty }\right). \end{array}$$




**Proof.** See Section A.6.  □

**Remark 14.** 
*Entropy rate of non-stationary Gaussian noise*

*By Lemma 2, a necessary condition for existence of the entropy rate of non-stationary Gaussian process $V^{n}$ is the convergence of the covariance of the Gaussian orthogonal innovations process of $V^{n}$, i.e., of $K_{{\hat{I}}_{t}}\overset{▵}{=}cov\left({\hat{I}}_{t},{\hat{I}}_{t}\right)$, since $\lim_{n\to \infty }\frac{1}{n}H\left(V^{n}\right)=\lim_{n\to \infty }\frac{1}{n}\sum_{t=1}^{n}H\left({\hat{I}}_{t}\right)$. We can determine such necessary and/or sufficient conditions from the convergence properties of the generalized Kalman filter equations [16,17] of Lemma 1.*



### 3.2. Convergence Properties of Generalized Matrix DREs to AREs

To address the asymptotic properties of estimation errors generated by the recursions of generalized Kalman filters, such as ${\hat{E}}_{t},t=1,2,\ldots$ of Theorem 3, generated by (Equation 178), we need to introduce the stabilizing solutions of generalized AREs. The next definition is useful in this respect.

**Definition 6.** 
*Stabilizing solutions of generalized matrix AREs*

*Let $\left(A,G,Q,S,R,C\right)\in R^{q\times q}\times R^{q\times k}\times R^{k\times k}\times R^{k\times p}\times R^{p\times p}\times R^{p\times q}$.*

*Define the generalized time-invariant matrix DRE*

(225)
$$\begin{array}{cc} \; & P_{t+1}=AP_{t}A^{T}+GQG^{T}-\left(AP_{t}C^{T}+GS\right){\left(R+CP_{t}C^{T}\right)}^{-1}.{\left(AP_{t}C^{T}+GS\right)}^{T},\;P_{1}=given,\\ \; & P_{t}\in S_{+}^{q\times q},\;t=1,\ldots ,\;R=R^{T}≻0,\\ \; & F^{CL}\left(P\right)\overset{▵}{=}A-\left(APC^{T}+GS\right){\left(R+CPC^{T}\right)}^{-1}C. \end{array}$$

*Moreover, define the corresponding generalized matrix ARE as follows:*

(226)
$$\begin{array}{cc} P= & APA^{T}+GQG^{T}-\left(APC^{T}+GS\right){\left(R+CPC^{T}\right)}^{-1}.{\left(APC^{T}+GS\right)}^{T},\;P\in S_{+}^{q\times q}. \end{array}$$

*A solution $P=P^{T}⪰0$ to the generalized matrix ARE (Equation 226), assuming it exists, is called stabilizing if $spec\left(F^{CL}\left(P\right)\right)\in D_{o}$. In this case, we say that $F^{CL}\left(P\right)$ is asymptotically stable, i.e., the eigenvalues of $F^{CL}\left(P\right)$ are stable.*



With respect to any of the above generalized matrices DRE and ARE, we introduce the important notions of detectability, unit circle controllability, and stabilizability. We use these notions to characterize the convergence properties of solutions of generalized matrix DREs, $P_{n}$, as n⟶∞, to a unique symmetric, non-negative, stabilizing solution *P* of the generalized matrix ARE. These notions are used to identify necessary and/or sufficient conditions for the error recursions of generalized Kalman filters, such as ${\hat{E}}_{t},t=1,2,\ldots$ of Theorem 3, generated by (Equation 178), to converge in a mean square sense, to a unique limit. However, we should distinguish whether the convergence is uniform for all initial conditions, $P_{1}⪰0$ or only for $P_{1}≻0$.

**Definition 7.** 
*Detectability, Stabilizability, and Unit Circle Controllability*

*Consider the generalized matrix ARE of Definition 6 and introduce the matrices*

(227)
$$\begin{array}{c} A^{*}\overset{▵}{=}A-GSR^{-1}C,\;B^{*}\overset{▵}{=}Q-SR^{-1}S^{T},\;B^{*}=B^{*,\frac{1}{2}}{\left(B^{*,\frac{1}{2}}\right)}^{T}. \end{array}$$

*(a) The pair $\left\{A,C\right\}$ is called detectable if there exists a matrix $K\in R^{q\times p}$ such that $spec\left(A-KC\right)\in D_{o}$, i.e., the eigenvalues λ of A−KC lie in $D_{o}$ (stable).*

*(b) The pair $\left\{A^{*},GB^{*,\frac{1}{2}}\right\}$ is called unit circle controllable if there exists a $K\in R^{k\times q}$ such that $spec\left(A^{*}-GB^{*,\frac{1}{2}}K\right)\notin \left\{c\in C:|c|=1\right\}$, i.e., all eigenvalues λ of $A^{*}-GB^{*,\frac{1}{2}}K$ are such that |λ|≠1.*

*(c) The pair $\left\{A^{*},GB^{*,\frac{1}{2}}\right\}$ is called stabilizable if there exists a $K\in R^{k\times q}$ such that $spec\left(A^{*}-GB^{*,\frac{1}{2}}K\right)\in D_{o}$, i.e., all all eigenvalues λ of $A^{*}-GB^{*,\frac{1}{2}}K$ lie in $D_{o}$.*

*(d) The pair $\left\{A,C\right\}$ is called observable if the rank condition holds:*

(228)
$$\begin{array}{c} rank\left(O\right)=q,\;O\overset{▵}{=}\left[\begin{array}{c} C\\ CA\\ \vdots \\ CA^{q-1} \end{array}\right]. \end{array}$$

*(e) The pair $\left\{A^{*},GB^{*,\frac{1}{2}}\right\}$ is called controllable if the rank condition holds:*

(229)
$$\begin{array}{c} rank\left(C\right)=q,\;C\overset{▵}{=}\left[\begin{array}{cccc} GB^{*,\frac{1}{2}} & A^{*}GB^{*,\frac{1}{2}} & \ldots & {\left(A^{*}\right)}^{q-1}GB^{*,\frac{1}{2}} \end{array}\right]. \end{array}$$




**Remark 15.** 
*The following are well known [16]. If the pair $\left\{A,C\right\}$ is observable, then it is detectable. If the pair $\left\{A^{*},GB^{*,\frac{1}{2}}\right\}$ is controllable, then it is stabilizable.*


The next theorem characterizes detectability, unit circle controllability, and stabilizability [17,40].

**Lemma 3** ([17,40]). *Necessary and sufficient conditions for detectability, unit circle controllability, and stabilizability*
*(a) The pair $\left\{A,C\right\}$ is detectable if and only if there exists no eigenvalue and eigenvector {λ,x}, Ax=λx, such that |λ|≥1 and such that Cx=0.**(b) The pair $\left\{A^{*},GB^{*,\frac{1}{2}}\right\}$ is unit circle controllable if and only if there exists no eigenvalue and eigenvector {λ,x}, $x^{T}{\left(A^{*}\right)}^{T}=x^{T}\lambda$, such that |λ|=1 and such that $x^{T}GB^{*,\frac{1}{2}}=0$.**(c) The pair $\left\{A^{*},GB^{*,\frac{1}{2}}\right\}$ is stabilizable if and only there exists no eigenvalue and eigenvector {λ,x}, $x^{T}{\left(A^{*}\right)}^{T}=x^{T}\lambda$, such that |λ|≥1 and such that $x^{T}GB^{*,\frac{1}{2}}=0$.*

In the next theorem, we summarize known results on sufficient and/or necessary conditions for the convergence of solutions $\left\{P_{t},t=1,2,\ldots ,n\right\}$ of the generalized time-invariant DRE (Equation 225), as n⟶∞, to a symmetric, non-negative P⪰0, stabilizing solution of the corresponding generalized ARE (Equation 226). We recall that the pair $\left\{A,C\right\}$ is detectable, which is a necessary condition forthe convergence of the sequence $\left\{P_{t},t=1,2,\ldots ,n\right\}$ as n⟶∞ to a non-negative *P*, which is a stabilizing solution of a corresponding generalized ARE. However, it is not sufficient. To have a sufficient condition, it is necessary that the pair $\left\{A^{*},GB^{*,\frac{1}{2}}\right\}$ is unit circle controllable; however, the limiting *P* is not necessarily the unique solution of P⪰0 of the generalized ARE. There may be multiple solutions depending on the initial condition $P_{1}⪰0$.

**Theorem 5** ([16,17]). *Convergence of time-invariant generalized DRE*
*Let $\left\{P_{t},t=1,2,\ldots ,n\right\}$ denote a sequence that satisfies the time-invariant generalized DRE (Equation 225) with arbitrary initial condition $P_{1}⪰0$. The following hold:**(1) Consider the generalized DRE (Equation 225) with a zero initial condition, i.e., $P_{1}=0$, and assume that the pair $\left\{A,C\right\}$ is detectable and that the pair $\left\{A^{*},GB^{*,\frac{1}{2}}\right\}$ is unit circle controllable.**Then, the sequence $\left\{P_{t}:t=1,2,\ldots ,n\right\}$ that satisfies the generalized DRE (Equation 225), with a zero initial condition $P_{1}=0$, converges to P, i.e., $\lim_{n⟶\infty }P_{n}=P$, where P satisfies the generalized matrix ARE (Equation 226) if and only if the pair $\left\{A^{*},GB^{*,\frac{1}{2}}\right\}$ is stabilizable.**(2) Assume that the pair $\left\{A,C\right\}$ is detectable and that the pair $\left\{A^{*},GB^{*,\frac{1}{2}}\right\}$ is unit circle controllable. Then there exists a unique stabilizing solution P⪰0 to the generalized ARE (Equation 226), i.e., such that $spec\left(F^{CL}\left(P\right)\right)\in D_{o}$ if and only if $\left\{A^{*},GB^{*,\frac{1}{2}}\right\}$ is stabilizable.**(3) If {A,C} is detectable and $\left\{A^{*},GB^{*,\frac{1}{2}}\right\}$ is stabilizable, then any solution $P_{t},t=1,2,\ldots ,n$ to the generalized matrix DRE (Equation 225) with arbitrary initial condition $P_{1}⪰0$ is such that $\lim_{n⟶\infty }P_{n}=P$, where P⪰0 is the unique solution of the generalized matrix ARE (Equation 226) with $spec\left(F^{CL}\left(P\right)\right)\in D_{o}$, i.e., it is stabilizing.**(4) {A,C} is detectable and $\left\{A^{*},GB^{*,\frac{1}{2}}\right\}$ unit circle controllable, which are necessary and sufficient conditions for any solution $P_{t},t=2,\ldots ,n$ to the generalized DRE (Equation 225) to converge, $\lim_{n⟶\infty }P_{n}=P$, from some initial condition $P_{1}⪰0$, where P⪰0 is a stabilizing solution of the generalized ARE (Equation 226), but it may not be unique (i.e., (Equation 226) may have multiple solutions P⪰0).*

**Proposition 4.** 
*Generalizations to asymptotic-time invariant coefficients*

*Suppose that the coefficients of the generalized matrix DRE (Equation 225), (A,C,G,Q,S) are replaced by $\left(A_{t},C_{t},G_{t},Q_{t},S_{t}\right)$, t=1,2,…,n, and they are asymptotically time-invariant, i.e.,*

(230)
$$\begin{array}{c} \lim_{n⟶\infty }\left(A_{n},C_{n},G_{n},Q_{n},S_{n}\right)=\left(A,C,G,Q,S\right). \end{array}$$

*Then, Theorem 5 remains valid.*



**Proof.** This is due to the well-known continuity properties of matrix DREs with respect to their coefficients, i.e., the convergence properties are characterized by the limiting pairs, $\left\{A,C\right\}$ and $\left\{A^{*},GB^{*,\frac{1}{2}}\right\}$.  □

### 3.3. Feedback Rates

Now, we return to the feedback rates of Definition 4. The next corollary is an application of Theorem 5 to the generalized Kalman filter of Lemma 1 (for the time-invariant PO-SS realization); it identifies conditions for existence of the entropy rate $H_{R}\left(V^{\infty }\right)$, irrespective of whether the noise is stable or unstable.

**Corollary 8.** 
*The entropy rate of PO-SS noise realization based on the generalized Kalman filter*

*Let $\Sigma_{t}^{o}=\Sigma_{t},t=1,2,\ldots$ denote the solution of the generalized matrix DRE (Equation 95) of the generalized Kalman filter of Lemma 1 of the time-invariant PO-SS realization of $V^{n}$ of Definition 2, i.e., $\left(A_{t},B_{t},C_{t},N_{t},K_{W_{t}}\right)=\left(A,B,C,N,K_{W}\right),\forall t$, generated by*

$$\begin{array}{cc} \Sigma_{t+1}^{o}= & A\Sigma_{t}^{o}A^{T}+BK_{W}B^{T}-\left(A\Sigma_{t}^{o}C^{T}+BK_{W}N^{T}\right){\left(NK_{W}N^{T}+C\Sigma_{t}^{o}C^{T}\right)}^{-1} \end{array}$$


(231)
$$\begin{array}{cc} \; & \;.{\left(A\Sigma_{t}^{o}C^{T}+BK_{W}N^{T}\right)}^{T},\;\Sigma_{t}^{o}⪰0,\;t=1,\ldots ,n,\;\Sigma_{1}^{o}=K_{S_{1}}⪰0. \end{array}$$


(232)
$$\begin{array}{cc} M^{CL}\left(\Sigma^{o}\right)\overset{▵}{=} & A-M\left(\Sigma^{o}\right)C,\;M\left(\Sigma^{o}\right)\overset{▵}{=}\left(A\Sigma^{o}C^{T}+B_{t}K_{W}N^{T}\right){\left(NK_{W}N^{T}+C\Sigma^{o}C^{T}\right)}^{-1}. \end{array}$$

*Let $\Sigma^{\infty }=\Sigma^{\infty ,T}⪰0$ be a solution of the corresponding generalized ARE*

(233)
$$\begin{array}{cc} \Sigma^{\infty }= & A\Sigma^{\infty }A^{T}+BK_{W}B^{T}\\ \; & -\left(A\Sigma^{\infty }C^{T}+BK_{W}N^{T}\right){\left(NK_{W}N^{T}+C\Sigma^{\infty }C^{T}\right)}^{-1}{\left(A\Sigma^{\infty }C^{T}+BK_{W}N^{T}\right)}^{T}. \end{array}$$

*Define the matrices*

(234)
$$\begin{array}{cc} \; & GQG^{T}\overset{▵}{=}BK_{W}B^{T},\;GS\overset{▵}{=}BK_{W}N^{T},\;R\overset{▵}{=}NK_{W}N^{T}\;⟹\;G\overset{▵}{=}B,\;Q\overset{▵}{=}K_{W},\;S\overset{▵}{=}K_{W}N^{T}, \end{array}$$


(235)
$$\begin{array}{cc} \; & A^{*}\overset{▵}{=}A-BK_{W}N^{T}{\left(NK_{W}N^{T}\right)}^{-1}C,\;B^{*}\overset{▵}{=}K_{W}-K_{W}N^{T}{\left(NK_{W}N^{T}\right)}^{-1}{\left(K_{W}N^{T}\right)}^{T}. \end{array}$$

*(a) All statements of Theorem 5 hold with (G,Q,S,R) as defined by (Equation 234) and (Equation 235).*

*In particular, suppose the following:*

*(i) {A,C} is detectable;*

*(ii) $\left\{A^{*},GB^{*,\frac{1}{2}}\right\}$ is stabilizable.*

*Then, any solution $\Sigma_{t}^{o},t=1,2,\ldots ,n$ to the generalized matrix DRE (Equation 231) with arbitrary initial condition $\Sigma_{1}^{o}⪰0$ is such that $\lim_{n⟶\infty }\Sigma_{n}^{o}=\Sigma^{\infty }$, where $\Sigma^{\infty }⪰0$ is the unique and stabilizing solution of the generalized matrix ARE (Equation 233), i.e., with $spec\left(M^{CL}\left(\Sigma^{\infty }\right)\right)\in D_{o}$.*

*(b) Suppose that (i) and (ii) hold. The entropy rate of $V^{n}$ is given by*

(236)
$$\begin{array}{cc} H_{R}\left(V^{\infty }\right)= & \lim_{n⟶\infty }\frac{1}{2n}\sum_{t=1}^{n}\log\left(2\pi e\left[C\Sigma_{t}^{o}C^{T}+NK_{W}N^{T}\right]\right) \end{array}$$


(237)
$$\begin{array}{cc} = & H\left({\hat{I}}_{t}^{\infty }\right)\overset{▵}{=}\frac{1}{2}\log\left(2\pi e\left[C\Sigma^{\infty }C^{T}+NK_{W}N^{T}\right]\right),\;\forall \Sigma_{1}^{o}⪰0,\;\forall t \end{array}$$

*where*

(238)
$$\begin{array}{c} {\hat{I}}_{t}^{\infty }\overset{▵}{=}C\left(S_{t}-{\hat{S}}_{t}^{\infty }\right)+NW_{t}\in N\left(0,C\Sigma^{\infty }C^{T}+NK_{W}N^{T}\right),\;t=1,2,\ldots , \end{array}$$

*is the stationary Gaussian innovations process, i.e., with $\Sigma_{t}^{o}$ replaced by $\Sigma^{\infty }$, and the entropy rate $H_{R}\left(V^{\infty }\right)$ is independent of the initial data $\Sigma_{1}^{o}⪰0$.*

*(c) Suppose in parts (a) and (b), the condition $\left\{A^{*},GB^{*,\frac{1}{2}}\right\}$ is stabilizable is replaced by*

*(iii) $\left\{A^{*},GB^{*,\frac{1}{2}}\right\}$ is unit circle controllable.*

*Then, the statements of parts (a) and (b) hold, for some $\Sigma_{1}^{o}⪰0$, but not for all $\Sigma_{1}^{o}⪰0$. Moreover, ${\hat{I}}_{t}^{\infty }$ is not necessarily a stationary process, i.e., it depends on the value of $\Sigma_{1}^{o}⪰0$.*



**Proof.** (a) These are direct applications of Theorem 5. (b) This follows from Lemma 2. (c) This is due to Theorem 5(4).  □

Next, we apply Corollary 8 to the non-stationary AR(a,c),a∈(−∞,∞),c∈(−∞,∞) noise.

**Lemma 4.** 
*Properties of solutions of DREs and AREs of AR(a,c),a∈(−∞,∞),c∈(−∞,∞) noise and entropy rate $H_{R}\left(V^{\infty }\right)$*

*Consider the AR(a,c),a∈(−∞,∞),c∈(−∞,∞) noise of Example 2(a), and the DRE $\Sigma_{t}^{o}\overset{▵}{=}\Sigma_{t},t=1,\ldots ,n$, generated by Corollary 6(a), i.e.,*

(239)
$$\begin{array}{cc} \; & \Sigma_{t+1}^{o}={\left(c\right)}^{2}\Sigma_{t}^{o}+K_{W}-{\left(c\Sigma_{t}^{o}\left(c-a\right)+K_{W}\right)}^{2}{\left(K_{W}+{\left(c-a\right)}^{2}\Sigma_{t}^{o}\right)}^{-1},\;t=1,\ldots ,n, \end{array}$$


(240)
$$\begin{array}{cc} \; & \Sigma_{1}^{o}=K_{S_{1}}=\frac{{\left(c_{0}\right)}^{2}K_{S_{0}}+{\left(a_{0}\right)}^{2}K_{W_{0}}}{{\left(c_{0}-a_{0}\right)}^{2}}\geq 0. \end{array}$$

*where $K_{W}>0,c\neq a$, $K_{S_{0}}\geq 0,K_{W_{0}}\geq 0$. Let $\Sigma^{\infty }\geq 0$ be a solution of the corresponding generalized ARE, as follows:*

(241)
$$\begin{array}{cc} \; & \Sigma^{\infty }={\left(c\right)}^{2}\Sigma^{\infty }+K_{W}-{\left(c\Sigma^{\infty }\left(c-a\right)+K_{W}\right)}^{2}{\left(K_{W}+{\left(c-a\right)}^{2}\Sigma^{\infty }\right)}^{-1}. \end{array}$$

*Then, the detectability and stabilizability pairs are*

(242)
$$\begin{array}{c} \left\{A,C\right\}=\left\{c,c-a\right\},\;\left\{A^{*},GB^{*,\frac{1}{2}}\right\}=\left\{a,0\right\}. \end{array}$$

*and the following hold:*

*(1) The pair {A,C}={c,c−a} is detectable ∀c∈(−∞,∞),a∈(−∞,∞) (the restriction c≠a is always assumed).*

*(2) The pair $\left\{A^{*},GB^{*,\frac{1}{2}}\right\}=\left\{a,0\right\}$ is unit circle controllable if and only if |a|≠1 (∀c∈(−∞,∞)).*

*(3) The pair $\left\{A^{*},GB^{*,\frac{1}{2}}\right\}=\left\{a,0\right\}$ is stabilizable if and only if a∈(−1,1) (∀c∈(−∞,∞)).*

*(4) Suppose c∈(−∞,∞) and a∈(−∞,∞). The sequence $\left\{\Sigma_{t}^{o},t=1,2,\ldots ,n\right\}$ that satisfies the generalized DRE with any initial condition, $\Sigma_{1}^{o}\geq 0$, converges to $\Sigma^{\infty }$, i.e., $\lim_{n⟶\infty }\Sigma_{n}^{o}=\Sigma^{\infty }$, where $\Sigma^{\infty }\geq 0$ satisfies the ARE (Equation 241) if and only if the $\left\{A^{*},GB^{*,\frac{1}{2}}\right\}=\left\{a,0\right\}$ is unit circle controllable, equivalently, |a|≠1. Moreover, the solutions of the quadratic Equation (Equation 241), without imposing $\Sigma^{\infty }\geq 0$ are*

(243)
$$\begin{array}{c} \Sigma^{\infty }=\left\{\begin{array}{cc} 0 & unique,stabilizing,\Sigma^{\infty }\geq 0\;sol.\;of\;\text{(241)}\;for\;c\in \left(-\infty ,\infty \right),\left|a\right|<1\\ \frac{K_{W}\left(a^{2}-1\right)}{{\left(c-a\right)}^{2}}>0 & maximal,stabilizing,\Sigma^{\infty }>0\;sol.\;of\;\text{(241)}\;for\;c\in \left(-\infty ,\infty \right),\left|a\right|>1\\ \frac{K_{W}\left(a^{2}-1\right)}{{\left(c-a\right)}^{2}}<0 & non\text{-}stabilizing,\Sigma^{\infty }<0\;sol.\;of\;\text{(241)}\;for\;c\in \left(-\infty ,\infty \right),\left|a\right|<1. \end{array}\right. \end{array}$$

*That is, $\lim_{n⟶\infty }\Sigma_{n}^{o}=\Sigma^{\infty }=0,\forall \Sigma_{1}^{o}\geq 0$, is the unique and stabilizing solution $\Sigma^{\infty }\geq 0$ of (Equation 241), i.e., such that $|M^{CL}\left(\Sigma^{\infty }\right)|<1$, if and only if |a|<1, and*

*$\lim_{n⟶\infty }\Sigma_{n}^{0}=\Sigma^{\infty }=\frac{K_{W}\left(a^{2}-1\right)}{{\left(c-a\right)}^{2}}>0,\forall \Sigma_{1}^{o}>0$ is the maximal and stabilizing solution $\Sigma^{\infty }$ of (Equation 241), i.e., such that $|M^{CL}\left(\Sigma^{\infty }\right)|<1$, if and only if |a|>1.*

*(5) Suppose c∈(−∞,∞) and |a|<1. Then, any solution $\Sigma_{t}^{o},t=1,2,\ldots ,n$ to the generalized DRE (Equation 239) with an arbitrary initial condition, $\Sigma_{1}^{o}\geq 0$, is such that $\lim_{n⟶\infty }\Sigma_{n}^{o}=\Sigma^{\infty }$, where $\Sigma^{\infty }\geq 0$ is the unique solution of the generalized ARE (Equation 241) with $M^{CL}\left(\Sigma^{\infty }\right)\in \left(-1,1\right)$, i.e., it is stabilizing. Moreover, $\Sigma^{\infty }=0$.*

*(6) (i) Suppose c∈(−∞,∞) and |a|<1. The entropy rate of $V_{t},\forall t\in Z_{+}$ is given by*

(244)
$$\begin{array}{cc} H_{R}\left(V^{\infty }\right)= & \lim_{n⟶\infty }\frac{1}{n}\sum_{t=1}^{n}\frac{1}{2}\log\left(2\pi e\left[{\left(c-a\right)}^{2}\Sigma_{t}^{o}+K_{W}\right]\right) \end{array}$$


(245)
$$\begin{array}{cc} = & \frac{1}{2}\log\left(2\pi eK_{W}\right),\;\forall \Sigma_{1}^{o}\geq 0. \end{array}$$

*(ii) Suppose c∈(−∞,∞) and |a|>1. The entropy rate of $V_{t},\forall t\in Z_{+}$ is given by*

(246)
$$\begin{array}{cc} H_{R}\left(V^{\infty }\right)= & \lim_{n⟶\infty }\frac{1}{n}\sum_{t=1}^{n}\frac{1}{2}\log\left(2\pi e\left[{\left(c-a\right)}^{2}\Sigma_{t}^{o}+K_{W}\right]\right) \end{array}$$


(247)
$$\begin{array}{cc} = & \frac{1}{2}\log\left(2\pi eK_{W}a^{2}\right),\;\forall \Sigma_{1}^{o}>0. \end{array}$$




**Proof.** See Section A.7.  □

**Remark 16.** 
*Lemma 4(4) emphasizes the fact that in the asymptotic analysis of $\left\{\Sigma_{t}^{o},t=1,2,\ldots ,\right\}$, which satisfies the DREs (Equation 239) and (Equation 240), its limiting value, $\lim_{n⟶\infty }\Sigma_{n}^{o}=\Sigma^{\infty }$, where $\Sigma^{\infty }\geq 0$ satisfies the ARE (Equation 241), with two solutions $\Sigma^{\infty }=0$ and $\Sigma^{\infty }=\frac{K_{W}\left(a^{2}-1\right)}{{\left(c-a\right)}^{2}}$. For any c∈(−∞,∞), it is clear that for |a|<1, the unique and stabilizing solution is $\Sigma^{\infty }=0$, $\forall \Sigma_{1}^{o}\geq 0$, since the other solution $\Sigma^{\infty }=\frac{K_{W}\left(a^{2}-1\right)}{{\left(c-a\right)}^{2}}<0$ is negative. On the other hand, for any c∈(−∞,∞) and |a|>1, the stabilizing solution is the maximal solution, $\Sigma^{\infty }=\frac{K_{W}\left(a^{2}-1\right)}{{\left(c-a\right)}^{2}}>0$, provided $\Sigma_{1}^{o}>0$.*


To gain additional insights, we discuss the application of Lemma 4 to the AR(c),c∈(−∞,∞) noise in the next remark.

**Remark 17.** 
*Entropy rate $H_{R}\left(V^{\infty }\right)$ of the AR(c),c∈(−∞,∞) noise*

*Consider the non-stationary AR(c),c∈(−∞,∞) noise defined by (Equation 124). Then, from Lemma 4, $\Sigma_{t}^{o},t=1,\ldots ,n$ is the solution of (Equation 239) and (Equation 240), with a=0 (see Corollary 6(b), (Equation 201)), and (Equation 241) degenerates to the ARE, as follows:*

(248)
$$\begin{array}{c} \Sigma^{\infty }={\left(c\right)}^{2}\Sigma^{\infty }+K_{W}-{\left({\left(c\right)}^{2}\Sigma^{\infty }+K_{W}\right)}^{2}{\left(K_{W}+{\left(c\right)}^{2}\Sigma^{\infty }\right)}^{-1} \end{array}$$

*For a=0, by (Equation 242), the pair {A,C}={c,c} is detectable, and the pair $\left\{A^{*},GB^{*,\frac{1}{2}}\right\}=\left\{0,0\right\}$ is stabilizable. The two solutions of the ARE (Equation 248), without imposing $\Sigma^{\infty }\geq 0$, are*

(249)
$$\begin{array}{c} \Sigma^{\infty }=\left\{\begin{array}{cc} 0 & the\;unique,\;stabilizing,\;non\text{-}negative\;solution\;of\;the\;ARE\\ -\frac{K_{W}}{c^{2}}<0 & the\;non\text{-}stabilizing,\;negative\;solution\;of\;the\;ARE. \end{array}\right. \end{array}$$

*That is, $\lim_{n⟶\infty }\Sigma_{n}^{o}=\Sigma^{\infty }\geq 0$, where $\Sigma^{\infty }=0$ is the unique (stabilizing) solution of the ARE and corresponds to the stable eigenvalue of the error equation (see (Equation 93), i.e., $M^{CL}\left(\Sigma^{\infty }\right)=c-\frac{K_{W}}{K_{W}}c=0$.*



Next, we compute the entropy rate $H_{R}\left(V^{\infty }\right)$ of the time-invariant non-stationary PO-SS$\left(a,c,b^{1},b^{2},d^{1},d^{2}\right)$ noise of Corollary 2 to show fundamental differences from the entropy rate $H_{R}\left(V^{\infty }\right)$ of the AR(a,c) noise of Lemma 4.

**Lemma 5.** 
*Properties of solutions of DREs and AREs of PO-SS$\left(a,c,b^{1}=b,b^{2}=0,d^{1}=0,d^{2}=d\right)$ noise and entropy rate $H_{R}\left(V^{\infty }\right)$*

*Consider the the time-invariant non-stationary PO-SS$\left(a,c,b^{1},b^{2}=0,d^{1}=0,d^{2}=d\right)$ noise of Example 1, i.e., given by*

(250)
$$\begin{array}{cc} \; & S_{t+1}=aS_{t}+bW_{t}^{1},\;t=1,2,\ldots ,n-1 \end{array}$$


(251)
$$\begin{array}{cc} \; & V_{t}=cS_{t}+dW_{t}^{2},\;t=1,\ldots ,n, \end{array}$$

*and the sequence $\Sigma_{t}^{o}\overset{▵}{=}\Sigma_{t},t=1,\ldots ,n$, generated by the DRE of Lemma 1 (see (Equation 113), i.e.,*

(252)
$$\begin{array}{cc} \; & \Sigma_{t+1}^{o}={\left(a\right)}^{2}\Sigma_{t}^{o}+{\left(b\right)}^{2}K_{W^{1}}\\ \; & -{\left(a\Sigma_{t}^{o}c\right)}^{2}{\left({\left(d\right)}^{2}K_{W^{2}}+{\left(c\right)}^{2}\Sigma_{t}^{o}\right)}^{-1},\;t=1,\ldots ,n,\;\Sigma_{1}^{o}=K_{S_{1}}\geq 0,\;\Sigma_{t}^{o}\geq 0 \end{array}$$

*where ${\left(b\right)}^{2}K_{W^{1}}\geq 0,{\left(d\right)}^{2}K_{W}^{2}>0$. Let $\Sigma^{\infty }\geq 0$ be the corresponding solution of generalized ARE:*

(253)
$$\begin{array}{c} \Sigma^{\infty }={\left(a\right)}^{2}\Sigma^{\infty }+{\left(b\right)}^{2}K_{W^{1}}-{\left(a\Sigma^{\infty }c\right)}^{2}{\left({\left(d\right)}^{2}K_{W^{2}}+{\left(c\right)}^{2}\Sigma^{\infty }\right)}^{-1}. \end{array}$$

*Then, the detectability and stabilizability pairs are*

(254)
$$\begin{array}{c} \left\{A,C\right\}=\left\{a,c\right\},\;\left\{A^{*},GB^{*,\frac{1}{2}}\right\}=\left\{a,b{\left(K_{W^{1}}\right)}^{\frac{1}{2}}\right\}. \end{array}$$

*and the following hold:*

*(1) The pair {A,C}={a,c} is detectable ∀c∈(−∞,∞),a∈(−∞,∞),c≠0. If c=0 the pair {A,C}={a,0} is detectable if and only if |a|<1.*

*(2) The pair $\left\{A^{*},GB^{*,\frac{1}{2}}\right\}=\left\{a,b{\left(K_{W^{1}}\right)}^{\frac{1}{2}}\right\}$ is unit circle controllable if and only if $|b{\left(K_{W^{1}}\right)}^{\frac{1}{2}}|\neq 1$, ∀a∈(−∞,∞),c∈(−∞,∞).*

*(3) The pair $\left\{A^{*},GB^{*,\frac{1}{2}}\right\}=\left\{a,b{\left(K_{W^{1}}\right)}^{\frac{1}{2}}\right\}$ is stabilizable if $b{\left(K_{W^{1}}\right)}^{\frac{1}{2}}\neq 0$, ∀a∈(∞,∞),c∈(−∞,∞). If $b{\left(K_{W^{1}}\right)}^{\frac{1}{2}}=0$ the pair $\left\{A^{*},GB^{*,\frac{1}{2}}\right\}=\left\{a,0\right\}$ is stabilizable if and only if |a|<1.*

*(4) Define the set*

(255)
$$\begin{array}{cc} L^{\infty }\overset{▵}{=}\{\left(a,c,{\left(b\right)}^{2}K_{W^{1}}\right)\in & {\left(-\infty ,\infty \right)}^{2}\times \left[0,\infty \right):\;(i)\;the\;pair\;\{A,C\}=\{a,c\}\;is\;detectable,\;and\\ \; & (ii)\;the\;pair\;\left\{A^{*},GB^{*,\frac{1}{2}}\right\}=\left\{a,b{\left(K_{W^{1}}\right)}^{\frac{1}{2}}\right\}\;is\;stabilizable\}. \end{array}$$

*For any $\left(a,c,b{\left(K_{W^{1}}\right)}^{\frac{1}{2}}\right)\in L^{\infty }$, any solution $\Sigma_{t}^{o},t=1,2,\ldots ,n$ to the (classical) DRE (Equation 252) with an arbitrary initial condition, $\Sigma_{1}^{o}\geq 0$ is such that $\lim_{n⟶\infty }\Sigma_{n}^{o}=\Sigma^{\infty }$, where $\Sigma^{\infty }\geq 0$ is the unique solution of the (classical) ARE (Equation 253) with $M^{CL}\left(\Sigma^{\infty }\right)\in \left(-1,1\right)$, i.e., it is stabilizing.*

*(5) For any $\left(a,c,b^{2}K_{W^{1}}\right)\in L^{\infty }$ of part (4), the entropy rate of $V_{t},\forall t\in Z_{+}$ is given by*

(256)
$$\begin{array}{cc} H_{R}\left(V^{\infty }\right)= & \lim_{n⟶\infty }\frac{1}{n}\sum_{t=1}^{n}\frac{1}{2}\log\left(2\pi e\left[{\left(c\right)}^{2}\Sigma_{t}^{o}+{\left(d\right)}^{2}K_{W^{2}}\right]\right) \end{array}$$


(257)
$$\begin{array}{cc} = & \frac{1}{2}\log\left(2\pi e\left[{\left(c\right)}^{2}\Sigma^{\infty }+{\left(d\right)}^{2}K_{W^{2}}\right]\right),\;\forall \Sigma_{1}^{o}\geq 0. \end{array}$$




**Proof.** Follows from Theorem 5.  □

Next, we turn our attention to the convergence properties of the entropy rate $H_{R}\left(Y^{\infty }\right)$, which is needed for the characterization of $C^{fb,o}\left(\kappa \right)$ of Definition 4.

**Theorem 6.** 
*Asymptotic properties of entropy rate $H_{R}\left(Y^{\infty }\right)$ of Theorem 3*

*Let $K_{t}^{o},t=1,\ldots ,$ be the solution of the generalized DRE (Equation 180) of the generalized Kalman filter of Theorem 3, corresponding to the time-invariant PO-SS realization of $V^{n}$ of Definition 2, $\left(A_{t},B_{t},C_{t},N_{t},K_{W_{t}}\right)=\left(A,B,C,N,K_{W}\right),\forall t$, with time-invariant strategies $\left(\Lambda_{t},K_{Z_{t}}\right)=\left(\Lambda^{\infty },K_{Z}^{\infty }\right)$, ∀t, generated by*

(258)
$$\begin{array}{cc} \; & K_{t+1}^{o}=AK_{t}^{o}A^{T}+M\left(\Sigma_{t}^{o}\right)K_{{\hat{I}}_{t}^{o}}{\left(M(\Sigma_{t}^{o})\right)}^{T}\\ \; & -\left(AK_{t}^{o}{\left(\Lambda^{\infty }+C\right)}^{T}+M\left(\Sigma_{t}^{o}\right)K_{{\hat{I}}_{t}^{o}}\right){\left(K_{{\hat{I}}_{t}^{o}}+K_{Z}^{\infty }+\left(\Lambda^{\infty }+C\right)K_{t}^{o}{\left(\Lambda^{\infty }+C\right)}^{T}\right)}^{-1}\\ \; & .{\left(AK_{t}^{o}{\left(\Lambda^{\infty }+C\right)}^{T}+M\left(\Sigma_{t}^{o}\right)K_{{\hat{I}}_{t}^{o}}\right)}^{T},\;K_{t}^{o}=K_{t}^{o,T}⪰0,\;t=1,\ldots ,n,\;K_{1}^{o}=0 \end{array}$$

*where*

(259)
$$\begin{array}{cc} \; & K_{{\hat{I}}_{t}^{o}}=C\Sigma_{t}^{o}C^{T}+NK_{W}N^{T},\;\Sigma_{t}^{o}\;is\;a\;solution\;of\;\text{(231)},\;M\left(\Sigma^{o}\right)\;is\;given\;by\;\text{(232)}, \end{array}$$


(260)
$$\begin{array}{cc} \; & F^{CL}\left(\Sigma^{o},K^{o}\right)\overset{▵}{=}A-F\left(\Sigma^{o},K^{o}\right)\left(\Lambda^{\infty }+C\right), \end{array}$$


(261)
$$\begin{array}{cc} \; & F\left(\Sigma^{o},K^{o}\right)\overset{▵}{=}\left(AK^{o}{\left(\Lambda^{\infty }+C\right)}^{T}+M\left(\Sigma^{o}\right)K_{{\hat{I}}^{o}}\right){\left\{K_{{\hat{I}}^{o}}+K_{Z^{\infty }}+\left(\Lambda^{\infty }+C\right)K^{o}{\left(\Lambda^{\infty }+C\right)}^{T}\right\}}^{-1}. \end{array}$$

*Define the corresponding generalized ARE by*

(262)
$$\begin{array}{cc} \; & K^{\infty }=AK^{\infty }A^{T}+M\left(\Sigma^{\infty }\right)K_{{\hat{I}}^{\infty }}{\left(M(\Sigma^{\infty })\right)}^{T}-\left(AK^{\infty }{\left(\Lambda^{\infty }+C\right)}^{T}+M\left(\Sigma^{\infty }\right)K_{{\hat{I}}^{\infty }}\right)(K_{{\hat{I}}^{\infty }}+K_{Z}^{\infty }\\ \; & +\left(\Lambda^{\infty }+C\right)K^{\infty }{\left(\Lambda^{\infty }+C\right)}^{T}{)}^{-1}{\left(AK^{\infty }{\left(\Lambda^{\infty }+C\right)}^{T}+M\left(\Sigma^{\infty }\right)K_{{\hat{I}}^{\infty }}\right)}^{T},\;K^{\infty }=K^{\infty ,T}⪰0. \end{array}$$

*where*

(263)
$$\begin{array}{cc} \; & K_{{\hat{I}}^{\infty }}=C\Sigma^{\infty }C^{T}+NK_{W}N^{T},\;\Sigma^{\infty }\;is\;a\;solution\;of\;\text{(233)},\;M\left(\Sigma^{\infty }\right)\;is\;given\;by\;\text{(232)}. \end{array}$$

*Introduce the matrices*

(264)
$$\begin{array}{cc} \; & C\left(\Lambda^{\infty }\right)\overset{▵}{=}\Lambda^{\infty }+C,\;GQG^{T}\overset{▵}{=}M\left(\Sigma^{\infty }\right)K_{{\hat{I}}^{\infty }}{\left(M(\Sigma^{\infty })\right)}^{T}, \end{array}$$


(265)
$$\begin{array}{cc} \; & GS\overset{▵}{=}M\left(\Sigma^{\infty }\right)K_{{\hat{I}}^{\infty }},\;R\left(K_{Z}^{\infty }\right)\overset{▵}{=}K_{{\hat{I}}^{\infty }}+K_{Z}^{\infty }. \end{array}$$


(266)
$$\begin{array}{cc} \; & ⟹\;\;G\overset{▵}{=}M\left(\Sigma^{\infty }\right),\;Q\overset{▵}{=}K_{{\hat{I}}^{\infty }},\;S\overset{▵}{=}K_{{\hat{I}}^{\infty }}, \end{array}$$


(267)
$$\begin{array}{cc} \; & A^{*}\left(\Lambda^{\infty },K_{Z}^{\infty }\right)\overset{▵}{=}A-M\left(\Sigma^{\infty }\right)K_{{\hat{I}}^{\infty }}{\left(K_{{\hat{I}}^{\infty }}+K_{Z}^{\infty }\right)}^{-1}\left(\Lambda^{\infty }+C\right), \end{array}$$


(268)
$$\begin{array}{cc} \; & B^{*}\left(K_{Z}^{\infty }\right)\overset{▵}{=}K_{{\hat{I}}^{\infty }}-K_{{\hat{I}}^{\infty }}{\left(K_{{\hat{I}}^{\infty }}+K_{Z}^{\infty }\right)}^{-1}K_{{\hat{I}}^{\infty }}. \end{array}$$

*Suppose that the detectability and stabilizability conditions of Corollary 8(i,ii) hold.*

*Then, all statements of Theorem 5 hold with $\left(C\left(\Lambda^{\infty }\right),G,Q,S,R\left(K_{Z}^{\infty }\right)\right)$ as defined by (Equation 265).*

*In particular, suppose*

*(i) $\left\{A,C\left(\Lambda^{\infty }\right)\right\}=\left\{A,\Lambda^{\infty }+C\right\}$ is detectable;*

*(ii) $\left\{A^{*}\left(\Lambda^{\infty },K_{Z}^{\infty }\right),GB^{*,\frac{1}{2}}\left(K_{Z}^{\infty }\right)\right\}$ is stabilizable.*

*Then, any solution $K_{t}^{o},t=1,2,\ldots ,n$ to the generalized matrix DRE (Equation 258) with an arbitrary initial condition $K_{1}^{o}⪰0$ is such that $\lim_{n⟶\infty }K_{n}^{o}=K^{\infty }$, where $K^{\infty }⪰0$ is the unique solution of the generalized matrix ARE (Equation 262) with $spec\left(F^{CL}\left(K^{\infty },\Sigma^{\infty }\right)\right)\in D_{o}$, i.e., it is stabilizing.*

*Moreover, the entropy rate of $Y^{n}$ is given by*

(269)
$$\begin{array}{cc} \; & H_{R}\left(Y^{\infty }\right)=\lim_{n⟶\infty }\frac{1}{n}\sum_{t=1}^{n}H\left(I_{t}^{o}\right) \end{array}$$


(270)
$$\begin{array}{cc} \; & =\lim_{n⟶\infty }\frac{1}{2n}\sum_{t=1}^{n}\log\left(2\pi e\left[\left(\Lambda^{\infty }+C\right)K_{t}^{o}{\left(\Lambda^{\infty }+C\right)}^{T}+K_{{\hat{I}}_{t}^{o}}+K_{Z}^{\infty }\right]\right) \end{array}$$


(271)
$$\begin{array}{cc} \; & =H\left(I_{t}^{\infty }\right)\overset{▵}{=}\frac{1}{2}\log\left(2\pi e\left[\left(\Lambda^{\infty }+C\right)K^{\infty }{\left(\Lambda^{\infty }+C\right)}^{T}+K_{{\hat{I}}^{\infty }}+K_{Z}^{\infty }\right]\right),\;\forall K_{1}^{o}⪰0,\;\forall t \end{array}$$

*where $I_{t}^{o},t=1,\ldots ,n$ is the innovations process of Theorem 3 (with indicated changes of time-invariant strategies) and where*

(272)
$$\begin{array}{cc} \; & I_{t}^{\infty }=\left(\Lambda^{\infty }+C\right)\left({\hat{S}}_{t}^{\infty }-\hat{{\hat{S}}_{t}^{\infty }}\right)+{\hat{I}}_{t}^{\infty }+Z_{t}, \end{array}$$


(273)
$$\begin{array}{cc} \; & I_{t}^{\infty }\in N\left(0,\left(\Lambda^{\infty }+C\right)K^{\infty }{\left(\Lambda^{\infty }+C\right)}^{T}+K_{{\hat{I}}^{\infty }}+K_{Z}^{\infty }\right),\;t=1,2,\ldots , \end{array}$$

*is the stationary Gaussian innovations process, i.e., with $\left(K_{t}^{o},\Sigma_{t}^{o}\right)$ replaced by $\left(K^{\infty },\Sigma^{\infty }\right)$.*



**Proof.** Since the detectability and stabilizability conditions of Corollary 8 hold, then the statements of Corollary 8 hold. By the continuity property of the solutions of generalized difference Riccati equations, with respect to its coefficients (see [16]), and the convergence of the sequence $\lim_{n⟶\infty }\Sigma_{n}^{\infty }=\Sigma^{\infty }$, where $\Sigma^{\infty }⪰0$ is the unique stabilizing solution of (Equation 233), then the statements of Theorem 6 hold, as stated. In particular, under the detectability and stabilizability Conditions (i) and (ii), $\lim_{n⟶\infty }K_{n}^{o}=K^{\infty }$, where $K^{\infty }⪰0$ is the unique and stabilizing solution of (Equation 262).  □

In the next lemma, we apply Theorem 6 to the AR(a,c),a∈(−∞,∞),c∈(−∞,∞) noise of Example 2(a) using Lemma 4.

**Lemma 6.** 
*Consider the AR(a,c),a∈(−∞,∞),c∈(−∞,∞) noise of Example 2(a), and the DRE $\Sigma_{t}^{o}\overset{▵}{=}\Sigma_{t},t=1,\ldots ,n$ and ARE of Lemma 4, (Equation 239)–(Equation 242).*

*Let $K_{t}^{o},t=1,\ldots ,n$ denote the solution of the DRE of Corollary 6(a), when $\Lambda_{t}=\Lambda^{\infty },K_{Z_{t}}=K_{Z}^{\infty },K_{t}=K_{t}^{o},\forall t$, i.e., given by*

$$\begin{array}{cc} K_{t+1}^{o}= & {\left(c\right)}^{2}K_{t}^{o}+{\left(M(\Sigma_{t}^{o})\right)}^{2}K_{{\hat{I}}_{t}^{o}}-{\left(cK_{t}^{o}\left(\Lambda^{\infty }+c-a\right)+M\left(\Sigma_{t}^{o}\right)K_{{\hat{I}}_{t}^{o}}\right)}^{2} \end{array}$$


(274)
$$\begin{array}{cc} \; & \;.{\left(K_{{\hat{I}}_{t}^{o}}+K_{Z}^{\infty }+{\left(\Lambda^{\infty }+c-a\right)}^{2}K_{t}^{o}\right)}^{-1},\;K_{1}^{o}=0,\;t=1,\ldots ,n, \end{array}$$


(275)
$$\begin{array}{cc} K_{Z}^{\infty }\geq & 0,\;K_{t}^{o}\geq 0,\;t=1,\ldots ,n \end{array}$$

*and where*

(276)
$$\begin{array}{cc} \; & M\left(\Sigma_{t}^{o}\right)\overset{▵}{=}\left(c\Sigma_{t}^{o}\left(c-a\right)+K_{W}\right){\left(K_{W}+{\left(c-a\right)}^{2}\Sigma_{t}^{o}\right)}^{-1}, \end{array}$$


(277)
$$\begin{array}{cc} \; & K_{{\hat{I}}_{t}^{o}}={\left(c-a\right)}^{2}\Sigma_{t}^{o}+K_{W},\;t=1,\ldots ,n. \end{array}$$

*Define the set*

(278)
$$\begin{array}{cc} L^{\infty }\overset{▵}{=}\{\left(a,c\right)\in {\left(-\infty ,\infty \right)}^{2}, & a\neq c:\;(i)\;the\;pair\;\{A,C\}=\{a,c-a\}\;is\;detectable,\;and\\ \; & (ii)\;the\;pair\;\left\{A^{*},GB^{*,\frac{1}{2}}\right\}=\left\{a,0\right\}\;is\;stabilizable\}. \end{array}$$

*For any $\left(a,c\right)\in L^{\infty }$, let $K^{\infty }\geq 0$ be a corresponding solution of the ARE (evaluated at $\lim_{n⟶\infty }\Sigma_{n}^{\infty }=\Sigma^{\infty }=0$),*

(279)
$$\begin{array}{cc} K^{\infty }= & {\left(c\right)}^{2}K^{\infty }+K_{W}-{\left(cK^{\infty }\left(\Lambda^{\infty }+c-a\right)+K_{W}\right)}^{2}{\left(K_{W}+K_{Z}^{\infty }+{\left(\Lambda^{\infty }+c-a\right)}^{2}K^{\infty }\right)}^{-1}. \end{array}$$


(280)
$$\begin{array}{cc} K_{Z}^{\infty }\geq & 0,\;K_{W}>0. \end{array}$$

*and define the pairs*

(281)
$$\begin{array}{cc} \; & \{A,C\left(\Lambda^{\infty }\right\}=\left\{c,\Lambda^{\infty }+c-a\right\}, \end{array}$$


(282)
$$\begin{array}{cc} \; & \left\{A^{*}\left(\Lambda^{\infty },K_{Z}^{\infty }\right),GB^{*,\frac{1}{2}}\left(K_{Z}^{\infty }\right)\right\}\\ \; & =\{c-K_{W}{\left(K_{W}+K_{Z}^{\infty }\right)}^{-1}\left(\Lambda^{\infty }+c-a\right),{\left(K_{W}-{\left(K_{W}\right)}^{2}{\left(K_{W}+K_{Z}^{\infty }\right)}^{-1}\right)}^{\frac{1}{2}}\}. \end{array}$$

*Then, the following hold:*

*(1) Suppose $\Lambda^{\infty }+c-a\neq 0$. Then, $\{A,C\left(\Lambda^{\infty }\right\}=\left\{c,\Lambda^{\infty }+c-a\right\}$ is detectable $\forall \left(a,c\right)\in {\left(-\infty ,\infty \right)}^{2}$.*

*(2) Suppose $\Lambda^{\infty }+c-a=0$. Then, $\{A,C\left(\Lambda^{\infty }\right\}=\left\{c,0\right\}$ is detectable for if and only if |c|<1∀a∈(−∞,∞).*

*(3) Suppose $K_{Z}^{\infty }=0$. Then, the pair $\left\{A^{*},GB^{*,\frac{1}{2}}\right\}=\left\{-\Lambda^{\infty }+a,0\right\}$ is unit circle controllable if and only if |Λ−a|≠1∀a∈(−∞,∞).*

*(4) Suppose $K_{Z}^{\infty }=0$. Then, the pair $\left\{A^{*},GB^{*,\frac{1}{2}}\right\}=\left\{-\Lambda^{\infty }+a,0\right\}$ is stabilizable if and only if |Λ−a|<1∀a∈(−∞,∞).*

*(5) Suppose $\Lambda^{\infty }+c-a\neq 0,\left|\Lambda -a\right|\neq 1$$\forall \left(a,c\right)\in {\left(-\infty ,\infty \right)}^{2}$, and $K_{Z}^{\infty }=0$, $\Sigma_{1}^{o}=0$. The sequence $K_{t}^{o},t=1,2,\ldots ,n$ that satisfies the generalized DRE (Equation 274) with a zero initial condition, $K_{1}^{o}=0$, converges to $K^{\infty }\geq 0$, i.e., $\lim_{n⟶\infty }K_{n}^{o}=K^{\infty }$, where $K^{\infty }$ satisfies the generalized ARE,*

(283)
$$\begin{array}{cc} K^{\infty }= & {\left(c\right)}^{2}K^{\infty }+K_{W}-{\left(cK^{\infty }\left(\Lambda^{\infty }+c-a\right)+K_{W}\right)}^{2}{\left(K_{W}+{\left(\Lambda^{\infty }+c-a\right)}^{2}K^{\infty }\right)}^{-1},\;K^{\infty }\geq 0 \end{array}$$

*if and only if |a|<1 (by Lemma 4(4)), and the pair $\left\{A^{*},GB^{*,\frac{1}{2}}\right\}=\left\{-\Lambda^{\infty }+a,0\right\}$ is stabilizable, equivalently, $|\Lambda^{\infty }-a|<1$.*

*Moreover, the solutions of the ARE (Equation 283), under the stabilizability condition, i.e., $|\Lambda^{\infty }-a|<1$, are*

(284)
$$\begin{array}{c} K^{\infty }=\left\{\begin{array}{cc} 0 & the\;unique,\;stabilizing,\;K^{\infty }\geq 0\;sol.\;of\;\text{(283)}\;for\;\left|\Lambda^{\infty }-a\right|<1\\ \frac{K_{W}\left({\left(\Lambda^{\infty }-a\right)}^{2}-1\right)}{{\left(\Lambda^{\infty }+c-a\right)}^{2}}<0 & the\;non\text{-}stabilizing,\;K^{\infty }<0\;sol.\;of\;\text{(283)}\;for\;\left|\Lambda^{\infty }-a\right|<1. \end{array}\right. \end{array}$$

*That is, $\lim_{n⟶\infty }\Sigma_{n}^{o}=\Sigma^{\infty }=0$ is the unique and stabilizing solution $\Sigma^{\infty }\geq 0$ of (Equation 283), i.e., such that $|M^{CL}\left(\Sigma^{\infty }\right)|<1$, if and only if $|\Lambda^{\infty }-a|<1,|a|<1$.*



**Proof.** The statements follow from Lemma 4, Theorem 6 (and general properties of Theorem 5).  □

**Remark 18.** 
*From Lemma 6(5), it follows that if $K_{Z}^{\infty }=0,\Sigma_{1}^{o}=0$, then the unique and stabilizing solution is $K^{\infty }=0$ and corresponds to $|\Lambda^{\infty }-a|<1,a\in \left(-1,1\right)$. This is an application of Theorem 5(1).*


In the next theorem, we characterize the asymptotic limit of Definition 4 by invoking Theorems 3 and 6 and Corollary 8.

**Theorem 7.** 
*Feedback capacity $C^{fb,o}\left(\kappa \right)$ of Theorem 3 for time-invariant strategies*

*Consider $C^{fb,o}\left(\kappa \right)$ of Definition 4 corresponding to Theorem 3, i.e., the PO-SS realization of $V^{n}$ of Definition 2 is time-invariant, $\left(A_{t},B_{t},C_{t},N_{t},K_{W_{t}}\right)=\left(A,B,C,N,K_{W}\right),\forall t$, and the strategies are time-invariant, $\left(\Lambda_{t},K_{Z_{t}}\right)=\left(\Lambda^{\infty },K_{Z}^{\infty }\right),\forall t$.*

*Define the set*

(285)
$$\begin{array}{cc} P^{\infty }\overset{▵}{=} & \{\left(\Lambda^{\infty },K_{Z}^{\infty }\right)\in \left(-\infty ,\infty \right)\times \left[0,\infty \right):\\ \; & (i)\{A,C\}\;of\;Corollary\;8\;is\;detectable;\\ \; & \left(ii\right)\left\{A^{*},GB^{*,\frac{1}{2}}\right\}\;of\;Corollary\;8\;is\;stabilizable,\;where\;\left(A^{*},B^{*}\right)\;is\;defined\;by\;\text{(235)};\\ \; & \left(iii\right)\left\{A,C\left(\Lambda^{\infty }\right)\right\}=\left\{A,\Lambda^{\infty }+C\right\}\;of\;Theorem\;6\;is\;detectable;\\ \; & \left(iv\right)\{A^{*}\left(\Lambda^{\infty },K_{Z}^{\infty }\right),GB^{*,\frac{1}{2}}\left(K_{Z}^{\infty }\right)\}\;of\;Theorem\;6\;is\;stabilizable;\\ \; & \left(A^{*}\left(\Lambda^{\infty },K_{Z}^{\infty }\right),B^{*}\left(K_{Z}^{\infty }\right)\right)is\;defined\;by\;\text{(268)}.\}. \end{array}$$

*Then,*

$$\begin{array}{cc} \; & C^{fb,o}\left(\kappa \right)\overset{▵}{=}\underset{\left(\Lambda^{\infty },K_{Z}^{\infty }\right)\in P^{\infty }:\;\lim_{n⟶\infty }\frac{1}{n}\sum_{t=1}^{n}\left(\Lambda^{\infty }K_{t}^{o}{\left(\Lambda^{\infty }\right)}^{T}+K_{Z}^{\infty }\right)\leq \kappa }{\sup}\{ \end{array}$$


(286)
$$\begin{array}{cc} \; & \lim_{n⟶\infty }\frac{1}{2n}\sum_{t=1}^{n}\log\left(\frac{\left(\Lambda^{\infty }+C\right)K_{t}^{o}{\left(\Lambda^{\infty }+C\right)}^{T}+C\Sigma_{t}^{o}C^{T}+NK_{W}N^{T}+K_{Z}^{\infty }}{C\Sigma_{t}^{o}C^{T}+NK_{W}N^{T}}\right)\} \end{array}$$


(287)
$$\begin{array}{cc} = & \underset{\left(\Lambda^{\infty },K_{Z}^{\infty }\right)\in P^{\infty }\left(\kappa \right)}{\sup}\frac{1}{2}\log\left(\frac{\left(\Lambda^{\infty }+C\right)K^{\infty }{\left(\Lambda^{\infty }+C\right)}^{T}+C\Sigma^{\infty }C^{T}+NK_{W}N^{T}+K_{Z}^{\infty }}{C\Sigma^{\infty }C^{T}+NK_{W}N^{T}}\right) \end{array}$$

*where*

(288)
$$\begin{array}{cc} P^{\infty }\left(\kappa \right)\overset{▵}{=} & \{\left(\Lambda^{\infty },K_{Z}^{\infty }\right)\in P^{\infty }:K_{Z}^{\infty }\geq 0,\;\Lambda^{\infty }K^{\infty }{\left(\Lambda^{\infty }\right)}^{T}+K_{Z}^{\infty }\leq \kappa ,\\ \; & K^{\infty }\;is\;the\;unique\;and\;stabilizing\;solution\;of\;\text{(262)},\;i.e.,\;|F^{CL}\left(\Sigma^{\infty },K^{\infty }\right)|<1\\ \; & \Sigma^{\infty }\;is\;the\;unique,\;stabilizing\;solution\;of\;\text{(233)},\;i.e.,\;|M^{CL}\left(\Sigma^{\infty }\right)|<1,\} \end{array}$$

*provided there exists κ∈[0,∞) such that the set $P^{\infty }\left(\kappa \right)$ is non-empty.*

*Moreover, the maximum element $\left(\Lambda^{\infty },K_{Z}^{\infty }\right)\in P^{\infty }\left(\kappa \right)$ is such that*

*(1) It induces asymptotic stationarity of the corresponding input and innovations processes (see Theorem 3 for specification);*

*(2) If $V^{n},n=1,2,\ldots$ is asymptotically stationary, then it induces asymptotic stationarity of the corresponding, input and output processes;*

*(3) For (i) and (ii), $C^{fb,o}\left(\kappa \right)$ is independent of the initial conditions $K_{1}^{o}⪰0,\Sigma_{1}^{o}⪰0$.*

*Furthermore, if the set $P^{\infty }\left(\kappa \right)$ is empty, replace stabilizability of $\left\{A^{*},GB^{*,\frac{1}{2}}\right\}$*
*and $\left\{A^{*}\left(\Lambda^{\infty },K_{Z}^{\infty }\right),GB^{*,\frac{1}{2}}\left(K_{Z}^{\infty }\right)\right\}$ by unit circle controllablilty, i.e., the maximal and stabilizing solutions $K^{\infty },\Sigma^{\infty }$ of the AREs are utilized.*



**Proof.** By Definition 4, Theorems 3 and 6 and Corollary 8, (Equation 286) follows. We defined the set $P^{\infty }$ using the detectability and stabilizability conditions of Corollary 8 and Theorem 6 to ensure the convergence of solutions $\left\{\left(K_{t}^{o},\Sigma_{t}^{o}\right):t=1,2,\ldots ,n\right\}$ of the generalized matrix DREs to unique non-negative, stabilizing solutions of the corresponding generalized matrix AREs. Then, for any element $\left(\Lambda^{\infty },K_{Z}^{\infty }\right)\in P^{\infty }$, both summands in (Equation 286) converge. This establishes the characterization of the right-hand side of (Equation 287). Parts (1)–(3) follow from the asymptotic properties of the Kalman filter (due to the stabilizability and detectability conditions). The last statement follows due to the relaxation, Theorem 5(4).  □

**Remark 19.** 
*In Theorem 7, if we replace stabilizability by unit circle controllability, then the supremum in (Equation 288) is over a larger set. However, the asymptotic limits $\left(\Sigma^{\infty },K^{\infty }\right)$ are not unique stabilizing solutions but are instead the maximal and stabilizing solutions.*


Theorem 7 also holds for asymptotically time-invariant strategies. We state this as a corollary.

**Corollary 9.** 
*Feedback capacity $C^{fb,o}\left(\kappa \right)$ of Theorem 3 for asymptotically time-invariant strategies*

*Consider the problem statement of Theorem 7 with asymptotically time-invariant strategies, $\lim_{n⟶\infty }\left(\Lambda_{n},K_{Z_{n}}\right)=\left(\Lambda^{\infty },K_{Z}^{\infty }\right)$ and corresponding $K_{n},n=1,2,\ldots$.*

*Then,*

$$\begin{array}{cc} \; & C^{fb,o}\left(\kappa \right)\overset{▵}{=}\underset{\lim_{n⟶\infty }\left(\Lambda_{n},K_{Z_{n}}\right)=\left(\Lambda^{\infty },K_{Z}^{\infty }\right)\in P^{\infty }:\;\lim_{n⟶\infty }\frac{1}{n}\sum_{t=1}^{n}\left(\Lambda_{t}K_{t}{\left(\Lambda_{t}\right)}^{T}+K_{Z_{t}}\right)\leq \kappa }{\sup}\{ \end{array}$$


(289)
$$\begin{array}{cc} \; & \lim_{n⟶\infty }\frac{1}{2n}\sum_{t=1}^{n}\log\left(\frac{\left(\Lambda_{t}+C\right)K_{t}{\left(\Lambda_{t}+C\right)}^{T}+C\Sigma_{t}C^{T}+NK_{W}N^{T}+K_{Z_{t}}}{C\Sigma_{t}C^{T}+NK_{W}N^{T}}\right)\} \end{array}$$


(290)
$$\begin{array}{cc} = & \text{(287)} \end{array}$$

*and the statements of Theorem 7 hold.*



**Proof.** This follows from Proposition 4 and Theorem 7.  □

**Remark 20.** 
*Explicit closed-form expressions of $C^{fb,o}\left(\kappa \right)$ are given in [1,2] for the stable and unstable AR and ARMA noise processes $V^{n}$. The expressions $C^{fb,o}\left(\kappa \right)$ consist of multiple regimes that depend on the parameters of noise, i.e., $\left(c,a,K_{W}\right)$ for the ARMA noise and value of κ. Moreover, for some regimes, $C^{fb,o}\left(\kappa \right)$ is achieved by an optimal $K_{Z}^{\infty }>0$, while for other regimes, it is achieved by $\lim_{n⟶\infty }K_{Z_{n}}=K_{Z}^{\infty }=0$, such that $K_{Z_{1}}>0,K_{Z_{t}}=0,t=2,3,\ldots ,n$.*


Next, we give the expression of feedback capacity $C^{fb}\left(\kappa \right)$, which is generally an upper bound on the expressions of Theorem 7 and Corollary 9.

**Theorem 8.** 
*Feedback capacity $C^{fb}\left(\kappa \right)$ of Theorem 3 for asymptotically time-invariant noise and strategies*

*Consider $C^{fb}\left(\kappa \right)$ of Definition 4 corresponding to Theorem 3, where $(\Lambda_{t},K_{Z_{t}}),t=1,\ldots ,n$ and the coefficients of the PO-SS realization of $V^{n}$ of Definition 2 are asymptotically time-invariant, i.e.,*

(291)
$$\begin{array}{c} \left(a\right)\;\lim_{n⟶\infty }\left(\Lambda_{n},K_{Z_{n}}\right)=\left(\Lambda^{\infty },K_{Z}^{\infty }\right);\;\left(b\right)\lim_{n⟶\infty }\left(A_{n},B_{n},C_{n},N_{n},K_{W_{n}}\right)=\left(A,B,C,N,K_{W}\right). \end{array}$$

*Let $P^{\infty ,ucc}$ correspond to $P^{\infty }=$ (Equation 285) with $\left\{A^{*},GB^{*,\frac{1}{2}}\right\}$ and $\left\{A^{*}\left(\Lambda^{\infty },K_{Z}^{\infty }\right),GB^{*,\frac{1}{2}}\left(K_{Z}^{\infty }\right)\right\}$ being replaced by unit circle controllablity.*

*Let $P^{\infty ,ucc}\left(\kappa \right)$ correspond to $P^{\infty }\left(\kappa \right)=$ (Equation 288), with $K^{\infty }$ and $\Sigma^{\infty }$ being the maximal and stabilizing solution of (Equation 262) and the maximal and stabilizing solution of (Equation 233), respectively.*

*Then,*

$$\begin{array}{cc} \; & C^{fb}\left(\kappa \right)\overset{▵}{=}\lim_{n⟶\infty }\underset{{\left\{\Lambda_{t},K_{Z_{t}}\right\}}_{t=1}^{n}:\lim_{n⟶\infty }\left(\Lambda_{n},K_{Z_{n}}\right)=\left(\Lambda^{\infty },K_{Z}^{\infty }\right)\in P^{\infty ,ucc},\frac{1}{n}\sum_{t=1}^{n}\left(\Lambda_{t}K_{t}{\left(\Lambda_{t}\right)}^{T}+K_{Z_{t}}\right)\leq \kappa }{\sup}\{ \end{array}$$


(292)
$$\begin{array}{cc} \; & \frac{1}{2n}\sum_{t=1}^{n}\log\left(\frac{\left(\Lambda_{t}+C_{t}\right)K_{t}{\left(\Lambda_{t}+C_{t}\right)}^{T}+C_{t}\Sigma_{t}C_{t}^{T}+N_{t}K_{W_{t}}N_{t}^{T}+K_{Z_{t}}}{C_{t}\Sigma_{t}C_{t}^{T}+N_{t}K_{W_{t}}N_{t}^{T}}\right)\} \end{array}$$


(293)
$$\begin{array}{cc} = & \underset{\left(\Lambda^{\infty },K_{Z}^{\infty }\right)\in P^{\infty ,ucc}\left(\kappa \right)}{\sup}\frac{1}{2}\log\left(\frac{\left(\Lambda^{\infty }+C\right)K^{\infty }{\left(\Lambda^{\infty }+C\right)}^{T}+C\Sigma^{\infty }C^{T}+NK_{W}N^{T}+K_{Z}^{\infty }}{C\Sigma^{\infty }C^{T}+NK_{W}N^{T}}\right) \end{array}$$


(294)
$$\begin{array}{cc} \geq & C^{fb,o}\left(\kappa \right). \end{array}$$




**Proof.** First, we note that Theorem 7 continuous to hold, if we consider asymptotically time-invariant strategies and coefficients, i.e., (a) and (b) (by Proposition 4), and the stabilizability conditions are replaced by unit circle controllability conditions. Hence, (Equation 288) remains valid with the set $P^{\infty }\left(\kappa \right)$ replaced by the larger set $P^{\infty ,ucc}\left(\kappa \right)$, giving the higher value (Equation 293) ≥ (Equation 288). For the derivation of (Equation 292) = (Equation 293), it suffices to show we can interchange the $\lim_{n⟶\infty }$ and the supremum in (Equation 292). This can be completed by using the definition of the set $P^{\infty ,ucc}\left(\kappa \right)$ and Conditions (a) and (b). The procedure, although lengthy, is similar to the one described in [41]; hence, we omit it.  □

**Conclusion 1.** 
*Degenerate versions of Theorem 7, Theorem 8 for feedback code of Definition 3, i.e., $\left(s,2^{nR},n\right)$, n=1,2,…*

*The characterizations of feedback capacity $C^{fb,o}\left(\kappa ,s\right)$ of the AGN channel (Equation 1) driven by a noise $V^{n}$ of Definition 2, for the code of Definition 3, i.e., $\left(s,2^{nR},n\right)$, n=1,2,…, are degenerate cases of Theorem 7 and Theorem 8, corresponding to $\Sigma_{t}=\Sigma_{t}^{s},t=1,\ldots ,\Sigma_{1}=\Sigma_{1}^{s}=0$. In particular, since Theorem 7 characterizes $C^{fb,o}\left(\kappa \right)$ for all initial data $\Sigma_{1}⪰0$, then it includes $\Sigma_{1}=\Sigma_{1}^{s}=0$. Moreover, it follows that $C^{fb,o}\left(\kappa \right)=C^{fb,o}\left(\kappa ,s\right)$, where $C^{fb,o}\left(\kappa ,s\right)$ is independent of the initial state $S_{1}^{s}=s$.*



We apply Theorem 7 to obtain $C^{fb,o}\left(\kappa \right)$ of the AR(a,c),a∈(−∞,∞),c∈(−∞,∞) noise.

**Corollary 10.** 
*Consider the AR(a,c),a∈(−∞,∞),c∈(−∞,∞) noise of Example 2(a).*

*Define the set*

(295)
$$\begin{array}{cc} P^{\infty }\overset{▵}{=} & \{\left(\Lambda^{\infty },K_{Z}^{\infty }\right)\in \left(-\infty ,\infty \right)\times \left[0,\infty \right):\\ \; & (i)c\in (-\infty ,\infty ),a\in (-1,1),c\neq a,\\ \; & \left(ii\right)\;the\;pair\;\left\{A,C\left(\Lambda^{\infty }\right)\right\}\overset{▵}{=}\left\{c,\Lambda^{\infty }+c-a\right\}\;is\;detectable,\\ \; & \left(ii\right)\;the\;pair\;\{A^{*}\left(\Lambda^{\infty },K_{Z}^{\infty }\right),GB^{*,\frac{1}{2}}\left(K_{Z}^{\infty }\right)\}\;is\;stabilizable,\;where\\ \; & A^{*}\left(\Lambda^{\infty },K_{Z}^{\infty }\right)\overset{▵}{=}\;c-K_{W}{\left(K_{W}+K_{Z}^{\infty }\right)}^{-1}\left(\Lambda^{\infty }+c-a\right),\\ \; & GB^{*,\frac{1}{2}}\left(K_{Z}^{\infty }\right)\overset{▵}{=}{\left(K_{W}-{\left(K_{W}\right)}^{2}{\left(K_{W}+K_{Z}^{\infty }\right)}^{-1}\right)}^{\frac{1}{2}}\}. \end{array}$$

*Then,*

(296)
$$\begin{array}{c} C^{fb,o}\left(\kappa \right)=\underset{\left(\Lambda^{\infty },K_{Z}^{\infty }\right)\in P^{\infty }\left(\kappa \right)}{\sup}\frac{1}{2}\log\left(\frac{{\left(\Lambda^{\infty }+c-a\right)}^{2}K^{\infty }+K_{W}+K_{Z}^{\infty }}{K_{W}}\right)=C^{fb,o}\left(\kappa ,s\right),\;\forall s \end{array}$$

*where*

(297)
$$\begin{array}{cc} P^{\infty } & \left(\kappa \right)\overset{▵}{=}\{\left(\Lambda^{\infty },K_{Z}^{\infty }\right)\in P^{\infty }:\;{\left(\Lambda^{\infty }\right)}^{2}K^{\infty }+K_{Z}^{\infty }\leq \kappa ,\\ \; & K^{\infty }\geq 0\;is\;the\;unique\;and\;stabilizing\;solution\;of\\ \; & K^{\infty }={\left(c\right)}^{2}K^{\infty }+K_{W}\\ \; & -{\left(cK^{\infty }\left(\Lambda^{\infty }+c-a\right)+K_{W}\right)}^{2}{\left(K_{W}+K_{Z}^{\infty }+{\left(\Lambda^{\infty }+c-a\right)}^{2}K^{\infty }\right)}^{-1}\} \end{array}$$

*provided that there exists κ∈[0,∞) such that the set $P^{\infty }\left(\kappa \right)$ is non-empty.*

*Moreover, the maximum element $\left(\Lambda^{\infty },K_{Z}^{\infty }\right)\in P^{\infty }\left(\kappa \right)$, is such that,*

*(1) It induces asymptotic stationarity of the corresponding, input and innovations processes;*

*(2) If $V^{n},n=1,2,\ldots$ is asymptotically stationary, then it induces asymptotic stationarity of the corresponding, input and output processes;*

*(3) For (i) and (ii), $C^{fb,o}\left(\kappa \right)$ and $C^{fb,o}\left(\kappa ,s\right)$ are independent of $\Sigma_{1}\geq 0$ and s, respectively, and the following identities hold.*

(298)
$$\begin{array}{c} C^{fb,o}\left(\kappa \right)=C^{fb,o}\left(\kappa ,s\right)=C^{fb,S,o}\left(\kappa ,s\right),\;\forall s \end{array}$$

*Furthermore, if the set $P^{\infty }\left(\kappa \right)$ is empty, replace stabilizability of $\left\{A^{*},GB^{*,\frac{1}{2}}\right\}$ by unit circle controllability, i.e., so that $K^{\infty }$ is the maximal and stabilizing solutions of the ARE.*



**Proof.** The first part is an application of Theorem 7, Lemmas 4 and 6. Parts (1)–(3) are due to the convergence properties of the Kalman filter (due to the stabilizability and detectability conditions). It remains to show (Equation 298). The equality $C^{fb,o}\left(\kappa \right)=C^{fb,o}\left(\kappa ,s\right),\forall s$ holds by Conclusion 1(a). The last equality holds due to the AR(a,c),a∈(−∞,∞),c∈(−∞,∞) noise. If the initial state $S_{1}=S_{1}^{s}=s$ is known to the encoder and the decoder, then Condition 1 of Section 1.1 holds. In addition, Condition 2 also holds, as can be easily verified from Equations (Equation 121) and (Equation 122).  □

**Remark 21.** 
*From Corollary 10, we obtain the degenerate cases, AR(c),c∈(−∞,∞), noise, i.e., setting a=0. The various implications of the detectability and stabilizability conditions for the AR(c),c∈(−∞,∞) noise are found in [1]. The corresponding $C^{fb,o}\left(\kappa ,s\right)$ states that for stable AR(c) and time-invariant strategies, feedback does not increase capacity (because of the stronger condition of stabilizabily). However, if unit circle controllability is imposed instead, then feedback increases capacity.*


## 4. Sequential Characterization of *n*–FTFI Capacity for Case (II) Formulation

In this section, we consider the Case (II) formulation, and we derive the characterization of feedback capacity, $C_{n}^{fb,S}\left(\kappa ,s\right)$, of the AGN channel (Equation 1) driven by a noise $V^{n}$ of Definition 2, i.e., for the code of Definition 3, $\left(s,2^{nR},n\right)$, n=1,2,…, when Conditions 1 and 2 of Section 1.1 hold.

**Definition 8.** 
*AGN channels driven by noise with invertible PO-SS realizations*

*The PO-SS realization of the noise of Definition 2 is called invertible if it satisfies the following condition:*

*(A1) Given the initial state $S_{1}=S_{1}^{s}=s$, the noise $V^{t-1}$ uniquely specifies the state $S^{t}$, for t=1,…,n, and vice versa.*



**Corollary 11.** 
*Characterization of n–FTFI Capacity for the Case (II) formulation*

*Consider the AGN channel (Equation 1) driven by a noise $V^{n}$ of Definition 8, and the code of Definition 3, $\left(s,2^{nR},n\right)$, n=1,2,…, i.e., Conditions 1 and 2 of Section 1.1 hold.*

*Define the n–FTFI capacity for a fixed initial state $S_{1}=S_{1}^{s}=s$ by*

(299)
$$\begin{array}{c} C_{n}^{fb,S}\left(\kappa ,s\right)=\underset{P_{\left[1,n\right]}^{s}\left(\kappa \right)}{\sup}H^{P}\left(Y^{n}|s\right)-H\left(V^{n}|s\right) \end{array}$$

*where the set $P_{\left[1,n\right]}^{s}\left(\kappa \right)$ is defined by*

(300)
$$\begin{array}{c} P_{\left[1,n\right]}^{s}\left(\kappa \right)\overset{▵}{=}\left\{P_{t}\left(dx_{t}|x^{t-1},y^{t-1},s\right),t=1,\ldots ,n:\frac{1}{n}E_{s}^{P}\left(\sum_{t=1}^{n}{\left(X_{t}\right)}^{2}\right)\leq \kappa \right\} \end{array}$$

*and where $E_{s}^{P}$ means $S_{1}=S_{1}^{s}=s$ is fixed, and the joint distribution depends on the elements of $P_{\left[1,n\right]}^{s}\left(\kappa \right)$.*

*Then, the following hold:*

*(a) The n–FTFI capacity, for a fixed $S_{1}=s$, is characterized by*

(301)
$$\begin{array}{cc} \; & C_{n}^{fb,S}\left(\kappa ,s\right)=\underset{{\bar{P}}_{\left[1,n\right]}^{s,M}\left(\kappa \right)}{\sup}\sum_{t=1}^{n}\left\{H^{{\bar{P}}^{M}}\left(Y_{t}|Y^{t-1},s\right)-H\left(V_{t}|V^{t-1},s\right)\right\} \end{array}$$


(302)
$$\begin{array}{cc} = & \underset{{\bar{P}}_{\left[1,n\right]}^{s,M}\left(\kappa \right)}{\sup}\sum_{t=1}^{n}\left\{H^{{\bar{P}}^{M}}\left(Y_{t}-E^{{\bar{P}}^{M}}\left\{Y_{t}\left|\right|Y^{t-1},s\right\}|s\right)-H\left(V_{t}-E\left\{V_{t}|V_{t}^{t-1},s\right\}|s\right)\right\} \end{array}$$

*where the ${\bar{P}}_{\left[1,n\right]}^{s,M}\left(\kappa \right)$ is defined by*

(303)
$$\begin{array}{cc} \; & {\bar{P}}_{\left[1,n\right]}^{s,M}\left(\kappa \right)\overset{▵}{=}\left\{{\bar{P}}_{t}^{M}\left(dx_{t}|s_{t},y^{t-1},s\right),t=1,\ldots ,n:\frac{1}{n+1}E_{s}^{{\bar{P}}^{M}}\left(\sum_{t=1}^{n}{\left(X_{t}\right)}^{2}\right)\leq \kappa \right\} \end{array}$$

*and where (Equation 18) is respected, ${\bar{P}}_{t}^{M}\left(dx_{t}|s_{t},y^{t-1},s\right)$, is conditionally Gaussian, with linear conditional mean and nonrandom conditional covariance, given by (the notation $S_{t}=S_{t}^{s},t=2,\ldots ,n$ means this sequence is generated from (Equation 19), when the initial state is fixed, $S_{1}=S_{1}^{s}=s$).*

(304)
$$\begin{array}{cc} \; & E^{{\bar{P}}^{M}}\left\{X_{t}|S_{t}^{s},Y^{t-1},S_{1}^{s}=s\right\}=\left\{\begin{array}{cc} \Lambda_{t}\left(S_{t}^{s}-E^{{\bar{P}}^{M}}\left\{S_{t}^{s}|Y^{t-1},S_{1}^{s}=s\right\}\right), & t=2,\ldots ,n\\ 0, & t=1, \end{array}\right. \end{array}$$


(305)
$$\begin{array}{cc} \; & K_{X_{t}|S_{t}^{s},Y^{t-1},S_{1}^{s}=s}\overset{▵}{=}cov\left(X_{t},X_{t}|S_{t}^{s},Y^{t-1},S_{1}^{s}=s\right)=K_{Z_{t}}⪰0,\;t=1,\ldots ,n. \end{array}$$

*and $H^{\bar{P}}\left(Y_{t}|Y^{t-1},s\right)$ is evaluated with respect to the probability distribution $P_{t}^{{\bar{P}}^{M}}\left(dy_{t}|y^{t-1},s\right)$, defined by*

(306)
$$\begin{array}{c} P_{t}^{{\bar{P}}^{M}}\left(dy_{t}|y^{t-1},s\right)=\int P_{t}\left(dy_{t}|x_{t},s_{t}\right)\;P_{t}^{{\bar{P}}^{M}}\left(dx_{t}|s_{t},y^{t-1},s\right)\;P_{t}^{{\bar{P}}^{M}}\left(ds_{t}|y^{t-1},s\right),\;t=1,\ldots ,n. \end{array}$$

*(b) Define the conditional means and conditional covariance for a fixed $S_{1}=S_{1}^{s}=s$ by*

(307)
$$\begin{array}{cc} K_{t}^{s}\overset{▵}{=} & cov(S_{t}^{s},S_{t}^{s}|Y^{t-1},S_{1}^{s}=s)=E_{s}^{{\bar{P}}^{M}}\left\{\left(S_{t}^{s}-{\hat{S}}_{t}^{s}\right){\left(S_{t}^{s}-{\hat{S}}_{t}^{s}\right)}^{T}\right\}, \end{array}$$


(308)
$$\begin{array}{cc} {\hat{S}}_{t}^{s}\overset{▵}{=} & E_{s}^{{\bar{P}}^{M}}\left\{S_{t}^{s}|Y^{t-1},S_{1}^{s}=s\right\},\;t=2,\ldots ,n,\;K_{1}^{s}\overset{▵}{=}cov(S_{1}^{s},S_{1}^{s}\left|S_{1}^{s}=s\right)=0,\;{\hat{S}}_{1}^{s}\overset{▵}{=}s. \end{array}$$

*The optimal channel input distribution of part (a) is induced by a jointly Gaussian process $X^{n}$, with a realization given by*

(309)
$$\begin{array}{cc} \; & X_{t}=\Lambda_{t}\left(S_{t}^{s}-{\hat{S}}_{t}^{s}\right)+Z_{t},\;X_{1}=Z_{1},\;t=2,\ldots ,n, \end{array}$$


(310)
$$\begin{array}{cc} \; & Z_{t}\in N\left(0,K_{Z_{t}}\right)\;independent\;of\;\left(S_{1},X^{t-1},V^{t-1},Y^{t-1}\right),\;t=1,\ldots ,n, \end{array}$$


(311)
$$\begin{array}{cc} \; & Y_{t}=\Lambda_{t}\left(S_{t}^{s}-{\hat{S}}_{t}^{s}\right)+Z_{t}+V_{t},\;t=1,\ldots ,n, \end{array}$$


(312)
$$\begin{array}{cc} \; & \;\;=\Lambda_{t}\left(S_{t}^{s}-{\hat{S}}_{t}^{s}\right)+C_{t}S_{t}^{s}+N_{t}W_{t}+Z_{t}, \end{array}$$


(313)
$$\begin{array}{cc} \; & \frac{1}{n}E_{s}^{{\bar{P}}^{M}}\left\{\sum_{t=1}^{n}{\left(X_{t}\right)}^{2}\right\}=\frac{1}{n}\sum_{t=1}^{n}\left(\Lambda_{t}K_{t}^{s}\Lambda_{t}^{T}+K_{Z_{t}}\right)\leq \kappa \end{array}$$

*where Λt is nonrandom.*

*The conditional means and conditional covariance ${\hat{S}}_{t}^{s}$ and $K_{t}^{s}$ are given by the generalized Kalman filter, as follows:*

*(i) ${\hat{S}}_{t}^{s}$ satisfies the Kalman filter recursion*

(314)
$$\begin{array}{cc} \; & {\hat{S}}_{t+1}^{s}=A_{t}{\hat{S}}_{t}^{s}+F_{t}\left(K_{t}^{s}\right)I_{t}^{s},\;{\hat{S}}_{1}=s,\\ \; & F_{t}\left(K_{t}^{s}\right)\overset{▵}{=}\left(A_{t}K_{t}^{s}{\left(\Lambda_{t}+C_{t}\right)}^{T}+B_{t}K_{W_{t}}N_{t}^{T}\right) \end{array}$$


(315)
$$\begin{array}{cc} \; & .{\left\{N_{t}K_{W_{t}}N_{t}^{T}+K_{Z_{t}}+\left(\Lambda_{t}+C_{t}\right)K_{t}^{s}{\left(\Lambda_{t}+C_{t}\right)}^{T}\right\}}^{-1}, \end{array}$$


(316)
$$\begin{array}{cc} \; & I_{t}^{s}\overset{▵}{=}Y_{t}-C_{t}{\hat{S}}_{t}^{s}=\left(\Lambda_{t}+C_{t}\right)\left(S_{t}^{s}-{\hat{S}}_{t}^{s}\right)+N_{t}W_{t}+Z_{t},\;t=1,\ldots ,n, \end{array}$$


(317)
$$\begin{array}{cc} \; & I_{t}^{s}\in N\left(0,K_{I_{t}^{s}}\right),\;t=1,\ldots ,n\;an\;orthogonal\;innovations\;process,\;i.e.,\\ \; & \;\;I_{t}^{s}\;is\;independent\;of\;I_{k}^{s},\;for\;all\;t\neq k,\;and\;I_{t}^{s}\;is\;independent\;of\;Y^{t-1}, \end{array}$$


(318)
$$\begin{array}{cc} \; & K_{Y_{t}|Y^{t-1},s}=K_{I_{t}^{s}}\overset{▵}{=}cov\left(I_{t},I_{t}|S_{1}^{s}=s\right)\\ \; & =\left(\Lambda_{t}+C_{t}\right)K_{t}^{s}{\left(\Lambda_{t}+C_{t}\right)}^{T}+N_{t}K_{W_{t}}N_{t}^{T}+K_{Z_{t}}. \end{array}$$


*(ii) The error $E_{t}^{s}\overset{▵}{=}S_{t}^{s}-{\hat{S}}_{t}^{s}$ satisfies the recursion*

(319)
$$\begin{array}{cc} E_{t+1}^{s}= & F_{t}^{CL}\left(K_{t}^{s}\right)E_{t}^{s}+\left(B_{t}-F_{t}\left(K_{t}^{s}\right)N_{t}\right)W_{t}-F_{t}\left(K_{t}^{s}\right)Z_{t},\;E_{1}^{s}=S_{1}^{s}-{\hat{S}}_{1}^{s}=0,\;t=1,\ldots ,n, \end{array}$$


(320)
$$\begin{array}{cc} F_{t}^{CL}( & K_{t}^{s})\overset{▵}{=}A_{t}-F_{t}\left(K_{t}^{s}\right)\left(\Lambda_{t}+C_{t}\right). \end{array}$$

*(iii) $K_{t}^{s}=E\left\{E_{t}^{s}{\left(E_{t}^{s}\right)}^{T}\right\}$ satisfies the generalized DRE*

(321)
$$\begin{array}{cc} \; & K_{t+1}^{s}=A_{t}K_{t}A_{t}^{T}+B_{t}K_{W_{t}}B_{t}^{T}-\left(B_{t}K_{W_{t}}N_{t}^{T}+A_{t}K_{t}^{s}{\left(\Lambda_{t}+C_{t}\right)}^{T}\right)\{N_{t}K_{W_{t}}N_{t}^{T}+K_{Z_{t}}\\ \; & +\left(\Lambda_{t}+C_{t}\right)K_{t}^{s}{\left(\Lambda_{t}+C_{t}\right)}^{T}{\}}^{-1}{\left(B_{t}K_{W_{t}}N_{t}^{T}+A_{t}K_{t}^{s}\left(\Lambda_{t}+C_{t}\right)\right)}^{T},\;K_{t}^{s}⪰0,\;K_{1}^{s}=0,\;t=1,\ldots ,n. \end{array}$$

*(c) The characterization of the n–FTFI capacity of part (a) is*

(322)
$$\begin{array}{cc} \; & C_{n}^{fb,S}\left(\kappa ,s\right)=\underset{\left(\Lambda_{t},K_{Z_{t}}\right),t=1,\ldots ,n:\;\frac{1}{n}E_{s}\left\{\sum_{t=1}^{n}{\left(X_{t}\right)}^{2}\right\}\leq \kappa }{\sup}\sum_{t=1}^{n}\log\frac{K_{Y_{t}|Y^{t-1},s}}{K_{V_{t}|V^{t-1},s}} \end{array}$$


(323)
$$\begin{array}{cc} \; & =\underset{\left(\Lambda_{t},K_{Z_{t}}\right),t=1,\ldots ,n:\;\frac{1}{n}E_{s}\left\{\sum_{t=1}^{n}{\left(X_{t}\right)}^{2}\right\}\leq \kappa }{\sup}\sum_{t=1}^{n}\left\{H\left(I_{t}^{s}\right)-H\left(N_{t}W_{t}\right)\right\} \end{array}$$


(324)
$$\begin{array}{cc} \; & =\underset{\left(\Lambda_{t},K_{Z_{t}}\right),t=1,\ldots ,n:\;\frac{1}{n}\sum_{t=1}^{n}\left(\Lambda_{t}K_{t}^{s}\Lambda_{t}^{T}+K_{Z_{t}}\right)\leq \kappa }{\sup}\{\\ \; & \frac{1}{2}\sum_{t=1}^{n}\log\left(\frac{\left(\Lambda_{t}+C_{t}\right)K_{t}^{s}{\left(\Lambda_{t}+C_{t}\right)}^{T}+N_{t}K_{W_{t}}N_{t}^{T}+K_{Z_{t}}}{N_{t}K_{W_{t}}N_{t}^{T}}\right)\}. \end{array}$$

*and the statements of part (b) hold.*



**Proof.** See Section A.8.  □

**Remark 22.** 
*The asymptotic analysis of Section 3, based on Definition 4, applies naturally to Corollary 11.*


**Corollary 12.** 
*Characterization of Feedback Capacity for the Case (II) formulation*

*Consider the statement of Corollary 11 for asymptotically time-invariant noise. The feedback capacity is given by*

(325)
$$\begin{array}{cc} \; & C^{fb,S}\left(\kappa ,s\right)=\lim_{n⟶\infty }\frac{1}{n}C_{n}^{fb,S}\left(\kappa ,s\right) \end{array}$$


(326)
$$\begin{array}{cc} \; & =\underset{\left(\Lambda^{\infty },K_{Z}^{\infty }\right)\in P^{\infty ,ucc,S}\left(\kappa \right)}{\sup}\frac{1}{2}\log\left(\frac{\left(\Lambda^{\infty }+C\right)K^{\infty }{\left(\Lambda^{\infty }+C\right)}^{T}+NK_{W}N^{T}+K_{Z}^{\infty }}{NK_{W}N^{T}}\right), \end{array}$$


(327)
$$\begin{array}{cc} \; & P^{\infty ,ucc,S}\left(\kappa \right)\overset{▵}{=}\{\left(\Lambda^{\infty },K_{Z}^{\infty }\right):K_{Z}^{\infty }\geq 0,\;\Lambda^{\infty }K^{\infty }{\left(\Lambda^{\infty }\right)}^{T}+K_{Z}^{\infty }\leq \kappa ,\;K^{\infty }\;is\;the\;maximal\;and\\ \; & stabilizing\;solutionof\;\text{(328)},\;i.e.,\;detectability\;and\;unit\;circle\;controllability\;hold\}, \end{array}$$


(328)
$$\begin{array}{cc} \; & K^{\infty }=AK^{\infty }A^{T}+BK_{W}B^{T}-\left(BK_{W}N^{T}+AK^{\infty }{\left(\Lambda^{\infty }+C\right)}^{T}\right)\{NK_{W}N^{T}+K_{Z}^{\infty }\\ \; & +\left(\Lambda^{\infty }+C\right)K^{\infty }{\left(\Lambda^{\infty }+C\right)}^{T}{\}}^{-1}{\left(BK_{W}N^{T}+AK^{\infty }\left(\Lambda^{\infty }+C\right)\right)}^{T}. \end{array}$$


*If the optimal solution is such that the stabilizability holds, then the feedback capacity is independent of the initial state $S_{1}=s$.*



**Proof.** This is a special case of Theorem 7, with $\Sigma^{\infty }=0$.  □

**Corollary 13.** 
*Feedback capacity of [4,7]*

*Consider the channels studied in [4,7], i.e., with time-invariant and stable realization.*

*(a) The time-domain n–FTFI capacity is $C_{n}^{fb,S}\left(\kappa ,s\right)$ given in Corollary 11.*

*(b) The time-domain feedback capacity is given by Corollary 12.*



**Proof.** Since the noise in [4,7] is time-invariant and stable, according to Definition 8, and the code is that of Definition 3, i.e., $\left(s,2^{nR},n\right)$, n=1,2,…, then their results are special cases of Corollaries 11 and 12.  □

In the next remark, we clarify the relation of Corollary 11 and the analysis of [4,8].

**Remark 23.** 
*Relations of Corollary 11 and [4,8]*

*(a) The state space problem analyzed in [8] is precisely $C_{n}^{fb,S}\left(\kappa ,s\right)$, when the noise is stationary and Gaussian, i.e., it corresponds to the Case (II) formulation. Corollary 11 is derived in [8] for the degenerate case of a time-invariant realization of the noise $V^{n}$, i.e., of Definition 8. However, the asymptotic analysis of [8] (Section VI) should be read with caution, because it did not impose the necessary and/or sufficient conditions for convergence of the sequence $K_{t}^{s},t=1,2,\ldots$ generated by the time-invariant version of the generalized DRE (Equation 321), i.e., $\lim_{n⟶\infty }K_{n}^{s}=K^{\infty }⪰0$, where $K^{\infty }⪰0$ is either the maximal or the unique and stabilizing solution of a corresponding generalized ARE.*

*(b) The problem analyzed [4] that led to [4] (Theorem 6.1, $C_{FB}$) is the per unit time limit of $C_{n}^{fb,S}\left(\kappa ,s\right)$, when the noise is stationary, two-sided or one-sided (asymptotically stationary) and Gaussian, i.e., it corresponds to the Case (II) formulation. The characterization of feedback capacity presented in [4] (Theorem 6.1, $C_{FB}$) presupposed that the following hold ((i)–(iii) are also assumed in [8] (Section VI)):*

*(i) The feedback code is Definition 3, i.e., $\left(s,2^{nR},n\right)$.*

*(ii) The noise is time-invariant and stable, and the PO-SS realization of the noise is invertible, as presented in Definition 8.*

*(iii) The definition of rate is $C^{fb,S,o}\left(\kappa ,s\right)$, with supremum and per unit time limit interchanged, and the supremum taken over using time-invariant channel input distributions.*

*(iv) The innovations covariance of the channel input process is asymptotically zero, i.e., $\lim_{n⟶\infty }K_{Z_{n}}=K_{Z}^{\infty }=0$. This implies the corresponding $\lim_{n⟶\infty }K_{n}^{s}=K^{\infty }\geq 0$ is the maximal and stabilizing solution of the corresponding matrix ARE, since detectability and unit circle controllability conditions hold, but not the stabilizability condition.*

*Items (i)–(iv) are confirmed from [4] (Lemma 6.1) (and comments above), which is used to derive [4] (Theorem 6.1, $C_{FB}$).*

*However, the characterization of feedback capacity in [4] (Theorem 6.1, $C_{FB}$) should be read with caution, because the stabilizability condition is violated, due to the requirement of the author, which states that $K_{Z}^{\infty }=0$ is optimal. By Theorem 5(1), for the choice $K_{Z}^{\infty }=0$, the only choice is the maximal and stabilizing solution of the generalized ARE presented in [4] (Theorem 6.1, $C_{FB}$).*

*However, it is easy to verify that [4] (Theorem 6.1, C_FB_) cannot be the capacity of asymptotically stationary noise because $C_{FB}$ depends on the covariance $K_{S_{1}}≻0$. Moreover, $K_{1}≻0$ is required.*

*Finally, we emphasize that the treatment of the ARMA noise in [2] clarifies the above issues.*



## 5. Conclusions

New equivalent sequential characterizations of Cover and Pombra [5] “*n*–block” feedback capacity formulas are derived using time-domain methods for additive Gaussian noise (AGN) channels driven by non-stationary Gaussian noise. New features of the equivalent characterizations encompass the representation of the optimal channel input process by a *sufficient statistic and Gaussian orthogonal innovations process*. The sequential characterizations of the *n*–block feedback capacity formula are expressed as a functional of two generalized matrix difference Riccati equations (DREs) of the filtering theory of Gaussian systems. The asymptotic analysis of the per unit time limit of the *n*–block”, called feedback capacity, is also presented for time-invariant and asymptotically time-invariant channel input distributions, using the tools from the theory of generalized matrix Riccati equations.

Prior analysis and characterizations of feedback capacity follows on from the analysis and derivation of the new sequential characterizations of feedback capacity, such as [4,7,11,12], who do not address the Cover and Pombra [5] feedback capacity problem, as the code definitions and noise assumptions in [4,7,11,12] (even under the restriction of stationary noise) are fundamentally different from those in [5]. This paper clarifies several of these points of confusion.

## Data Availability

No data are contained within this article.

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
