# Peer review of "New Formulas of Feedback Capacity for AGN Channels with Memory: A Time-Domain Sufficient Statistic Approach [Author-notes fn1-entropy-27-00207]"

_entropy, 2025, doi:10.3390/e27020207_

Round 1
Reviewer 1 Report
Comments and Suggestions for Authors
This paper studies feedback capacity for AGN channels under two formulations depending on how the initial states of the noise and the state space model of the noise sequence are specified.
The paper has made several contributions.
First, it addresses an existing issue which stems from ref. [4] and has been discussed by other authors in follow-up research. Notably, [4] does not correspond to the Cover and Pombra characterization of the n-block or transmission feedback capacity formula and its asymptotic limit while the gap in the proof of [4] has been pointed out by other researchers. Due to the importance of feedback capacity in the literature of information theory, the analysis in the paper is useful and provides clarification of the situation centering the problem in ref [4].
In the further asymptotic analysis of feedback capacity, the authors provide their characterization in terms of two algebraic Riccati equations.
Overall, the paper has made interesting contributions in settling a problem arising from ref [4] and some other papers. The asymptotic analysis in the paper is elegant and useful.
The paper is acceptable provided the authors can make some minor revisions.
The discussions on ref [4] have been scattered in various parts of the paper. One may ask what extra conditions are needed for Theorems 4.1 and 6.1, both from [4], to hold? Or if the two theorems hold but the proofs are wrong, can they now follow as corollary of certain theorems in the current paper?
In the asymptotic analysis, the inequality in (1.1) is replaced by its limiting form. What happens if one has a hard constraint like E[X_t^2]\le kappa for all t?
Author Response
The authors wish to thank the reviewer for the encouraging comments and the suggestions.
Comment 1: The discussions on ref [4] have been scattered in various parts of the paper. One may ask what extra conditions are needed for Theorems 4.1 and 6.1, both from [4], to hold? Or if the two theorems hold but the proofs are wrong, can they now follow as corollary of certain theorems in the current paper?
Response 1: We further clarified that Theorem 6.1 in ref.[4] addresses the capacity for the code of Definition 3, under Condition 2, page 6. We include Remark 1 page 9 to further explain the issues.
The final conclusion is found Corollary 13, page 50. This is an independent prove, based on the material of Section 4. This is a special case of Corollary 12, that deals with Case II) for stable and unstable noise. Theorem 6.1 in ref.[4] assumes the innovation part of the input, i.e., $K_Z^\infty=0$. There is no prove to support this in ref.[6], for general multidimensional state variable. Rather, one needs to solve the optimization problem of corollary 13. For ARMA noise we solved the optimization problem, and verified for stable noise $K_Z^\infty=0$ but for unstable noise$K_Z^\infty > 0$ (see ref.[2]).
Actually, Theorem 6.1 was due to Theorem 4.1 that claimed the power spectral density of the innovations part of the input is zero. But this claimed has gaps (as identified in ref.[6]).
Theorem 4,1 is not discussed in our paper, because it deals with frequency-domain methods. However, based on our unpublished work, Theorem 4.1 is incorrect, and cannot be fixed.
Comment 2: In the asymptotic analysis, the inequality in (1.1) is replaced by its limiting form. What happens if one has a hard constraint like E[X_t^2]\le kappa for all t?
Response 2: In the asymptotic the hard constraint will converge to the same constraint of our asymptotic equations. Only the $n-$FTFI capacities will be different, but in the limit they are identical.
Reviewer 2 Report
Comments and Suggestions for Authors
This paper presents new results on the characterization of feedback capacity for memory additive Gaussian noise channels, under the formulation that the noise process is partially observable, abiding by the original setup of Cover-Pombra. Adopting an innovation approach, a time-domain characterization of finite-length feedback capacity is derived, expressed as a functional of two generalized matrix difference Riccati equations. Subsequently, via convergence properties of these equations, existence of asymptotic limit of finite-length feedback capacity is studied, with necessary and/or sufficient conditions identified.
This paper is lengthy with heavy notations. This is understandable given the large number of stochastic processes and matrices involved. Furthermore, since the authors also attempted to comment on prior works from time to time, making clarifications and drawing connections, the content of this paper becomes even more difficult to read. Given the very limited review turnaround time, it is impossible to check the mathematical derivations in detail. Assuming that there is no basic technical error, the reviewer feels that this paper has its merit and can be accepted for publication.
Some minor comments:
- Def. 2, why both (1.19) and (1.20) involve the same noise W_t? In state space models it is usually the case where the state noise and the observation noise are independent.
- In (1.28) justification of (a), the orthogonality principle itself is insufficient to eliminate the conditioning. Since everything is jointly Gaussian, the orthogonality principle implies independence, and then conditioning does not alter differential entropy.
- The bottom line of p. 14, “Theorem 2.(d)” seems to be a typo.
- The authors seem to be somewhat arbitrary in notations when writing integrals and derivatives; for example, in (2.152) and other places.
- It is not quite clear why specific emphasis has been placed on the technical gap in Kim’s paper. The authors basically considered a different model from Kim’s, and adopted a different technical approach.
Author Response
We wish to thank the reviewer for the positive comments and suggestions.
Comment 1: Def. 2, why both (1.19) and (1.20) involve the same noise W_t? In state space models it is usually the case where the state noise and the observation noise are independent?
Response 1: On page 5, in the definition we clarified that one can also consider the special case when the two noises are independent. Basically, the reason we use this model is due to its generality and the fact that such a model is also considered in the literature. The state space of the ARMA noise on page 3 has such a form.
Comment 2: In (1.28) justification of (a), the orthogonality principle itself is insufficient to eliminate the conditioning. Since everything is jointly Gaussian, the orthogonality principle implies independence, and then conditioning does not alter differential entropy.
Response 2: We clarified this below (1.31), page 7.
Comment 3: The bottom line of p. 14, “Theorem 2.(d)” seems to be a typo.
Response 3: The typo is corrected, page 16.
Comment 4: The authors seem to be somewhat arbitrary in notations when writing integrals and derivatives; for example, in (2.152) and other places.
Response 4: We use a standard notation for Prob{X \in dx}, the probability of RV taking values in dx. We also defined these distributions, see page 24, (2.146).
Comment 5: It is not quite clear why specific emphasis has been placed on the technical gap in Kim’s paper. The authors basically considered a different model from Kim’s, and adopted a different technical approach.
Response 5: The reason is because ref.[4] claimed to have solved the problem of Cover and Pombra ref.[5]. Our paper shows that this is not the case. Ref.[4] solved a much easier problem based on Case II) formulation, i.e., the code of Definition 3, under Conditions 1) and 2) page 6. We further explain this in Remark 1, page 10, and included Corollary 13, page 50. We believe it is important to point out this fact, especially since other researchers are not aware of this.